# Accurate Large-sample Uncertainty Quantification using Stochastic Gradient Markov Chain Monte Carlo

**Yu Wang** [1]  **Jie Ding** [2]  **Jonathan H. Huggins** [1][3]

## Abstract

Tuning algorithms such as stochastic gradient descent (SGD) and stochastic gradient Langevin dynamics (SGLD) for approximate sampling and uncertainty quantification remains challenging, particularly in the practically relevant settings when the batch size is large or the model is misspecified. Existing theory that provides tuning guidance relies on continuous-time limits or strong statistical assumptions, which can become quantitatively inaccurate in these regimes. We address these shortcomings by proposing new discrete-time approximations to SG(L)D with and without momentum, which enables accurate predictions of the stationary covariance, iterate average covariance, and integrated autocorrelation time. Moreover, we prove quantitative, non-asymptotic error bounds showing that these estimates are sufficiently accurate for practical tuning and uncertainty quantification. Numerical experiments demonstrate that our theory yields improved tuning guidance across a range of models and data-generating distributions where existing approaches fail, including when using the $\beta$-divergence rather than log-loss to obtain statistically robust inferences.

## 1. Introduction

Stochastic gradient–based methods have become the default tool for large-sample optimization in machine learning. Algorithms such as stochastic gradient descent (SGD) and its variants dominate modern practice because subsampling dramatically reduces per-iteration computational cost while having strong empirical performance and favorable generalization properties (Bottou, 2010; Hardt et al., 2016; Goodfellow et al., 2016).

From a Bayesian perspective, subsampling-based Markov chain Monte Carlo (MCMC) methods seem to offer an analogous path toward scalable sampling and uncertainty quantification (UQ). In particular, stochastic gradient MCMC (SG-MCMC) algorithms such as stochastic gradient Langevin dynamics (SGLD) replace full-data likelihood gradients with unbiased minibatch estimates, promising posterior sampling at a computational cost comparable to SGD (Welling & Teh, 2011; Li et al., 2016; Raginsky et al., 2017; Brosse et al., 2018; Nemeth & Fearnhead, 2021). In practice, however, SG-MCMC methods are notoriously difficult to tune because the step size, batch size, and temperature parameters must be carefully chosen to control discretization bias and mixing behavior while simultaneously providing accurate UQ (Nemeth & Fearnhead, 2021; Coullon et al., 2023; Negrea et al., 2023; Rajpal et al., 2025; Kim et al., 2024; Mauri & Zanella, 2024; Alexos et al., 2022; Paulin et al., 2025; Akyildiz & Sabanis, 2024). These challenges are exacerbated when the statistical model is misspecified, a setting in which standard Bayesian posteriors are no longer well-calibrated. The same calibration issue applies when using a generalized Bayesian loss, whether the model is correctly specified or not (Bissiri et al., 2016; Jewson et al., 2018).

Recent work has begun to address these challenges by explicitly combining algorithmic and statistical asymptotic perspectives. For example, Mandt et al. (2017) adopted a heuristic perspective that was motivated by two lines of work. The first considers scaling limits in stochastic approximations and show that, after appropriate rescaling of space and time, the iterates jointly converge to a continuous-time Ornstein–Uhlenbeck process (Kushner & Huang, 1981; Pflug, 1986; Walk, 1977; Kushner & Yang, 1993; Kushner & Yin, 2003). The second concerns the asymptotics of the Bayesian posterior, known as Bernstein–von Mises (or Bayesian Central Limit) theorems (Kleijn & van der Vaart, 2012; Van der Vaart, 2000).

More recently, Negrea et al. (2023); Wang et al. (2025) formalize and extend the heuristic arguments of Mandt et al. (2017) by analyzing stochastic gradient algorithms through joint limits in which both the dataset size and algorithm parameters (e.g., step size and batch size) scale together.

[1]Department of Mathematics & Statistics, Boston University [2]Questrom School of Business, Boston University [3]Faculty of Computing & Data Sciences, Boston University. Correspondence to: Jonathan H. Huggins <huggins@bu.edu>.

*Proceedings of the $43^{rd}$ International Conference on Machine Learning*, Seoul, South Korea. PMLR 306, 2026. Copyright 2026 by the author(s).

Further, Wang & Huggins (2026) extend the results of Negrea et al. (2023) to models with local latent variables. These results, which characterize the limiting stochastic process of the iterate sample paths, make it possible to not only determine the limiting stationary distribution (which is important for UQ) but also the mixing time and iterate average distribution, which determine the algorithm's computational efficiency and the accuracy of posterior expectation estimates. Hence, these results are able to provide precise tuning advice that maximizes computational efficiency while targeting the desired form of UQ such as frequentist coverage (White, 1982), Bayesian model uncertainty (Kleijn & van der Vaart, 2012), or both (Huggins & Miller, 2024).

A major limitation of these results, however, is that they rely on taking continuous-time stochastic differential equation (SDE) limits, which approximate discrete-time algorithms only in the vanishing step-size regime (Wang et al., 2025; Li et al., 2019). These limiting approximations become quantitatively inaccurate precisely in the large batch-size regimes most relevant to practice. The problem is that using a large batch size requires using a relatively large step size (Goyal et al., 2017; Negrea et al., 2023), so continuous-time approximations can substantially mischaracterize stationary covariance structure, which can result in inaccurate UQ.

Figure 1 illustrates how the these issues can arise even in simple misspecified linear models. In this example, as the batch size increases, the accuracy of the tuning rules derived from SDE limits decreases rapidly, leading to the stationary covariance failing to match the sandwich covariance $\mathcal{S}_\star$ (White, 1982). Such failures persist even with increasing data size, highlighting a fundamental limitation of continuous-time approximations for guiding practical tuning decisions (Wang et al., 2025).

Recent work has used discrete-time approximations to stochastic gradient algorithms that remain valid at large batch sizes and/or large step sizes (Dieuleveut et al., 2020; Liu et al., 2021; Ziyin et al., 2022). While promising, existing results either assume a constant noise covariance, apply only to linear models, or to not account for model misspecification. Moreover, most approximations lack rigorous non-asymptotic error guarantees; and none provide estimates for the mixing time or iterate-average distribution. Figure 1 illustrates how, as a result, they can fall short of providing reliable guidance for uncertainty quantification – in this case, due to model misspecification.

In this work, we address these limitations by developing a discrete-time theoretical framework for SGD and SGLD that remains accurate at large batch sizes, and under model misspecification. Table 1 compares our approach to alternatives. Our contributions are as follows:

1. **(minor)** We introduce a *proxy algorithm framework* that clarifies the differences and limitations of existing

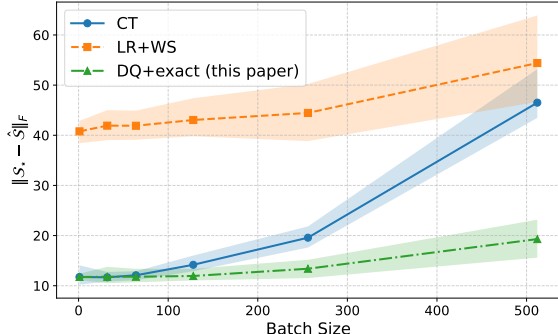

*Figure 1.* Misspecified linear regression with heteroskedastic noise. Data are generated according to $y_n \sim \mathcal{N}(x_n^\top \theta_\star, 1 + \|x_i\|_2^2)$, where $\theta_\star \sim \mathcal{N}(0, I_D)$ is fixed and $x_n \overset{\text{iid}}{\sim} \mathcal{N}(0, I_D)$. A linear model is fitted using constant-step-size SGD. $\mathcal{S}_\star = \mathcal{J}_\star^{-1} \mathcal{I}_\star \mathcal{J}_\star^{-1}$: sandwich covariance; $\hat{\mathcal{S}}$: covariance obtained under step-size tuning rules derived from different theories.

approaches, and thereby helps identify where further theory is needed. (Section 3)

2. **(major)** We derive a *new discrete-time approximation for SGD and SGLD* (with and without momentum) that remains accurate for large batch sizes and misspecified models. (Section 4)

3. **(major)** We provide *quantitative, non-asymptotic error analyses* demonstrating that the resulting stationary covariance estimates are sufficiently accurate for practical tuning *for the purpose of sampling and uncertainty quantification*. (Section 4.2)

4. **(major)** We use our results to propose a practical, tuning-free procedure for scalable uncertainty quantification (Algorithm 1). Through numerical experiments, we show that our theory provides improved tuning guidance for a different models, batch size regimes, and loss functions. (Section 6)

5. **(minor)** Finally, while our focus in the paper is on uncertainty quantification and sampling, our results also shed light on the training dynamics and generalization behavior of SGD and its use for frequentist inference (Jantre et al., 2024; Hwang et al., 2022; Chang et al., 2017; Lyle et al., 2020; Mandt et al., 2017; Zhu et al., 2019; Lewkowycz et al., 2020; Keskar et al., 2017; Hoffer et al., 2017; Mori & Ueda, 2020). For completeness, we illustrate some of these directions, which may be of interest to the wider ML community, through some preliminary experiments (Appendix E).

*Table 1.* Comparison of approximations used to tune SG(L)D for sampling. References are to the works most directly relevant to tuning. **Large batch:** Is the approach accurate for large batch sizes? **Non-const. noise:** Does the approach account for non-constant stochastic gradient noise? **General model/loss:** Does the approach account for model misspecification or the use of a generalized loss? **Mixing:** Does the approach provide mixing time and iterate average covariance estimates? **Bounds:** Are quantitative error bounds available?

| Approach | Large batch | Non-const. noise | General model/loss | Mixing | Bounds |
|---|---|---|---|---|---|
| **Continuous-time** (Mandt et al., 2017; Negrea et al., 2023; Wang et al., 2025) | ✗ | ✗ | ✓ | ✓ | ✓ |
| **Discrete quadratic + constant noise** (Dieuleveut et al., 2020; Liu et al., 2021) | ✓ | ✗ | ✓ | ✗ | ✓ |
| **Linear regression + well-specified** (Ziyin et al., 2022) | ✓ | ✓ | ✗ | ✗ | ✗ |
| **Discrete quadratic + exact noise** (this work) | ✓ | ✓ | ✓ | ✓ | ✓ |

## 2. Background

### 2.1. Setting

Let $\{x_n\}_{n=1}^N$ denote the observed data with $x_n \in \mathbb{X}$. For parameter $\theta \in \mathbb{R}^D$, assume an observation-level differentiable loss or negative log-likelihood $\ell : \mathbb{X} \times \mathbb{R}^D \to \mathbb{R}$, and regularizer $\mathcal{R} : \mathbb{R}^D \to \mathbb{R}$, which in the sampling setting we should interpret as a negative log prior $-\log \pi_0(\theta)$ (up to an additive constant). Together, these lead to the negative potential (or loss)

$$\mathcal{L}(\theta) := N^{-1} \sum_{n=1}^N \ell(x_n, \theta) + N^{-1}\mathcal{R}(\theta).$$

Define the stochastic gradient

$$G_t(\theta) := B^{-1} \sum_{n \in S_t} \nabla \ell(x_n, \theta) + N^{-1}\nabla \mathcal{R}(\theta),$$

where $S_t = \{I_{t1}, I_{t2}, \dots, I_{tB}\}$ is a set of $B$ independent random integers sampled uniformly from $\{1, \dots, N\}$ either with or without replacement. *Stochastic gradient Langevin dynamics* (SGLD; Welling & Teh, 2011) is a Markov chain Monte Carlo (MCMC) algorithm with the single-step update equation

$$\theta_t = \theta_{t-1} - \Lambda \, G_t(\theta_{t-1}) + \sqrt{2\beta^{-1}\Lambda}\, \xi_{t-1}, \qquad (1)$$

where $\Lambda \in \mathbb{R}^{D \times D}$ is a positive definite step size matrix, $\beta \in (0, \infty]$ is the inverse temperature (canonically set to $\beta = N$), and $\xi_{t-1} \overset{\text{iid}}{\sim} \mathcal{N}(0, I)$. SGLD is the prototypical example of a *subsampling MCMC* algorithm, variants of which have been applied for learning a wide variety of large-sample models (Ahn et al., 2012; Nemeth & Fearnhead, 2021; Aicher et al., 2025; Kim et al., 2024; Rajpal et al., 2025; Mauri & Zanella, 2024; Alexos et al., 2022; Paulin et al., 2025). If $\beta = \infty$ (with $1/\infty := 0$), then SGLD reduces to SGD. Setting $\Lambda = \lambda I_D$ for some $\lambda > 0$ results in the usual formulation of SG(L)D with fixed step size $\lambda$.

*Remark* 2.1. We focus on the fixed step size case in this work. While diminishing step sizes guarantee asymptotic exactness by driving stochastic-gradient noise and discretization error to zero, using a fixed step size usually leads to substantially faster convergence (Dieuleveut et al., 2020; Vollmer et al., 2016; Teh et al., 2016; Merad & Gaïffas, 2025) and empirically leads to better generalization by discouraging convergence to sharp minima (Keskar et al., 2017), instead biasing iterates toward flatter solutions that tend to have larger posterior mass (MacKay, 1992; Rissanen, 1983).

### 2.2. Uncertainty Quantification

Both SGD and SGLD have been used for quantifying uncertainty about model parameters (Welling & Teh, 2011; Ahn et al., 2012; Nemeth & Fearnhead, 2021; Mandt et al., 2017). Assuming observations are i.i.d. from an unknown distribution $P_\star$, then the optimal parameter is given by $\theta_\star := \arg\min_\theta \mathbb{E}[\ell(X, \theta)]$, where $X \sim P_\star$. In the Bayesian setting, the Bernstein-von Mises theorem states that the posterior is approximately $\mathcal{N}(\widehat{\theta}, \mathcal{J}_\star^{-1}/N)$, where $\mathcal{J}_\star := \mathbb{E}[\nabla_\theta^2 \ell(X, \theta_\star)]$ (Kleijn & van der Vaart, 2012). Thus, one possible goal when using SG(L)D is to obtain samples with a distribution that is approximately equal to $\mathcal{N}(\widehat{\theta}, \mathcal{J}_\star^{-1}/N)$. However, the sampling distribution of $\widehat{\theta}$ is asymptotically normal with mean $\theta_\star$ and covariance equal to $\mathcal{J}_\star^{-1}\mathcal{I}_\star\mathcal{J}_\star^{-1}/N$, where $\mathcal{I}_\star := \mathbb{E}[\nabla_\theta \ell(X, Y, \theta_\star)\nabla_\theta \ell(X, Y, \theta_\star)^\top]$ (White, 1982). The matrix $\mathcal{J}_\star^{-1}\mathcal{I}_\star\mathcal{J}_\star^{-1}$ is known as the "sandwich" covariance matrix, and it suggests that for proper uncertainty quantification we want the stationary SG(L)D distribution to be approximately $\mathcal{N}(\widehat{\theta}, \mathcal{J}_\star^{-1}\mathcal{I}_\star\mathcal{J}_\star^{-1}/N)$. When the model is correctly specified, $\mathcal{I}_\star = \mathcal{J}_\star$, so the sandwich covariance is equal to $\mathcal{J}_\star^{-1}$ and the Bayesian posterior (and the Laplace approximation) provides correct uncertainty quantification. In the case of a generalized loss, there is no notion of well-specification, and so the "model covariance" $\mathcal{J}_\star^{-1}$ is not a coherent target for uncertainty quantification (Bissiri et al., 2016; Jewson et al., 2018). Therefore, tuning SG(L)D to satisfy $\Sigma_\theta \approx \mathcal{J}_\star^{-1}\mathcal{I}_\star\mathcal{J}_\star^{-1}/N$ will capture the sampling uncertainty in both the model-based and generalized loss settings.

# 3. Proxy Algorithms

Given the extensive use of SGD and SGLD, both algorithms have been studied from a wide variety of perspectives. Many such analyses can be viewed as proposing a *proxy algorithm*: an alternative stochastic process that is "close" to the actual algorithm of interest. The idea is to characterize important properties of the proxy algorithm, then argue either heuristically or rigorously that these properties can be transferred back to apply to the original (exact) algorithm. We will focus on proxy algorithms that, at least implicitly, require that the loss is well-approximated by a quadratic function:

$$\mathcal{L}(\theta_t) \approx \tilde{\mathcal{L}}(\theta_t) := \tfrac{1}{2}\big(\theta_t - \widehat{\theta}^{(N)}\big)^\top \widehat{H}\big(\theta_t - \widehat{\theta}^{(N)}\big) + \text{const},$$

where $\widehat{H} := \nabla^2 \mathcal{L}(\widehat{\theta})$ is the Hessian of the loss (evaluated at $\widehat{\theta}$). While such a condition may seem quite limiting, it turns out to be reasonable in many interesting settings.

**Continuous-time proxies.** Perhaps the most popular proxy approach is to replace discrete dynamics of the iterative algorithm by a continuous-time stochastic process (Mandt et al., 2017; Zhu et al., 2019; Negrea et al., 2023). Let $\widehat{C} = \text{Cov}(G_1(\widehat{\theta}))$ denote the gradient noise covariance at the minimizer and let $W_t$ be a $D$-dimensional Brownian motion. Focusing on the case of SGD for clarity, the Ornstein–Uhlenbeck process $(\vartheta_t)_{t \geq 0}$ defined by the stochastic differential equation (SDE)

$$\mathrm{d}\vartheta_t = -\Lambda \widehat{H}\vartheta_t \mathrm{d}t + \Lambda \widehat{C}^{1/2}\mathrm{d}W_t, \qquad (2)$$

provides a proxy to the discrete-time dynamics after appropriate rescaling and discretization. [1] This approach can be made rigorous via both numerical analysis and statistical (large-sample) perspectives (Kushner & Yang, 1993; Kushner & Huang, 1981; Kushner & Yin, 2003; Negrea et al., 2023; Wang et al., 2025). Similar types of arguments have also been widely used to study MCMC algorithms that do not use subsampling (Roberts & Rosenthal, 1998; Dalalyan, 2017; Roberts & Rosenthal, 2001; Wibisono, 2018).

The continuous-time approach is appealing because $(\vartheta_t)_{t \geq 0}$ is a Gaussian process, so its properties are straightforward to analyze (Mandt et al., 2017; Negrea et al., 2023; Kushner & Yang, 1993). For example, if the process has stationary distribution $\pi_\vartheta$, the covariance matrix of the stationary distribution $\Sigma_\vartheta := \text{Cov}(\pi_\vartheta)$ must satisfy $\Sigma_\vartheta \widehat{H} + \widehat{H}\Sigma_\vartheta = \Lambda \widehat{C}$

---

[1] Li et al. (2017; 2019) propose using stochastic modified equations (SMEs) to approximate SGD and perform error analysis. However, SMEs serve as a close approximation to SGD only for small learning rates, making it challenging to justify this approach for non-vanishing values of $\lambda$ (Li et al., 2017). Moreover, in most cases, SMEs are not conducive to exact analysis; this leads to, for example, Li et al. (2019, Section 5.1) focusing on cases with an explicit solution that match Equation (2).

(Gardiner, 1985). In particular, setting

$$\Lambda = (\Sigma\widehat{H} + \widehat{H}\Sigma)\widehat{C}^{-1} \qquad (3)$$

results in a stationary covariance of $\Sigma_\vartheta = \Sigma$.[2] Furthermore, Negrea et al. (2023) show that the asymptotic mixing time is heuristically equal to $2/\lambda_{\min}(\Lambda\widehat{H})$ iterations, where $\lambda_{\min}(A)$ denotes the minimum eigenvalue of matrix $A$. This result suggests that, to optimize mixing time, set $\Lambda \propto \widehat{H}^{-1}$.

**Discrete-time proxies.** The continuous-time proxy approach requires the step size matrix $\Lambda$ to be sufficiently small that *(i)* the continuous-time dynamics (driven by Gaussian noise) is a good approximation to the discrete-time dynamics and *(ii)* the gradient noise is approximately constant (that is, $G_t(\theta_{t-1}) \approx G_t(\widehat{\theta})$ for all $t = 1, \ldots, T$). However, in practice it is often desirable to use a relatively large batch size (e.g, 1%–10% of the data), in which case following the guidance of Negrea et al. (2023) requires the use of a relatively large $\Lambda$ – exactly the regime in which the continuous-time theory often breaks down, leading to inaccurate predictions about real algorithm behavior (Liu et al., 2021; Ziyin et al., 2022). The importance of capturing the location-dependence of noise has been widely observed (Simsekli et al., 2019; 2020; Hodgkinson & Mahoney, 2021; Meng et al., 2020; Mori et al., 2022; Ziyin et al., 2022).

A number of papers aim to overcome these limitations by using the discrete-time proxy algorithm

$$\psi_t = \psi_{t-1} - \tfrac{\Lambda}{B}\sum_{n \in S_t} \widehat{H}_n(\psi_{t-1} - \widehat{\theta}), \qquad (4)$$

where $\widehat{H}_n := \nabla^2\ell(x_n, \widehat{\theta})$. Assuming it exists, let $\pi_\psi$ denote the stationary distribution of the proxy algorithm given in Equation (4), let $\Sigma_\psi := \text{Cov}(\pi_\psi)$, and for $\psi_\infty \sim \pi_\psi$, let $\overline{C}_\psi := \mathbb{E}[\text{Cov}\{G_1(\psi_\infty)\}]$ denote the expected covariance of the gradient noise. Liu et al. (2021) show that the stationary covariance $\Sigma_\psi$ of discrete-time update described in Equation (4) satisfies

$$\Lambda\widehat{H}\Sigma_\psi + \Sigma_\psi \widehat{H}\Lambda = \Lambda\left(\overline{C}_\psi + \widehat{H}\Sigma_\psi \widehat{H}\right)\Lambda. \qquad (5)$$

Dieuleveut et al. (2020) provides exact discrete-time analyses of constant-step stochastic gradient descent, treating SGD as a time-homogeneous Markov chain rather than as a discretization of a continuous-time diffusion. In the quadratic setting, they also show that SGD converges to a stationary distribution whose covariance satisfies Equation (5), as also given by Liu et al. (2021). Notably, the higher-order covariance terms $\Lambda\overline{C}_\psi\Lambda$ – which capture finite learning-rate effects intrinsic to the discrete-time dynamics – is missing from diffusion-based SDE approximations.

---

[2] Or, when we are interested in characterizing the stationary covariance, if $\Sigma_\vartheta$ and $\widehat{H}$ commute, then that $\Sigma_\vartheta = \tfrac{1}{2}\Lambda\widehat{C}\widehat{H}^{-1}$.

A key challenge when using Equation (5) is that it only provides an *implicit* characterization of $\Sigma_\psi$ because the average noise covariance $\overline{C}_\psi$ also depends on the stationary distribution $\pi_\psi$. Hence, $\overline{C}_\psi$ must either be approximated or, in special cases, computed exactly. An important special case is the linear regression model, where $x_n = (z_n, y_n) \in \mathbb{R}^D \times \mathbb{R}$ and the observation-level loss is $\ell(x_n, \theta) = \frac{1}{2\sigma^2}(y_n - \theta^\top z_n)^2$. In a follow-up to Liu et al. (2021), Ziyin et al. (2022) show that, assuming $z_n \sim \mathcal{N}(0, A)$ and the model is well-specified (i.e., $y_n \sim \mathcal{N}(\theta_\star^\top z_n, \sigma^2)$ for some $\theta_\star \in \mathbb{R}^D$), for large $N$,

$$\overline{C}_\psi \approx B^{-1}\left(A\Sigma_\psi A + \mathrm{Tr}\left[A\Sigma_\psi\right]A + \sigma^2 A\right). \quad (6)$$

Using this approximate expected covariance for SGD noise, Ziyin et al. (2022) are able to show, for example, better test loss estimation, the benefits of negative regularization, the role of overparameterization in the steady-state dynamics of SGD, and power-law tail behavior of SGD noise.

Hence, using discrete-time proxies rather than continuous-time ones can lead to more precise tuning advice and new insights. Nevertheless, as summarized in Table 1, existing approaches are not yet sufficiently reliable for practical use: some leave the noise covariance implicit (Dieuleveut et al., 2020), others rely on heuristic approximations to the noise covariance (Liu et al., 2021), and others focus on restricted settings, such as well-specified models with $N \gg D$ (Ziyin et al., 2022). In addition, they do not characterize the mixing time or iterate average error. Our results, which are presented in the next section, address all of these limitations, as described in the last row of Table 1 and illustrated in Figure 1.

## 4. A New Proxy Algorithm for Analyzing SG(L)D

Our approach to creating an improved proxy algorithm is to apply a second-order Taylor approximation to each loss term $\ell_n(\theta) := \ell(x_n, \theta)$:

$$\tilde{\ell}_n(\theta) := \ell_n(\widehat{\theta}) + \nabla\ell_n^\top(\widehat{\theta})(\theta - \widehat{\theta})$$
$$+ \frac{1}{2}(\theta - \widehat{\theta})^\top\nabla^2\ell_n(\widehat{\theta})(\theta - \widehat{\theta}). \quad (7)$$

We apply SG(L)D (with or without momentum) to the approximation, $\tilde{\mathcal{L}}(\theta) := N^{-1}\sum_{n=1}^N \tilde{\ell}_n(\theta) + N^{-1}\mathcal{R}(\theta)$. Letting $\mathcal{J}_n := \nabla^2\ell_n(\widehat{\theta})$ and using Equations (1) and (7), the update equation for our proxy algorithm is

$$\psi_t = \psi_{t-1} - \Lambda\left[G_t(\widehat{\theta}) + \nabla G_t(\widehat{\theta})(\psi_{t-1} - \widehat{\theta})\right]$$
$$+ \sqrt{2\beta^{-1}\Lambda}\,\xi_{t-1}. \quad (8)$$

Assuming the iterates $(\psi_t)_{t\geq 0}$ have a well-defined stationary distribution, the stationary covariance $\Sigma_\psi$ provides an

approximation $\widehat{\Sigma}_\theta := \Sigma_\psi$ to $\Sigma_\theta$. The quadratic form of Equation (7) facilitates analyses that allow us to address limitations of previous work. First, the quadratic loss results is the linear structure of the SG(L)D update given in Equation (8), which makes it amenable to direct analysis. Building on the techniques of previous work (Liu et al., 2021; Ziyin et al., 2022), in Section 4.1 we derive an *exact*, solvable relationship between $\Sigma_\psi$ and $\Lambda$. *Thus, unlike previous results, we do not require any additional assumptions or approximations.*

In addition, the use of a Taylor series approximation for the observation-level losses lends itself to rigorous error analysis. Specifically, we are able to bound the Wasserstein distance between the distributions of $\psi_t$ and $\theta_t$ in Section 4.2. Using this result we obtain relative error bounds on the marginal standard deviation and covariance matrix estimates under standard assumptions, which hold for logistic regression and, assuming a bounded parameter space, for Poisson and gamma regression as well (see, e.g., Brosse et al., 2018; Moulines & Bach, 2011; Toulis et al., 2014):

(A) The observation-level losses $\ell_1, \ldots, \ell_N$ are convex.

(B) For each $n = 1, \ldots, N$, for finite positive $L_n$ and $M_n$, the loss $\ell_n$ is $L_n$-smooth and satisfies $\sup_\theta \sum_{d=1}^D \|\nabla^2(\partial_d\ell_n(\theta))\|^2 \leq M_n^2$.

(C) For some $\mu > 0$, the loss $\mathcal{L}$ is $\mu$-strongly convex.

**Theorem 4.1.** *If Assumptions (A)–(C) hold and $\Lambda = \lambda I_D$ for some $\lambda \in (0, 1/(2L))$, then there exist constants $C_v$ and $C_s$ independent of $\lambda$ such that*

$$\|\Sigma_\theta - \Sigma_\psi\|/\|\Sigma_\theta\| \leq C_v\lambda^{1/2} \quad (9)$$

*and, for $d = 1, \ldots, D$,*

$$|\sigma_{\theta,d} - \sigma_{\psi,d}|/\sigma_{\theta,d} \leq C_s\lambda^{1/2}. \quad (10)$$

Hence, it follows from our results that the approximation $\widehat{\Sigma}_\theta := \Sigma_\psi$ is close enough to $\Sigma_\theta$ to provide a practically useful estimate.

### 4.1. Stationary Analysis

To prove our main result Theorem 4.1, we first obtain an exact relationship between the learning rate matrix $\Lambda$, the stationary covariance $\Sigma_\psi$, and the average noise $\overline{C}_\psi$.

**Proposition 4.2.** *Assuming the iterates $(\psi_t)_{t\geq 0}$ have a well-defined stationary distribution, the stationary covariance $\Sigma_\psi$ satisfies*

$$\Lambda\widehat{H}\Sigma_\psi + \Sigma_\psi\widehat{H}\Lambda = \Lambda\left(\overline{C}_\psi + \widehat{H}\Sigma_\psi\widehat{H}\right)\Lambda + 2\beta^{-1}\Lambda. \quad (11)$$

It follows from Equation (11) that to obtain a solvable relationship between $\Lambda$ and $\Sigma_\psi$ using Proposition 4.2, we must

compute the expected covariance $\overline{C}_\psi$. Such calculation is feasible when using $\mathcal{N}(0, \Gamma^{-1})$ with $\Gamma \in \mathbb{R}^{D \times D}$ positive-definite, as a prior for $\theta$ – that is, using $\mathcal{R}(\theta) = \frac{1}{2}\theta^\top \Gamma \theta$.

**Theorem 4.3.** *For the proxy algorithm Equation (8), if $\mathcal{R}(\theta) = \frac{1}{2}\theta^\top \Gamma \theta^\top$ and the mini-batches are sampled with replacement, then*

$$\overline{C}_\psi = \frac{1}{B}\left( \mathcal{I} - \frac{\|\Gamma\widehat{\theta}\|^2}{N^2} + \frac{1}{N}\sum_{n=1}^{N} \mathcal{J}_n \Sigma_\psi \mathcal{J}_n - \mathcal{J}\Sigma_\psi \mathcal{J} \right), \tag{12}$$

*where $\mathcal{I} := \frac{1}{N}\sum_{n=1}^{N} \nabla\ell_n(\widehat{\theta})\nabla\ell_n(\widehat{\theta})^\top$. If the mini-batches are sampled without replacement, the same result holds but with the right-hand side multiplied by $(N-B)/(N-1)$.*

Plugging Equation (12) into Equation (11) provides an exact relationship between $\Lambda$ and $\Sigma_\psi$. Hence, given a fixed learning rate matrix $\Lambda$ (or a scalar learning rate $\lambda$), we can, in principle, compute the stationary covariance $\Sigma_\psi$ as an estimate for $\Sigma_\theta$.

Our final result in this section improves upon the heuristic mixing time estimate of Negrea et al. (2023) (see Appendix A for more on mixing time). Unlike the stationary covariance case, our result is identical except for an additive $-1$.

**Proposition 4.4.** *Consider the proxy update in Equation (8) and suppose that $0 < \lambda < 2/\mu_{\max}(\hat{H})$, where $\mu_{\max}(A)$ and $\mu_{\min}(A)$ denote the largest and smallest eigenvalues of a matrix $A$, respectively. Under $L$-smoothness, it simplifies to $0 < \lambda < 2/L$, which is consistent with step-size condition used in Dieuleveut et al. (2020). Under this condition, the resulting SG(L)D Markov chain admits a unique stationary distribution $\pi_\theta$. For each $v \in \mathbb{R}^D$, define the projection $f_v(\theta) := v^\top \theta$ and let*

$$\rho_{k,v} := \mathrm{Corr}_{\pi_\theta}(v^\top \theta_0, v^\top \theta_k),$$

*and*

$$\tau_{\mathrm{int}}(f_v) := 1 + 2\sum_{t=1}^{\infty} \rho_{k,v}.$$

*Then the worst-case integrated autocorrelation time $\tau := \sup_v \tau_{\mathrm{int}}(f_v)$ is equal to $2/\mu_{\min}(\Lambda\hat{H}) - 1$ iterations.*

### 4.2. Error Analysis

We assess the accuracy of our proxy algorithm by bounding the 2-Wasserstein distance between the distributions of $\theta_t$ and $\psi_t$. The 2-Wasserstein distance between distributions $\pi$ and $\tilde{\pi}$ is given by

$$W_2(\pi, \tilde{\pi}) = \inf \mathbb{E}(\|\theta - \tilde{\theta}\|^2)^{1/2},$$

where the infimum is over all joint distributions of $(\theta, \tilde{\theta})$ such that $\theta \sim \pi$ and $\tilde{\theta} \sim \tilde{\pi}$. A small Wasserstein distance

between distributions implies the covariance and marginal standard deviations are also close. Let $\sigma_{\theta,d} := \Sigma_{\theta,dd}^{1/2}$ and $\sigma_{\psi,d} := \Sigma_{\psi,dd}^{1/2}$. Then, by Huggins et al. (2020, Theorem 3.4), $W_2(\pi_\theta, \pi_\psi) \leq \varepsilon$ implies that

$$\begin{aligned}
|\sigma_{\theta,d} - \sigma_{\psi,d}| &\leq \varepsilon \ (d = 1, \dots, D) \\
\|\Sigma_\theta - \Sigma_\psi\| &\leq 2\varepsilon(\|\Sigma_\theta\|^{1/2} \wedge \|\Sigma_\psi\|^{1/2} + \varepsilon).
\end{aligned} \tag{13}$$

Hence, bounding $W_2(\pi_\theta, \pi_\psi)$ enables us to bound the error of the proxy stationary covariance $\Sigma_\psi$.

We first give a bound on the Wasserstein distance between the distributions of $\theta_t$ and $\psi_t$, which we denote by, respectively, $\pi_{\theta,t}$ and $\pi_{\psi,t}$.

**Theorem 4.5.** *If Assumptions (A)–(C) hold and $\Lambda = \lambda I_D$ for some $\lambda \in (0, 1/(2L))$, then, letting $\bar{\beta} := 1 - \lambda\mu(1 - 2\lambda L)$, $\overline{M^p} := N^{-1}\sum_{n=1}^{N} M_n^p, p \in \{1, 2\}$, and $C_s := \mathbb{E}(\|\psi_s - \widehat{\theta}\|^4)$, for all $t = 1, 2, \dots,$*

$$W_2^2(\pi_{\theta,t}, \pi_{\psi,t}) \tag{14}$$

$$\leq \bar{\beta}^t W_2^2(\theta_0, \psi_0) + \lambda\left\{ \frac{\lambda\overline{M^2}}{2} + \frac{\overline{M}^2}{4\mu} \right\}\sum_{s=1}^{t} \bar{\beta}^{t-s} C_{s-1}.$$

Theorem 4.5 is quite general, and we conjecture it could be useful beyond our application to bounding the stationary covariance error. Typically we would expect to take $\psi_0 = \theta_0$, in which case the first term on the righthand side of Equation (14) is zero. We note that Theorem 4.5 is similar in spirit to the 2-Wasserstein bound provided by Jin et al. (2024) for a continuous-time Langevin-based proxy algorithm that uses Poissonized data subsampling; however, Jin et al. (2024) do not use their proxy algorithm to estimate the stationary covariance of SG(L)D.

Using Equation (14) to obtain an explicit quantitative bound requires upper-bounding the 4th moment of $\psi_t$, which we do in Lemma F.2. The following corollary gives our main error bound, which for simplicity we state for the case of SGD since the SGLD case is qualitatively identical.

**Corollary 4.6.** *Under the same assumptions as Theorem 4.5 and with $\beta = \infty$ (i.e., for the case of SGD), if $\lambda < \min\{B\hat{\mu}/(200L^2), 1/(4L)\}$, then there exists an explicit constant $A$ given in Equation (F.12) such that $W_2(\pi_\theta, \pi_\psi) \leq A\lambda/B$.*

*Remark* 4.7 (Dimension dependence). Recall that $D$ is the parameter dimension. For $I \sim \mathrm{Unif}(\{1, \dots, N\})$ independent, define the single-sample stochastic gradient $g_I := \nabla\ell(x_I, \widehat{\theta})$. Suppose that $g_I$ satisfies the mild scaling requirement $\mathbb{E}[\|g_I\|^2] = O(D)$ and $\mathbb{E}[\|g_I\|^4] = O(D^2)$. Such a condition holds, for example, for generalized linear model fit to data with sub-Gaussian covariate distribution (Vershynin, 2018). Then $W_2(\pi_\theta, \pi_\psi) \leq C D (\lambda/B + 1/\beta)$ with $C$ independent of $D$. Thus, our proxy algorithm remains accurate

**Algorithm 1** SG(L)D with Target Covariance Tuning.

**DQ+exact:** discrete quadratic + exact noise (this work).
**CT:** continuous time. **DQ+const:** discrete quadratic + constant noise. **LR+WS:** linear regression + well-specified.

**Require:** Dataset $\{x_n\}_{n=1}^N$, tuning method choice, per-sample loss $\ell(\theta; x)$, offline subsample size $M$, inverse temperature $\beta$, batch size $B$, number of iteration $T$

  **Step 1: Offline UQ tuning**
1: Subsample $M$ observations $\{x'_m\}_{m=1}^M \subseteq \{x_n\}_{n=1}^N$
2: Use subsample to obtain MAP estimate $\hat{\theta}$
  *Estimate sandwich covariance at $\hat{\theta}$ for UQ:*
3: $\quad \widehat{\mathcal{J}} \leftarrow \frac{1}{M} \sum_{m=1}^M \nabla^2 \ell(\hat{\theta}; x'_m)$
4: $\quad \widehat{\mathcal{I}} \leftarrow \frac{1}{M} \sum_{m=1}^M \nabla \ell(\hat{\theta}; x'_m) \nabla \ell(\hat{\theta}; x'_m)^\top$
5: $\quad \widehat{\mathcal{S}} \leftarrow \widehat{\mathcal{J}}^{-1} \widehat{\mathcal{I}} \widehat{\mathcal{J}}^{-1}$
6: *Determine best $\Lambda$ using chosen tuning method:*
  **DQ+exact:** solve eqs. (11) and (12) with $\Sigma_\psi = \widehat{\mathcal{S}}$
  **CT:** use eq. (3) with $\Sigma = \widehat{\mathcal{S}}$
  **DQ+const:** solve eq. (5) with $\Sigma_\psi = \widehat{\mathcal{S}}$ and $\overline{C}_\psi = \widehat{\mathcal{J}}$
  **LR+WS:** solve eqs. (5) and (6) with $\Sigma_\psi = \widehat{\mathcal{S}}$
  **Step 2: Preconditioned SG(L)D sampling**
7: Initialize $\theta_0 \leftarrow \hat{\theta}$ (or any warm start).
8: **for** $t = 1$ **to** $T$ **do**
9: $\quad$ Sample minibatch $\mathcal{B}_t \subset \{1, \dots, N\}$ with $|\mathcal{B}_t| = B$.
10: $\quad$ Compute gradient $g_t \leftarrow \frac{1}{B} \sum_{n \in \mathcal{B}_t} \nabla \ell(\theta_t; x_n)$
11: $\quad$ Sample update $\theta_t \sim \mathcal{N}(\theta_{t-1} - \Lambda g_t, 2\beta^{-1}\Lambda)$
12: **end for**
13: **return** $\{\theta_t\}_{t=1}^T$ and $\Lambda$.

in high dimensions provided that $\lambda/B + 1/\beta \ll 1/(CD)$. Since typically $\lambda = O(1/N)$ and $\beta = \infty$ (for SGD) or $\beta = N$ (for SGLD), it follows that in that case we require either *(i)* $N \gg CD$ for SGD or SGLD, or *(ii)* $NB \gg CD$ for SGD. Note that the latter case supports high-dimensional problems as long as the batch size is sufficiently large. See Appendix C for further details and discussion, including the sparse high-dimensional regime.

### 4.3. SGLD with Momentum

Our theoretical results extend to the case of SG(L)D with momentum. These extensions are tight, in the sense that we recover our non-momentum results as special cases. Due to space limitations, we defer details to Appendix B.

## 5. A General Procedure for Calibrated SG(L)D Sampling

Algorithm 1 outlines a practical tuning procedure for SG(L)D uncertainty calibration, which covers all the approaches listed in Table 1. When tuning of $\Lambda$ using our proposed approach (DQ+exact), Algorithm 1 is applicable to the large-sample, low-to-moderate dimensional

regime. Its computational complexity is $O(MD^2 + D^3) + O(T(BD + D^2))$ where the first term corresponds to a one-time offline cost using a subsample of size $M$, and the second term is the cost of $T$ stochastic gradient iterations with minibatch size $B \ll N$. The $O(D^3)$ term is incurred only once and becomes negligible when $N \gg D^3$. Since the mixing time of tuned SG(L)D is $O(1)$ epochs (equivalently $O(N/B)$ iterations), relative Monte Carlo error $\delta$ is achievable with $T = O(N/[B\delta])$ iterations. Hence, the overall computational complexity is $O(N[D + D^2/B]/\delta)$. This result also suggests a benefit to using a large batch size of at least $B \gg D$ to reduce the number of preconditioning operations, improving the computational efficiency of SG(L)D.

The tuning procedure also requires $O(D^2)$ memory to store and manipulate quantities such as Hessian $\hat{\mathcal{J}}$ and Fisher information $\hat{\mathcal{I}}$, making accurate computation challenging in very high-dimensional settings. A promising future direction is to develop structured, low-rank, diagonal, or trajectory-based approximations of $\hat{\mathcal{J}}$ or $\hat{\mathcal{I}}$ computations to improve scalability.

## 6. Experiments

We compare the accuracy of the learning rate tuning guidance provided by our theory versus previous work (see Table 1). For fair comparison, we follow Algorithm 1, with the only difference across approaches being how $\Lambda$ is determined. In our experiments, we use SGD (so, $\beta = \infty$). The code for all experiments is publicly available at https://github.com/wangyu1369/large-sample-sgmcmc-uq.

In our experiments we compute $\Lambda$ using a numerical optimization procedure since obtaining a close-form solution is challenging (Hammarling, 1982; Ye et al., 1998). Specifically, we substitute the stationary noise expression from Equation (12) into the stationary covariance equation Equation (11) and set $\Sigma_\psi = \widehat{\mathcal{S}}$. This yields a matrix equation of the form $F(\Lambda) = 0$, where $\Lambda$ is the only unknown. We solve this system numerically by vectorizing $\Lambda$ and applying scipy.optimize.root with the Powell hybrid method. While our approach requires solving jointly Equations (11) and (12), this cost is incurred only once per problem and is negligible compared to the dominant cost of running SG(L)D trajectories. We empirically verify this in Section D.2.

### 6.1. Robust Linear Regression

While standard Bayesian inference and maximum likelihood estimation optimize the KL divergence, the resulting log loss is notoriously sensitive to outliers and misspecification, allowing a single atypical datapoint to dom-

*Table 2.* Results for linear regression experiments with simulated data. Calibration error is the Kolmogorov–Smirnov distance to Unif$(0, 1)$. Covariance error is $\|\mathcal{S}_\star - \hat{\mathcal{S}}\|_F / \|\mathcal{S}_\star\|_F$. Within each metric row and loss block, for a fixed batch size $B$, bold indicates methods whose 95% confidence intervals overlap with the confidence interval of the method with the lowest mean error. Confidence intervals are computed over 30 independent runs and are reported in the full table in Appendix D. Sandwich Gauss is included as the target sandwich Gaussian reference, while NUTS and the exact posterior are included to illustrate the discrepancy between the posterior distribution and the sandwich target.

| | Log loss | | | | $\beta$-loss ($\beta = 1.5$) | | | | |
|---|---|---|---|---|---|---|---|---|---|
| $B$ | Posterior | CT | LR+WS | DQ+exact | NUTS | Sandwich Gauss | CT | LR+WS | DQ+exact |
| **Calibration error** | | | | | | | | | |
| 16 | 0.418 | **0.171** | 0.529 | **0.169** | 0.195 | 0.156 | **0.201** | **0.178** | **0.172** |
| $\lfloor 0.1 \times N \rfloor$ | 0.418 | **0.179** | 0.517 | **0.174** | 0.195 | 0.156 | **0.196** | **0.177** | **0.190** |
| **Covariance error** | | | | | | | | | |
| 16 | 0.943 | **0.672** | 0.995 | **0.664** | 0.795 | 0.000 | **0.640** | 1.115 | **0.695** |
| $\lfloor 0.1 \times N \rfloor$ | 0.943 | 0.975 | 0.996 | **0.672** | 0.799 | 0.000 | 1.006 | 1.322 | **0.748** |

*Table 3.* Results for linear regression experiments with Boston housing data. See Table 2 caption for further explanation. $\infty$ denotes divergence under this tuning guidance.

| | Log loss | | | | $\beta$-loss ($\beta = 1.5$) | | | | |
|---|---|---|---|---|---|---|---|---|---|
| $B$ | Posterior | CT | LR+WS | DQ+exact | NUTS | Sandwich Gauss | CT | LR+WS | DQ+exact |
| **Covariance error** | | | | | | | | | |
| 16 | 0.358 | **0.247** | $9.23 \times 10^8$ | **0.337** | 2.528 | 0 | **2.054** | $\infty$ | **2.782** |
| $\lfloor 0.1 \times N \rfloor$ | 0.358 | 0.589 | $1.40 \times 10^7$ | **0.352** | 2.528 | 0 | 3.126 | $\infty$ | **1.398** |

*Table 4.* Results for Poisson regression experiments. See Table 2 caption for further explanation.

| | | Simulated | | Credit |
|---|---|---|---|---|
| $B$ | method | calib. err. | cov. err. | cov. err. |
| 16 | CT | **0.069** | **0.207** | **0.132** |
| | DQ+const | 0.646 | 0.672 | 0.982 |
| | DQ+exact | **0.074** | **0.208** | **0.157** |
| $\lfloor 0.1 \times N \rfloor$ | CT | 0.089 | 0.230 | 0.191 |
| | DQ+const | 1.376 | 0.991 | 0.997 |
| | DQ+exact | **0.075** | **0.211** | **0.154** |

inate the gradient. The $\beta$-divergence provides a robust alternative by downweighting low-probability observations through a tunable power parameter $\beta$, effectively controlling heavy-tailed effects (Ghosh & Basu, 2016; Jewson et al., 2018; 2024). We consider a regression setting with observations $x = (y, z)$ where $y$ denotes the response and $z$ the covariates. Then the $\beta$-divergence loss is defined as $\ell(\theta; x) = -\frac{1}{\beta - 1} f(y; \theta, z)^{\beta - 1} + \frac{1}{\beta} \int f(y'; \theta, z)^\beta \, dy'$, where $f(y; \theta, z)$ denotes the likelihood. For a detailed discussion of tuning guidance under the $\beta$-divergence loss, see Section D.3.

***Simulated misspecified data with outliers.*** First, we consider a misspecified linear regression model with heteroskedastic errors. Data $\{(x_n, y_n)\}_{n=1}^N$ are generated according to $y_n \mid x_n \sim \mathcal{N}(x_n^\top \theta_\star, 1 + \|x_n\|_2^2)$, where the true parameter $\theta_\star \sim \mathcal{N}(0, I_D)$ is fixed throughout the experiment, and the covariates are drawn independently as $x_n \sim \mathcal{N}(0, I_D)$.

To model heavy-tailed contamination, a fraction $p \in [0, 1]$ of samples is selected uniformly at random and replaced by outliers. For these contaminated observations, responses are generated as $y_n \mid x_n \sim \mathcal{N}(x_n^\top \theta_\star + b, \ s^2(1 + \|x_n\|_2^2))$, where $s > 1$ controls variance inflation and $b$ introduces a mean shift. Unless otherwise specified, we set $D = 50$, $N = 5000$, $p = 0.01$, $b = 5.0$, and $s = 5.0$.

As shown in Table 2, LR+WS performs poorly under log loss due to its reliance on well-specified model assumptions. Replacing the log loss with the $\beta$-divergence substantially improves its quantile calibration. However, comparable calibration does not imply accurate uncertainty quantification: existing tunings (CT and LR+WS) exhibit significant covariance mismatch, particularly for large batch sizes. In contrast, our tuning consistently yields a stationary covariance closest to the target sandwich covariance $\mathcal{S}_\star$, with the largest gains observed in the large batch size regime.

***Boston housing data*** (**Harrison & Rubinfeld, 1978**)  We next consider the Boston housing dataset, where the linear model is strongly misspecified. In this setting, LR+WS becomes unstable and produces extremely large covariance errors. These results corroborate the simulated experiments and highlight the importance of our tuning guidance for accurate uncertainty quantification in misspecified and large-batch regimes.

### 6.2. Poisson Regression

Finally, we demonstrate how our theory provides tuning guidance for the more challenging case of Poisson regression.

***Simulated data.***  We first consider simulated data generated from the assumed model $y_n \sim \text{Poisson}(\exp\{x_n^\top \theta_\star\})$, where $\theta_\star \sim \mathcal{N}(0, I_D)$ and $x_n \overset{\text{iid}}{\sim} \mathcal{N}(0, I_D)$. We use a sample size of $N = 5{,}000$ and set dimension $D = 50$.

***Credit data*** (**Hofmann, 1994**)  The German credit data contains data on $D = 20$ variables and the credibility of $N = 1{,}000$ loan applicants.

For both, we consider batch sizes $B \in \{16, \lfloor 0.1 \times N \rfloor\}$. The results in Table 4 show that while continuous-time tuning remains competitive for small batch sizes, its accuracy degrades substantially as the batch size increases, leading to larger covariance and calibration errors. In contrast, the tuning guidance derived from our discrete-time theory consistently yields more accurate uncertainty quantification in the large-batch regime. Existing discrete-time approaches based on quadratic objectives and constant noise suffer from severe miscalibration once these restrictive assumptions are violated. The improved calibration and covariance accuracy demonstrate that our method provides reliable UQ beyond the regimes where continuous-time or constant-noise approximations apply.

### 6.3. Neural Network

We further compare the stationary covariance predicted by the different theories in Table 1 on a real-world neural network task. Specifically, we fit a two-hidden-layer $\texttt{tanh}$ neural network with hidden widths $(2, 3)$ on the *Diabetes* dataset (**Efron et al., 2004**).

As we can see from Figure 2, at small learning rates, all methods are comparable. However, as the learning rate increases, continuous-time approximations become quantitatively inaccurate, while our discrete-time method remains accurate. While our non-asymptotic error analysis assumes convexity, the characterization of the stationary covariance $\Sigma_\psi$ (Proposition 4.2) and minibatch noise $\overline{C}_\psi$ (Theorem 4.3) does not rely on this assumption. Empirically, we observe

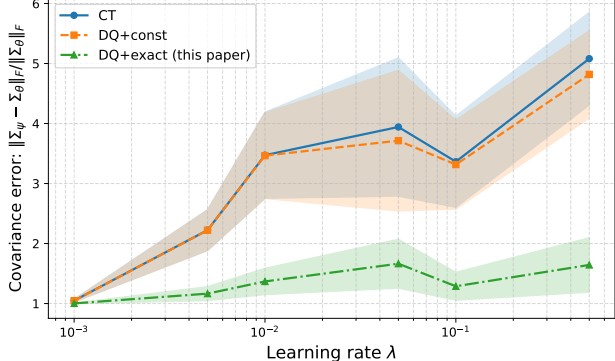

*Figure 2.* Covariance prediction error for neural network with hidden on the *Diabetes* dataset. The error is measured as $\|\Sigma_\psi - \Sigma_\theta\|_F / \|\Sigma_\theta\|_F$, where $\Sigma_\theta$ is the empirical stationary covariance estimated from SGD tail iterates and $\Sigma_\psi$ is the covariance predicted by each theory. Shaded regions denote $95\%$ confidence intervals for the mean across 30 independent repetitions.

that our discrete-time proxy remains effective for neural networks, suggesting that the approach can potentially extend beyond the convex setting and motivating future work on non-convex analysis.

## 7. Conclusion

We study uncertainty quantification for stochastic gradient methods from a discrete-time perspective. Our results show that accurate characterization of SGD and SGLD stationary behavior requires moving beyond continuous-time approximations, particularly at large batch sizes and non-vanishing learning rates. By explicitly modeling the stationary covariance and minibatch-induced noise structure, our framework provides principled and practical tuning strategies for SGD and SGLD under both well-specified and misspecified settings. Empirically, we demonstrate improved covariance estimation and calibration across synthetic and real-world tasks when using Algorithm 1.

A limitation of our finite-sample analysis is that it relies on strong convexity assumptions. Empirically, we observe that the proposed discrete-time proxy remains effective for neural networks (Section 6.3), suggesting that the approach may extend beyond convex settings. Developing finite-sample guarantees for non-convex models remains an important direction for future work. Another natural extension is to characterize more precisely the regimes of learning rate and batch size in which continuous-time approximations fail, and to identify sharp thresholds at which discrete-time effects dominate stationary behavior.

## Acknowledgments

Y. Wang and J. H. Huggins were partially supported by National Science Foundation CAREER award IIS-2340586.

## Impact Statement

This paper presents work whose goal is to advance the field of probabilistic machine learning. There are many potential societal consequences of our work, none which we feel must be specifically highlighted here.

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

## A. Discussions on Mixing Speed

Beyond matching a desired stationary covariance, practical uncertainty quantification also requires that the SG(L)D Markov chain mixes rapidly so that Monte Carlo estimates are accurate at low computational cost. Let $(\theta_t)_{t \geq 0}$ denote the SG(L)D iterates and let $\pi_\theta$ be their stationary distribution. Given a scalar functional $f : \mathbb{R}^D \to \mathbb{R}$, let $\pi_\theta(f) := \int f(\theta)\,\pi_\theta(\mathrm{d}\theta)$ denote its expectation under the invariant distribution, which is the quantity we ultimately want to estimate. Given $T$ iterates, the standard Monte Carlo estimator for $\pi_\theta(f)$ is $\hat{f}_T := T^{-1} \sum_{t=1}^{T} f(\theta_t)$.

To isolate the effect of mixing, suppose the chain is started at stationarity: $\theta_0 \sim \pi_\theta$. Letting $\rho_k(f) := \mathrm{Corr}_{\pi_\theta}(f(\theta_0), f(\theta_k))$ denote the lag-$k$ autocorrelation of the stationary time series $(f(\theta_t))_{t \geq 0}$, the *integrated autocorrelation time*

$$\tau_{\mathrm{int}}(f) := 1 + 2 \sum_{t=1}^{\infty} \rho_k(f)$$

(Geyer, 1992; Sokal, 1997) quantifies how much serial dependence inflates Monte Carlo variance relative to $T$ i.i.d. draws: rapid mixing corresponds to fast decay and/or negative values of $\rho_t(f)$, which results in small $\tau_{\mathrm{int}}(f)$. In particular, $\hat{f}_T$ is unbiased and, under standard regularity conditions, its variance takes the form

$$\mathrm{Var}(\hat{f}_T) \approx \frac{\mathrm{Var}_{\pi_\theta}(f)}{T}\,\tau_{\mathrm{int}}(f),$$

(Jones, 2004; Geyer, 1992) where $\mathrm{Var}_{\pi_\theta}(f)$ is the marginal variance of $f(\theta)$ when $\theta \sim \pi_\theta$. Equivalently, the *effective sample size* is $T / \tau_{\mathrm{int}}(f)$ (Gelman et al., 1995), making $\tau_{\mathrm{int}}(f)$ a direct measure of sampling efficiency.

## B. SGLD with Momentum

SGLD with momentum $\kappa$ is defined by the one-step update equations

$$\begin{cases} m_t = \kappa m_{t-1} + G_t(\theta_{t-1}) \\ \theta_t = \theta_{t-1} - \Lambda m_t + \sqrt{2\beta^{-1}\Lambda}\,\xi_{t-1}. \end{cases} \tag{B.1}$$

Combining Equation (B.1) with the approximation given in Equation (7) leads to the proxy algorithm with one-step update

$$\begin{cases} \nu_t = \kappa\nu_{t-1} + G_t(\widehat{\theta}) + \nabla G_t(\widehat{\theta})\left(\psi_{t-1} - \widehat{\theta}\right) \\ \psi_t = \psi_{t-1} - \Lambda\nu_t + \sqrt{2\beta^{-1}\Lambda}\,\xi_{t-1} \end{cases} \tag{B.2}$$

We first present the stationary covariance of SGLD with momentum, which recovers the case without momentum by taking $\kappa = 0$. Proofs of results in this section are in Appendix G.

**Proposition B.1.** *If the iterates are updated according to Equation* (B.2) *and they have a stationary distribution, then the stationary covariance $\Sigma_\psi$ satisfies*

$$(1-\kappa)(\Lambda\widehat{H}\Sigma + \Sigma\widehat{H}\Lambda) + \frac{\kappa}{1-\kappa^2}(\Lambda\widehat{H}\Lambda\widehat{H}\Sigma + \Sigma\widehat{H}\Lambda\widehat{H}\Lambda) = \Lambda\overline{C}_\psi\Lambda + \frac{1+\kappa^2}{1-\kappa^2}\Lambda\widehat{H}\Sigma\widehat{H}\Lambda + (1+\kappa^2)\frac{2\Lambda}{\beta}. \tag{B.3}$$

**Proposition B.2.** *Under the same hypotheses as Proposition B.1, the iterate average $\bar{\psi}_k = \frac{1}{k}\sum_{k'=1}^{k}\psi_{k'}$ has stationary covariance*

$$\Sigma_\psi^{(k)} = \frac{1}{k^2}\left(k\Sigma_\psi + 2\sum_{k'=1}^{k-1}\left(I - \Lambda\widehat{H}\right)^{k'}\Sigma_\psi\right),$$

*where $\Sigma_\psi$ is defined by Equation* (B.3).

In the momentum setting, we obtain 2-Wasserstein error bounds analogous to the non-momentum case. The main difference is that the contraction factor and the corresponding constants now depend on the momentum parameter $\kappa$.

**Theorem B.3.** *latexIf assumptions (A)–(C) hold and $\Lambda = \lambda I_D$ for some $\lambda > 0$, $\kappa \in (0, 1)$, $\lambda \in (0, (1-\kappa)/(4L))$, and $\frac{2\lambda L^2}{1-\kappa} + \frac{2L^2\kappa(1+\lambda)}{(1-\kappa)^2} + \kappa < \mu$, then, letting $\bar{\beta} := \rho(A) < 1$ for the coefficient matrix $A$ defined in Equation* (G.9), $\overline{M^p} := N^{-1}\sum_{n=1}^{N} M_n^p$ *for $p \in \{1, 2\}$, $C_s := \mathbb{E}\|\psi_s - \widehat{\theta}\|^4$, and $\mathcal{P}$ as in Equation* (G.3)*, for all $t = 1, 2, \ldots,$*

$$W_2^2\big(\pi_{\theta,t}, \pi_{\psi,t}\big) \leq \bar{\beta}^{\,t}\,W_2^2\big(\pi_{\theta,0}, \pi_{\psi,0}\big) + \mathcal{P}\sum_{s=1}^{t}\bar{\beta}^{\,t-s}\,C_{s-1}.$$

Finally, we find that, with the momentum proportional to $\lambda$, the Wasserstein error remains of order $\lambda/B$; moreover, the bound recovers the non-momentum result when $\kappa = 0$.

**Corollary B.4.** *Under the same assumptions as Theorem B.3 with $\beta = \infty$, assume the* scaled momentum *regime $\kappa = c_\kappa \lambda$ with $0 < c_\kappa \leq \min\left\{\frac{\mu^2}{32L^2}, \frac{\hat{\mu}}{c_1\hat{L}^3}\right\}$ and $\lambda \leq \min\left\{1, \frac{1}{\hat{L}}, \frac{\mu}{c_2L^2}, \frac{B\hat{\mu}}{c_3L^2}, (\hat{\mu}/c_4)^{1/4}\right\}$. Then there exists a constant $A_\star > 0$ (given at Equation (G.24)) such that*

$$W_2(\pi_\theta, \pi_\psi) \leq A_\star \frac{\lambda}{B}.$$

# C. Application to High-dimensional Problems

**Dense setting.** Let $D$ denote the parameter dimension, let $I \sim \mathrm{Unif}(\{1, \ldots, N\})$ be independent, and define $g_I := \nabla_\theta \ell(x_I, y_I, \hat{\theta}), \tau_2^2 := \mathbb{E}\|g_I\|^2$, and $\tau_4^4 := \mathbb{E}\|g_I\|^4$. Assume there exist constants $c_2, c_4 < \infty$ independent of $D$ such that $\tau_2^2 \leq c_2 D$ and $\tau_4^4 \leq c_4 D^2$. Such bounds hold, for example, under sub-Gaussian designs with uniformly bounded GLM weights; see Vershynin, 2018. Let $\xi \sim \mathcal{N}(0, I_D)$ so that $\mathbb{E}\|\xi\|^2 = D$ and $\mathbb{E}\|\xi\|^4 = D(D+2)$. Corollaries 4.6 and B.4 yield

$$W_2(\pi_\theta, \pi_\psi) \leq A_{\mathrm{eff}}\left(\frac{\lambda}{B} + \frac{1}{\beta}\right),$$

where $A_{\mathrm{eff}}$ is the explicit constant in the corresponding corollary (e.g., $A_{\mathrm{eff}} = \sqrt{A}$ when the corollary is stated as $W_2^2 \leq A(\lambda/B + 1/\beta)^2$; see Equations (F.12) and (G.24) for the explicit definitions. If the curvature constants entering these corollaries (e.g., $\mu, L, \hat{\mu}, \hat{L}$) are bounded above and below by constants independent of $d$, then inserting the bounds on $\tau_2, \tau_4$, and the Gaussian moments above into Equations (F.12) and (G.24) shows that there exists $C > 0$ independent of $D$ such that $A_{\mathrm{eff}} \leq CD$, and hence

$$W_2(\pi_\theta, \pi_\psi) \leq C D\left(\frac{\lambda}{B} + \frac{1}{\beta}\right).$$

In particular, if $B \geq cD$ then $W_2(\pi_\theta, \pi_\psi) \leq C'(\lambda + D/\beta)$, and thus $W_2(\pi_\theta, \pi_\psi) \leq C''\lambda$ uniformly in $D$ when $\beta = \infty$ or when $\beta \geq c'''D/\lambda$. If instead $\beta = N$ and $N/D \in [\gamma_{\min}, \gamma_{\max}]$ for fixed $0 < \gamma_{\min} \leq \gamma_{\max} < \infty$, then $d/\beta = D/N \in [1/\gamma_{\max}, 1/\gamma_{\min}]$ and the bound need not vanish as $D \to \infty$.

**Sparse setting.** Alternatively, we can consider the sparse regime. Let $S \subseteq [D]$ with $|S| = s \ll D$ and let $P_S$ be the coordinate projector. Assume both the exact and proxy chains evolve on the affine subspace $\mathcal{A}_S := \hat{\theta} + \mathrm{range}(P_S)$; for example, if $P_{S^c}\theta_0 = P_{S^c}\psi_0 = P_{S^c}\hat{\theta}$ and $P_S$ is applied to every drift and injected-noise term so that $\theta_t, \psi_t \in \mathcal{A}_S$ for all $t$. Define the restricted gradient moments at $\hat{\theta}$ by $g_{I,S} := P_S\nabla_\theta\ell(x_I, y_I, \hat{\theta}), \tau_{2,S}^2 := \mathbb{E}\|g_{I,S}\|^2$ and $\tau_{4,S}^4 := \mathbb{E}\|g_{I,S}\|^4$. Assume the curvature constants, when restricted to $\mathcal{A}_S$, are bounded above and below by constants independent of $D$ and $s$, and there exist constants $c_2, c_4 < \infty$ independent of $D$ and $s$ such that $\tau_{4,S}^2 \leq c_2 s$ and $\tau_{4,S}^4 \leq c_4 s^2$ A sufficient condition is isotropic sub-Gaussian designs on $S$ with uniformly bounded per-sample weights; see (Vershynin, 2018). If the injected noise is also projected, then for $\xi \sim \mathcal{N}(0, I_D)$ it holds that $\mathbb{E}\|P_S\xi\|^2 = s$ and $\mathbb{E}\|P_S\xi\|^4 = s(s+2)$. Then the same inspection of the constants in Corollary 4.6 and Corollary B.4 yields a constant $C > 0$ independent of $D$ and $s$ such that

$$W_2(\pi_\theta, \pi_\psi) \leq Cs\left(\frac{\lambda}{B} + \frac{1}{\beta}\right).$$

In particular, if $B \geq cs$ then $W_2(\pi_\theta, \pi_\psi) \leq C'(\lambda + s/\beta)$, hence $W_2(\pi_\theta, \pi_\psi) \leq C''\lambda$ for $\beta = \infty$ and also for $\beta \geq c''' s/\lambda$. If the injected noise is not rank-$s$, then the diffusion moments scale with $D$ (since $\mathbb{E}\|\xi\|^2 = D$ and $\mathbb{E}\|\xi\|^4 = D(D+2)$), so the temperature-dependent contribution generally scales with $D/\beta$ rather than $s/\beta$.

# D. Additional Experiment Details

## D.1. Empirical Validation of the Wasserstein Bound

We empirically validate the Wasserstein error bound in Corollary 4.6 using Poisson regression in both well-specified and misspecified settings. We generate covariates $x_i \in \mathbb{R}^D$ and responses according to two synthetic data-generating mechanisms. In the well-specified setting, the responses are generated from

$$y_i \sim \mathrm{Poisson}\left(\exp\{x_i^\top \theta_\star\}\right),$$

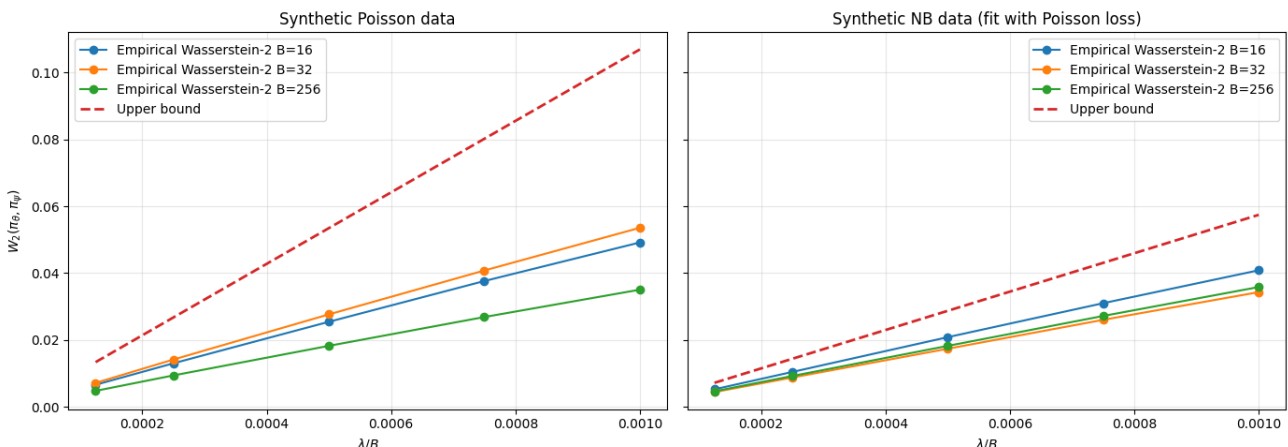

*Figure D.1.* Empirical validation of the Wasserstein error bound in Corollary 4.6. **Left:** well-specified Poisson data fitted with a Poisson model. **Right:** misspecified negative binomial data fitted with a Poisson model.

*Table D.1.* Linear regression: comparison of $\Lambda$ computation and MCMC running time. Entries are mean wall-clock time (seconds) averaged over independent runs.

| $D$ | $\Lambda$ computation time | MCMC time | $\Lambda$ computation time / MCMC |
|---|---|---|---|
| 5 | $9.2 \times 10^{-5}$ | 1.49 | $6.2 \times 10^{-5}$ |
| 10 | $8.4 \times 10^{-4}$ | 1.67 | $5.0 \times 10^{-4}$ |
| 20 | $6.5 \times 10^{-4}$ | 1.83 | $3.6 \times 10^{-4}$ |
| 50 | $8.0 \times 10^{-4}$ | 3.03 | $2.6 \times 10^{-4}$ |

and the fitted model is also Poisson regression. In the misspecified setting, the responses are generated from a negative binomial model,

$$y_i \sim \text{NegBin}(r_i, p_i), \qquad p_i = \frac{r_i}{r_i + \exp\{x_i^\top \theta_\star\}},$$

but are still fitted using a Poisson model. We use sample size $N = 2{,}000$ and dimension $D = 50$.

To instantiate the theoretical upper bound, we use plug-in estimates evaluated at $\widehat{\theta}$. Specifically, we estimate the smoothness constant $L$ and strong convexity constant $\mu$ using the largest and smallest eigenvalues of the empirical Hessian, respectively. The higher-order quantities $\overline{M}$, $\overline{M^2}$, and $\tau_4$ are estimated empirically from the sample gradients. We then evaluate $W_2(\pi_\theta, \pi_\psi)$ across different batch sizes $B$ and different values of the ratio $\lambda/B$.

As shown in Figure D.1, the empirical Wasserstein distance exhibits a clear approximately linear scaling in $\lambda/B$ across all batch sizes. Moreover, in both the well-specified Poisson setting and the misspecified negative binomial setting, the empirical curves remain below the theoretical upper bound. This confirms that the bound captures the correct dependence on $\lambda/B$, although it is conservative in magnitude.

### D.2. Computational Cost of Determining $\Lambda$

In this subsection, we benchmark the wall-clock cost of computing the preconditioner $\Lambda$—by solving the matrix equation induced by Equations (11) and (12)—against the cost of running the resulting MCMC chain. Section D.2 show that $\Lambda$ can be computed extremely quickly relative to sampling: across all tested dimensions, $\Lambda$ construction takes at most a few milliseconds and is typically $10^{-5}$–$10^{-3}$ of the MCMC runtime. In the following exploratory study, we fix $N = 1000$ and $B = 64$, and run MCMC for 10 epochs.

*Table D.2.* Poisson regression: comparison of $\Lambda$ computation and MCMC time. Entries are mean wall-clock time (seconds) averaged over independent runs.

| $D$ | $\Lambda$ computation time | MCMC time | $\Lambda$ computation time / MCMC |
|---|---|---|---|
| 5  | $9.5 \times 10^{-5}$ | 1.71 | $5.6 \times 10^{-5}$ |
| 10 | $4.0 \times 10^{-3}$ | 1.74 | $2.3 \times 10^{-3}$ |
| 20 | $8.5 \times 10^{-4}$ | 1.79 | $4.8 \times 10^{-4}$ |
| 50 | $7.8 \times 10^{-4}$ | 1.81 | $4.3 \times 10^{-4}$ |

### D.3. Details of the $\beta$-divergence.

Under the definition of $\beta$-divergence $\ell^{(\beta)}(y, f(\cdot; \theta)) = -\frac{1}{\beta-1} f(y; \theta)^{\beta-1} + \frac{1}{\beta} \int f(z; \theta)^{\beta} dz$, the loss $\mathcal{L}$ can be rewriten as follows

$$\mathcal{L}(\theta) = \frac{1}{N} \sum_{n=1}^{N} \ell^{(\beta)}(y_n, f(\cdot; \theta)) + \frac{1}{N} \mathcal{R}(\theta)$$

$$= -\frac{1}{N} \sum_{n=1}^{N} \frac{1}{\beta-1} f(y_n; \theta)^{\beta-1} + \frac{1}{\beta} \int f(z; \theta)^{\beta} dz + \frac{1}{N} \mathcal{R}(\theta)$$

$$= \frac{1}{N} \sum_{n=1}^{N} \tilde{\ell}_n^{(\beta)} + \frac{1}{N} \left( \Omega^{(\beta)}(\theta) + \mathcal{R}(\theta) \right),$$

where $\tilde{\ell}_n^{(\beta)}(\theta) = -\frac{1}{\beta-1} f(y_n; \theta)^{\beta-1}$ and $\Omega^{(\beta)}(\theta) = \frac{N}{\beta} \int f(z; \theta)^{\beta} dz$.

Then the loss $\mathcal{L}$ can be rewritten as

$$\mathcal{L}(\theta) = \frac{1}{N} \sum_{n=1}^{N} \ell_n^{(\beta)}(\theta),$$

where $\ell_n^{(\beta)}(\theta) = \tilde{\ell}_n^{(\beta)} + \frac{1}{\beta} \int f(z; \theta)^{\beta} dz + \frac{1}{N} \mathcal{R}(\theta)$.

Similarly, we can compute $\widehat{\mathcal{J}}^{(\beta)} = \frac{1}{N} \sum_{n=1}^{N} \nabla^2 \ell_n^{(\beta)}(\hat{\theta})$ and $\widehat{\mathcal{I}}^{(\beta)} = \frac{1}{N} \sum_{n=1}^{N} \nabla \ell_n^{(\beta)}(\hat{\theta})(\nabla \ell_n^{(\beta)}(\hat{\theta}))^{\top}$ to use our Algorithm 1 under $\beta$-divergence loss.

### D.4. Full Experiment Results

Due to space constraints, we are unable to report parameter errors and detailed confidence intervals in the main text. This subsection therefore presents the complete experimental results for all settings. To aid interpretation of the reported relative covariance errors, we additionally compare the marginal variances of the estimated covariance $\hat{\mathcal{S}}$ with those of the target covariance $\mathcal{S}_{\star}$.

Figure D.2 shows that for both datasets, using our results leads to the desired marginal variances when using either a small or large batch size. The continuous-time tuning performs well when the batch size is small, since a small batch size requires using a small learning rate. However, the variances are too large in the large batch size case. The large-sample+well-specified tuning, on the other hand, leads to excessive variance for both small and large batch size regimes since the assumption that the model is well-specified is violated. Figure D.3 shows that theories based on the heuristic SGD noise $\overline{C} = \frac{1}{B} H$ lead to an excessively large stationary covariance for the simulated data but a too smaller covariance for the German credit data. The continuous-time tuning leads to too large covariance for the large batch size in both cases. In our theory, on the other hand, is accurate in all scenarios.

## E. Application: Stationary Covariance for a Fixed Learning Rate

In this section, we discuss how our theory can be used to justify the stationary covariance structure at a fixed learning rate.

*Table D.3.* Full results for linear regression experiments with simulated data. Calibration error is the Kolmogorov–Smirnov distance to $\mathrm{Unif}(0, 1)$; lower is better. Covariance error is $(\|\mathcal{S}_\star - \hat{\mathcal{S}}\|_F)/\|\mathcal{S}_\star\|_F$; lower is better. Within each metric row and loss block, for a fixed batch size $B$, bold indicates methods whose 95% confidence intervals overlap with the confidence interval of the method with the lowest mean error.

| | | Log loss | | | | $\beta$-loss ($\beta = 1.5$) | | | |
|---|---|---|---|---|---|---|---|---|---|
| $B$ | Posterior | CT | LR+WS | DQ+exact | NUTS | Sandwich Gauss | CT | LR+WS | DQ+exact |
| **Calibration error** | | | | | | | | | |
| 16 | 0.418 | **0.171** [0.141, 0.206] | 0.529 [0.484, 0.580] | **0.169** [0.129, 0.214] | 0.195 | 0.156 | **0.201** [0.162, 0.241] | **0.178** [0.148, 0.207] | **0.172** [0.133, 0.210] |
| $\lfloor 0.1 \times N \rfloor$ | 0.418 | **0.179** [0.154, 0.207] | 0.517 [0.469, 0.581] | **0.174** [0.139, 0.216] | 0.195 | 0.156 | **0.196** [0.157, 0.228] | **0.177** [0.142, 0.216] | **0.190** [0.151, 0.231] |
| **Covariance error** | | | | | | | | | |
| 16 | 0.943 | **0.672** [0.629, 0.715] | 0.995 [0.995, 0.996] | **0.664** [0.616, 0.710] | 0.795 | 0.000 | **0.640** [0.595, 0.685] | 1.115 [1.023, 1.204] | **0.695** [0.651, 0.734] |
| $\lfloor 0.1 \times N \rfloor$ | 0.943 | 0.975 [0.905, 1.045] | 0.996 [0.995, 0.996] | **0.672** [0.625, 0.726] | 0.799 | 0.000 | 1.006 [0.948, 1.068] | 1.322 [1.211, 1.438] | **0.748** [0.713, 0.784] |

*Table D.4.* Full results for linear regression experiments with Boston housing data. See Table D.3 caption for further explanation.

| | | Log loss | | | | $\beta$-loss ($\beta = 1.5$) | | | |
|---|---|---|---|---|---|---|---|---|---|
| $B$ | Posterior | CT | LR+WS | DQ+exact | NUTS | Sandwich Gauss | CT | LR+WS | DQ+exact |
| **Covariance error** | | | | | | | | | |
| 16 | 0.358 | **0.247** [0.194, 0.310] | $9.23 \times 10^8$ [$3.62 \times 10^4, 6.17 \times 10^9$] | **0.337** [0.262, 0.405] | 2.528 | 0 | **2.054** [1.723, 2.328] | $\infty$ | **2.782** [0.965, 9.328] |
| $\lfloor 0.1 \times N \rfloor$ | 0.358 | 0.589 [0.443, 0.804] | $1.40 \times 10^7$ [$4.85 \times 10^3, 9.01 \times 10^7$] | **0.352** [0.274, 0.441] | 2.528 | 0 | 3.126 [2.313, 5.338] | $\infty$ | **1.398** [0.844, 2.132] |

*Table D.5.* Results for Poisson regression experiments. See Table D.3 caption for further explanation.

| | | Simulated | | Credit |
|---|---|---|---|---|
| $B$ | method | calib. err. | cov. err. | cov. err. |
| 16 | CT | **0.069** [0.062,0.075] | **0.207** [0.199,0.215] | **0.132** [0.112,0.168] |
| | DQ+const | 0.646 [0.639,1.344] | 0.672 [0.664,0.678] | 0.982 [0.975,0.987] |
| | DQ+exact | **0.074** [0.068,0.080] | **0.208** [0.201,0.217] | **0.157** [0.132,0.193] |
| $\lfloor 0.1 \times N \rfloor$ | CT | 0.089 [0.078,0.100] | 0.230 [0.218,0.245] | 0.191 [0.155,0.240] |
| | DQ+const | 1.376 [1.370,1.382] | 0.991 [0.990,0.992] | 0.997 [0.996,0.999] |
| | DQ+exact | **0.075** [0.066,0.083] | **0.211** [0.203,0.220] | **0.154** [0.138,0.181] |

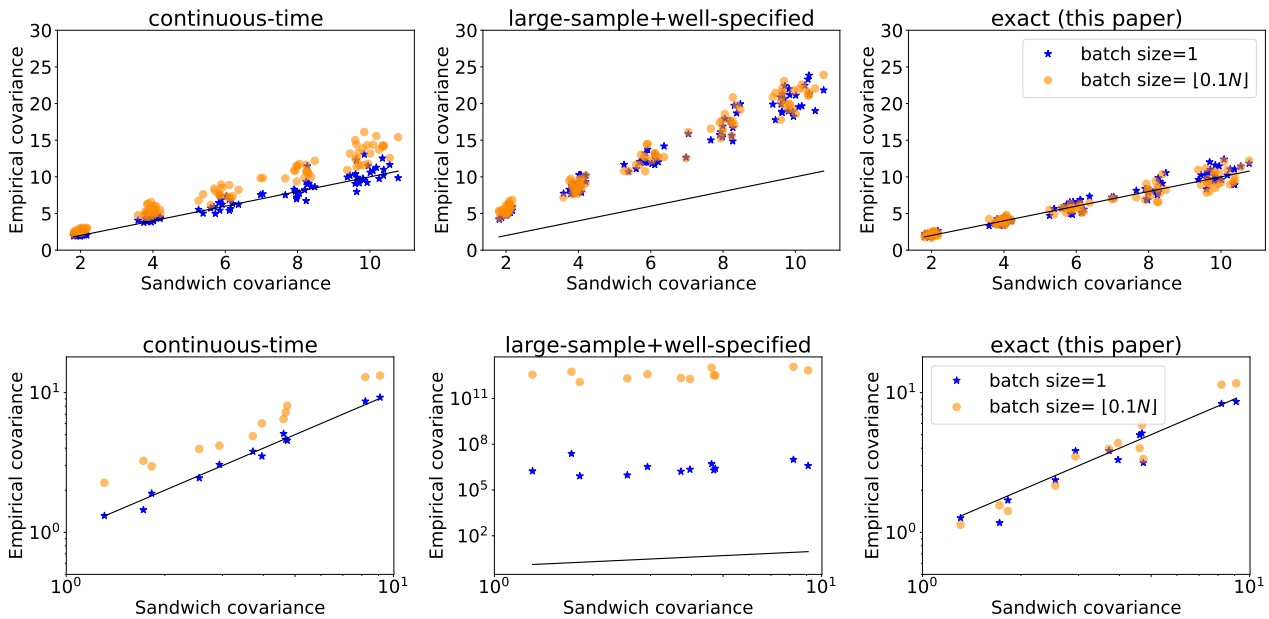

*Figure D.2.* Comparison of step size tuning guidance for linear regression with **(top)** simulated misspecified data with heteroskedastic noise and **(bottom)** the classic Boston housing dataset.

### E.1. Linear Regression

As an illustration of the usefulness of, and new insights provided by Theorem 4.3, we first focus on the special case of linear regression without regularization (i.e., where $\mathcal{R} \equiv 0$). Since in the case of linear regression the proxy algorithm is identical to the exact algorithm, we will give all our results in terms of the original process $(\theta_t)_{t \geq 0}$. In linear regression we can specialize Equation (12) to obtain

$$\overline{C}_\theta = \tfrac{1}{B}(N^{-1}\textstyle\sum_{n=1}^N x_n x_n^\top \Sigma_\theta x_n x_n^\top - \widehat{H}\Sigma_\theta \widehat{H}) + \tfrac{1}{BN}\textstyle\sum_{n=1}^N r_n^2 x_n x_n^\top, \tag{E.1}$$

where $r_n = y_n - \widehat{\theta}^\top x_n$ is the residual and $\widehat{H} = N^{-1}\sum_{n=1}^N x_n x_n^\top$.

**Relation to large-sample approximation of Ziyin et al. (2022).** We can recover the approximation given in Equation (6) by making the same simplifying assumptions and approximations (see Section E.3.1 for details). First, if $x_n \sim \mathcal{N}(0, A)$ and $N$ is large, then, using the properties of the Gaussian, the first term on the righthand side of Equation (E.1) is well-approximated by $2A\Sigma_\theta A + \mathrm{Tr}[A\Sigma_\psi]A$ and $\widehat{H} \approx A$. Hence, the first two terms together are approximately equal to $A\Sigma_\theta A + \mathrm{Tr}[A\Sigma_\psi]A$. However, in many scenarios the covariates may not be normally distribution (e.g., they may be binary or have heavier tails) and $N$ may not be large relative to the parameter/covariate dimension $D$. To simplify the final term in Equation (E.1), we must also assume the model is well-specified, which implies that $x_n$ and $r_n$ are independent and $r_n \sim \mathcal{N}(0, \sigma^2)$. Hence, when $N$ is large, the final term is approximately $\sigma^2 A$. However, when the model is misspecified, the term $N^{-1}\sum_{i=1}^N r_i^2 x_i x_i^\top$ can capture additional variability due to, for example, a poor model fit, heteroskedastic errors, and/or heavy-tailed errors. We illustrate this latter point next.

**Numerical illustrations.** To validate our theory, we compare the predicted stationary covariance structure obtained from combining Proposition 4.2 and Equation (E.1) with predictions based on (1) the continuous-time theory and (2) the discrete-time theory that assumes large $N$ and a well-specified model. We focus on the effect of varying the (scalar) learning rate.

***Simulated misspecified data.*** First, we consider a misspecified simulated dataset with heteroskedastic error generated according to the model

$$y_n \sim \mathcal{N}(x_n^\top \theta_\star, 1 + \|x_i\|_2^2), \tag{E.2}$$

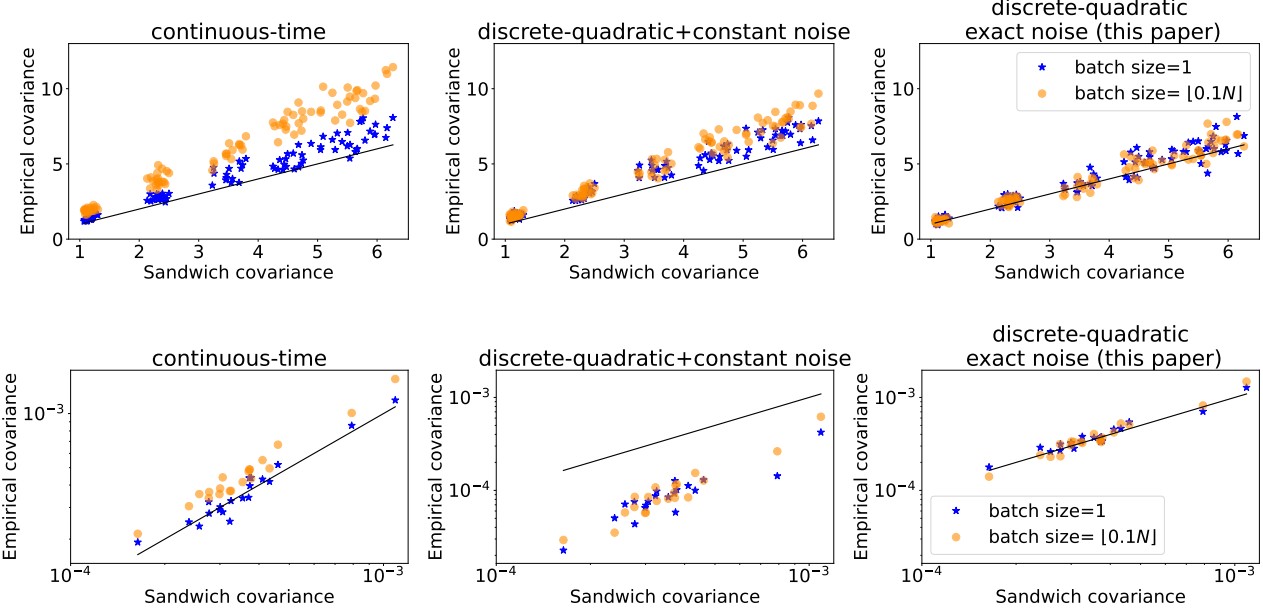

*Figure D.3.* Comparison of step size tuning guidance for Poisson regression with **(top)** simulated well-specified data and **(bottom)** the German credit data.

where $\theta_\star \sim \mathcal{N}(0, I_D)$ is fixed and $x_n \overset{\text{iid}}{\sim} \mathcal{N}(0, I_D)$. We take $D = 20$ and $N = 2{,}000$. Figure E.1(left) illustrates the predicted covariance for the parameters $(\theta_1, \theta_2)^\top$. The results show that our theory delivers the most accurate covariance predictions across all learning rate levels. In contrast, the continuous-time theory underestimates the parameter variances, while the discrete-time approximation that assume $N$ is large and the model is correct overestimates them.

***Boston housing data.*** Next, we reconsider the real-world Boston housing data Similar to the results on simulated data, Figure E.1(right) demonstrates that our theory can accurately predict the covariance. The alternative approximations consistently underestimate it.

### E.2. Poisson Regression

Similar to the linear regression experiments, we compare the stationary covariance predicted by our theory with those derived from continuous-time theory and the discrete-time quadratic loss proxy with constant noise (that is, using $\overline{C}_\psi \approx \frac{1}{B}\widehat{H}$ in Equation (11)). However, unlike in linear regression, the proxy algorithm is no longer exact, and so we must rely on our error analysis to justify its use.

| Learning Rate $\lambda$ | continuous-time | discrete-quadratic+constant noise | discrete-quadratic+exact noise |
|---|---|---|---|
| $\|\Sigma_\psi - \Sigma_\theta\|_F$ **for Poisson regression on well-specified simulated dataset** | | | |
| 0.1 | 0.237 | 0.302 | **0.030** |
| 0.3 | 0.479 | 0.631 | **0.096** |
| 0.5 | 0.545 | 0.651 | **0.202** |
| $\|\Sigma_\psi - \Sigma_\theta\|_F$ **for Poisson regression on misspecified German credit dataset** | | | |
| 0.1 | 0.0367 | 0.037 | **0.004** |
| 0.3 | 0.098 | 0.099 | **0.025** |
| 0.5 | 0.124 | 0.126 | **0.041** |

*Table E.1.* Comparison of difference between estimated stationary covariance structure $\Sigma_\psi$ and the ground truth using Frobenius norm for Poisson regression.

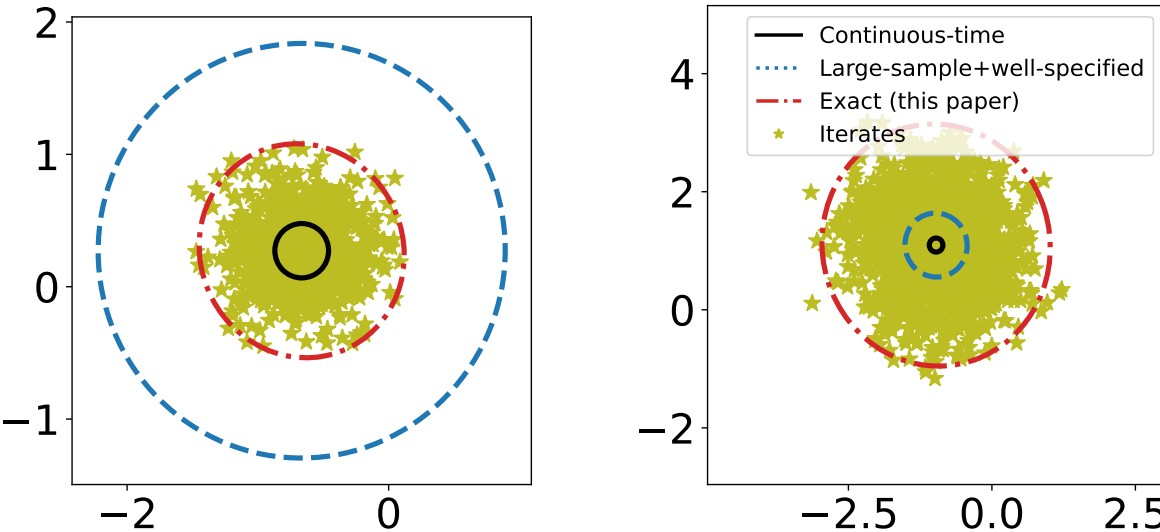

*Figure E.1.* Comparison of estimated stationary covariance structure for linear regression at $3\sigma$ confidence region on **(left)** simulated misspecified data with heteroskedastic noise and **(right)** the classic Boston housing dataset with $\lambda = 0.1$ and $B = 32$. Our theory provides more accurate stationary covariance predictions in both cases.

Figure E.2(right) shows that our theory provides an accurate estimate of the stationary covariance while alternatives provide severe underestimates.

For both simulated and real-world dataset, our approximation demonstrates an improvement in accuracy with errors that are 3–10 times smaller than the baseline approaches as shown in Table E.1.

### E.3. Optimal weight decay and batch size

A direct application of accurate stationary covariance prediction is to estimate the test loss. To simplify our analysis, we will focus on linear regression. The test loss depends on the stationary covariance by $\mathcal{L}_{\text{test}} = \frac{1}{M} \sum_{m=1}^{M} (y_m - \widehat{\theta}^\top x_m)^2 + \frac{1}{M} \sum_{m=1}^{M} x_m^\top \Sigma_\theta x_m$ (see Section E.3.2), where $\{(x_m, y_m)\}_{m=1}^{M}$ is the test dataset. As illustrated in Figure E.3, our theory offers the most accurate test loss estimation across different decay weights and batch sizes.

#### E.3.1. MORE DISCUSSION ABOUT SECTION E.1

Recall that in linear regression we can specialize Equation (12) to obtain

$$\overline{C}_\theta = \frac{1}{B}(N^{-1}\sum_{n=1}^{N} x_n x_n^\top \Sigma_\theta x_n x_n^\top - \widehat{H}\Sigma_\theta \widehat{H}) + \frac{1}{BN}\sum_{n=1}^{N} r_n^2 x_n x_n^\top,$$

where $r_n = y_n - \widehat{\theta}^\top x_n$ is the residual and $\widehat{H} = N^{-1}\sum_{n=1}^{N} x_n x_n^\top$.

Now suppose that the data $\{(x_n, y_n)\}_{n=1}^{N}$ are generated from a linear model, there exists a $\theta_\star \in \mathbb{R}^D$ such that $y_n = x_n^\top \theta_\star + \epsilon_n$, where $\epsilon_n \overset{\text{iid}}{\sim} \mathcal{N}(0, \sigma^2)$, for $i = 1, 2, ..., N$. Now we will focus on the MSE loss defined as

$$\mathcal{L}(\theta) = \frac{1}{N}\sum_{n=1}^{N} \ell(x_n, y_n, \theta) = \frac{1}{2N\sigma^2}\sum_{n=1}^{N}(y_n - x_n^\top \theta)^2.$$

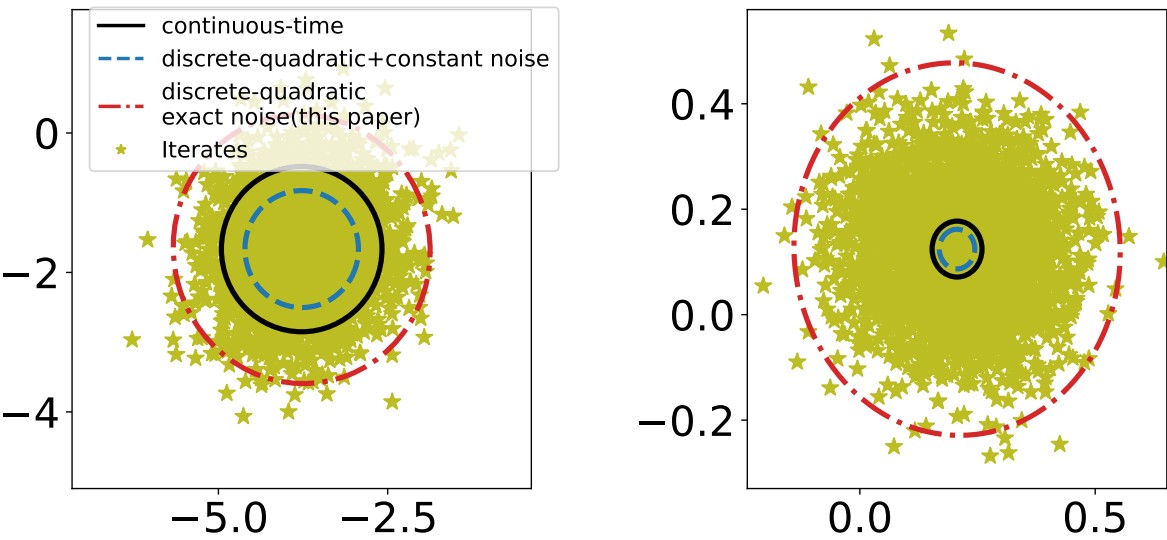

*Figure E.2.* Comparison of estimated stationary covariance structure for Poisson regression at $3\sigma$ confidence region with **(left)** simulated well-specified data and **(right)** the German credit data by setting batch size $\lambda = 0.1$, and $B = 32$.

Note that $\widehat{\theta} \sim \mathcal{N}\left(\theta_\star, \sigma^2 \left(\mathbf{X}^\top \mathbf{X}\right)^{-1}\right)$, where $\mathbf{X} \in \mathbb{R}^{N \times D}$, then we have

$$
\frac{1}{N}\mathbb{E}\left[\sum_{i=1}^{N} r_i^2 x_i x_i^\top\right] = \frac{1}{N}\sum_{i=1}^{N}\mathbb{E}\left[\left(y_i - x_i^\top \widehat{\theta}\right)^2\right] x_i x_i^\top
$$

$$
= \frac{1}{N}\sum_{i=1}^{N}\left(\mathbb{E}\left[(y_i - \mathbb{E}[y_i])^2\right] + \mathbb{E}\left[\left(x_i^\top \widehat{\theta} - \mathbb{E}[y_i]\right)^2\right]\right) x_i x_i^\top
$$

$$
= \frac{1}{N}\sum_{i=1}^{N}\sigma^2\left(I + \left(\mathbf{X}^\top \mathbf{X}\right)^{-1}\right) x_i x_i^\top
$$

$$
= \sigma^2\left(A + \frac{1}{N}I\right).
$$

Then we have

$$
\lim_{N \, to\infty}\mathbb{E}\left[\sum_{i=1}^{N} r_i^2 x_i x_i^\top\right] = \sigma^2 A.
$$

Under the assumptions of $x_n \sim \mathcal{N}(0, A)$ and $N$ being large, we have

$$
\lim_{N \to \infty}\frac{1}{N}\sum_{n=1}^{N} x_n x_n^\top \Sigma_\theta x_n x_n^\top - \widehat{H}\Sigma_\theta \widehat{H} = A\Sigma_\theta A + \mathrm{Tr}[A\Sigma_\psi]A.
$$

Then, we will get exactly the same result of Lemma 1 in Ziyin et al. (2022).

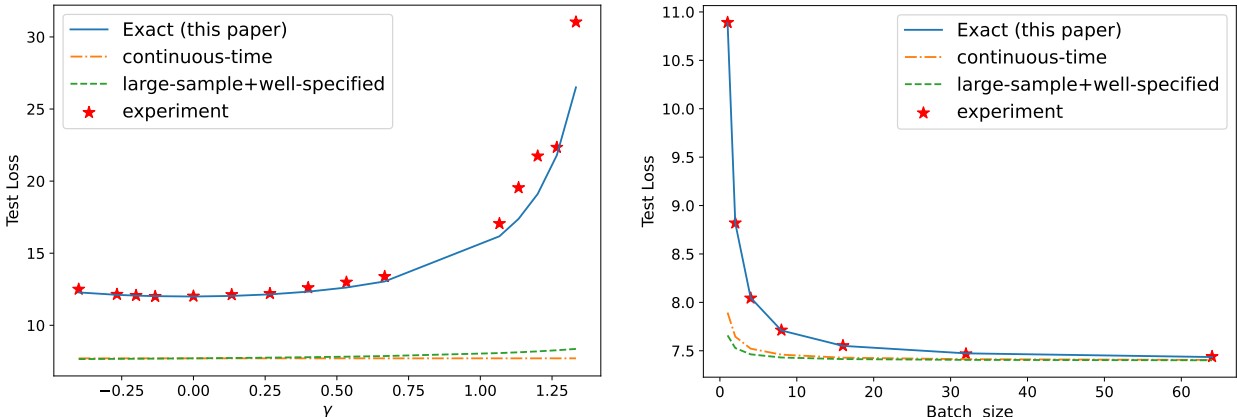

*Figure E.3.* Comparison of estimated test loss ridge regression ($\Gamma = \gamma I_D$) on simulated misspecified data with heteroskedastic noise considered in Equation (E.2). **(left)** We set $\lambda = 0.1$, $B = 32$. **(right)** We set $\lambda = 0.1$, $\gamma = 0$.

### E.3.2. TEST LOSS OF LINEAR REGRESSION

The test loss in linear regression can be decomposed as follows:

$$
\begin{aligned}
\mathcal{L}_{\text{test}} &= \mathbb{E}\left[\frac{1}{M}\sum_{m=1}^{M}\left(y_m - \theta_t^\top x_m\right)^2\right] \\
&= \mathbb{E}\left[\frac{1}{M}\sum_{m=1}^{M}\left(y_m - \hat{\theta}^\top x_m + (\hat{\theta} - \theta_t)^\top x_m\right)^2\right] \\
&= \frac{1}{M}\sum_{m=1}^{M}\left(y_m - \hat{\theta}^\top x_m\right)^2 + \frac{1}{M}\sum_{m=1}^{M} x_m^\top \mathbb{E}\left[(\theta_t - \hat{\theta})(\theta_t - \hat{\theta})^\top\right] x_m \\
&= \frac{1}{M}\sum_{m=1}^{M}\left(y_m - \hat{\theta}^\top x_m\right)^2 + \frac{1}{M}\sum_{m=1}^{M} x_m^\top \Sigma_\theta x_m,
\end{aligned}
$$

where $\Sigma_\theta = \mathbb{E}\left[(\theta_t - \hat{\theta})(\theta_t - \hat{\theta})^\top\right]$.

## F. Proofs from Main Text

**Lemma F.1.** *Assume the parameters $\psi$ are updated based on discrete-time proxy algorithm Equation (7), $\mathcal{R}(\psi) = \frac{1}{2}\psi^\top \Gamma \psi$, and the stationary distribution of $\psi$ exists, then the stationary mean $\mu_\psi$ satisfies $\mu_\psi = \hat{\theta}$. If the parameters $\psi$ are updated based on discrete-time proxy algorithm Equation (7), and the stationary distribution of $\psi$ exists, then the stationary mean $\mu_\psi$ satisfies $\mu_\psi = \hat{\theta}$.*

*Proof.* Now we assume that $\psi_0$ are sampled from the stationary distribution. Then by taking expectation we have

$$
\begin{aligned}
\mu_\psi = \mathbb{E}\left[\psi_t\right] &= \mathbb{E}\left[\psi_{t-1} - \Lambda\left\{G_t(\hat{\theta}) + \nabla G_t(\hat{\theta})\left(\psi_{t-1} - \hat{\theta}\right)\right\} + \sqrt{2\beta^{-1}\Lambda}\,\xi_{t-1}\right] \\
&= \mu_\psi - \Lambda\mathbb{E}(G_t(\hat{\theta})) - \Lambda\mathbb{E}\left(\nabla G_t(\hat{\theta})(\psi_{t-1} - \hat{\theta})\right)
\end{aligned}
$$

Then we have

$$
\Lambda\mathbb{E}(G_t(\hat{\theta})) + \Lambda(\mathcal{J} + \frac{1}{N}\Gamma)(\mu_\psi - \hat{\theta}) = 0
$$

Since $\widehat{\theta}$ satisfies $\nabla\mathcal{L}(\widehat{\theta}) = 0$, we have

$$\mathbb{E}(G_t(\hat{\theta})) = \frac{\Gamma}{N}\widehat{\theta} + \frac{1}{N}\sum_{n=1}^{N}\nabla\ell\left(x_n, y_n, \widehat{\theta}\right) = 0. \tag{F.1}$$

Therefore, combining the previous two displayed equations and if , we conclude that $\mu_\psi = \widehat{\theta}$.

$\square$

### F.1. Proof of Proposition 4.2

*Proof.* The proxy algorithm leads to the discrete-time update

$$\psi_t = \psi_{t-1} - \Lambda\left[G_t(\widehat{\theta}) + \nabla G_t(\widehat{\theta})(\psi_{t-1} - \widehat{\theta})\right] + \sqrt{2\beta^{-1}\Lambda}\,\xi_{t-1},$$

where $\xi_{t-1} \sim \mathcal{N}(0, I)$. Let

$$\eta_{t-1} = \left[G_t(\widehat{\theta}) + \nabla G_t(\widehat{\theta})(\psi_{t-1} - \widehat{\theta})\right] - \mathbb{E}\left[G_t(\widehat{\theta}) + \nabla G_t(\widehat{\theta})(\psi_{t-1} - \widehat{\theta})\right],$$

it follows from Equation (F.1) that $\mathrm{Cov}(\eta_{t-1}) = \overline{C}_\psi$. Noting that $\mathcal{J} = \frac{1}{N}\sum_{n=1}^{N}\mathcal{J}_n$, we have

$$\widehat{H} = \mathbb{E}\left[\nabla G_t(\widehat{\theta})\right] = \mathcal{J} + \frac{1}{N}\Gamma.$$

Equation (8) can also be rewritten as

$$\begin{aligned}
\psi_t - \widehat{\theta} &= \psi_{t-1} - \widehat{\theta} - \Lambda\left[G_t(\widehat{\theta}) + \nabla G_t(\widehat{\theta})(\psi_{t-1} - \widehat{\theta})\right] + \sqrt{2\beta^{-1}\Lambda}\,\xi_{t-1} \\
&= \psi_{t-1} - \widehat{\theta} - \Lambda\mathbb{E}\left[G_t(\widehat{\theta}) + \nabla G_t(\widehat{\theta})(\psi_{t-1} - \widehat{\theta})\right] - \Lambda\eta_{t-1} + \sqrt{2\beta^{-1}\Lambda}\,\xi_{t-1} \\
&= \psi_{t-1} - \widehat{\theta} - \Lambda\left(\frac{1}{N}\sum_{n=1}^{N}\mathcal{J}_n - \frac{1}{N}\Gamma\right)(\psi_{t-1} - \widehat{\theta}) - \Lambda\eta_{t-1} + \sqrt{2\beta^{-1}\Lambda}\,\xi_{t-1} \\
&= \left(I - \Lambda\mathcal{J} - \frac{1}{N}\Lambda\Gamma\right)\left(\psi_{t-1} - \widehat{\theta}\right) - \Lambda\eta_{t-1} + \sqrt{2\beta^{-1}\Lambda}\,\xi_{t-1} \\
&= \left(I - \Lambda\widehat{H}\right)\left(\psi_{t-1} - \widehat{\theta}\right) - \Lambda\eta_{t-1} + \sqrt{2\beta^{-1}\Lambda}\,\xi_{t-1}.
\end{aligned}$$

Note that

$$\begin{aligned}
\Sigma_\psi &= \mathbb{E}\left[\left(\psi_t - \widehat{\theta}\right)\left(\psi_t - \widehat{\theta}\right)^\top\right] \\
&= \left(I - \Lambda\widehat{H}\right)\Sigma_\psi\left(I - \Lambda\widehat{H}\right)^\top + \Lambda\overline{C}_\psi\Lambda + \frac{2\Lambda}{\beta}.
\end{aligned}$$

Then, after some algebra, we have

$$\Lambda\widehat{H}\Sigma_\psi + \Sigma_\psi\widehat{H}\Lambda = \Lambda\left(\overline{C}_\psi + \widehat{H}\Sigma_\psi\widehat{H}\right)\Lambda + \frac{2\Lambda}{\beta}.$$

$\square$

### F.2. Proof of Theorem 4.3

The covariance of the gradient noise for parameter $\psi$ is given by

$$C(\psi) = \begin{cases} \frac{1}{B}\left[\frac{1}{N}\sum_{n=1}^{N}\nabla\tilde{\ell}_n(\psi)\nabla\tilde{\ell}_n(\psi)^\top - \nabla\tilde{\mathcal{L}}(\psi)\nabla\tilde{\mathcal{L}}(\psi)^\top\right] & \text{if with replacement} \\ \frac{N-B}{B(N-1)}\left[\frac{1}{N}\sum_{n=1}^{N}\nabla\tilde{\ell}_n(\psi)\nabla\tilde{\ell}_n(\psi)^\top - \nabla\tilde{\mathcal{L}}(\psi)\nabla\tilde{\mathcal{L}}(\psi)^\top\right] & \text{if without replacement.} \end{cases}$$

We focus on sampling with replacement since our results can be easily extended to the sampling without replacement case by substituting each $C(\psi)$ term with $\frac{N-B}{N-1}C(\psi)$.

We have

$$
\begin{aligned}
C(\psi_{t-1}) &= \frac{1}{NB}\sum_{n=1}^{N}\nabla\ell_n\left(\psi_{t-1}\right)\nabla\ell_n\left(\psi_{t-1}\right)^{\top} - \frac{1}{B}\nabla\mathcal{L}\left(\psi_{t-1}\right)\nabla\mathcal{L}\left(\psi_{t-1}\right)^{\top} \\
&= \underbrace{\frac{1}{B}\frac{1}{N}\sum_{n=1}^{N}\left[\nabla\ell\left(x_n,y_n,\widehat{\theta}\right) + \mathcal{J}_n\left(\psi_{t-1}-\widehat{\theta}\right)\right]\left[\nabla\ell\left(x_n,y_n,\widehat{\theta}\right) + \mathcal{J}_n\left(\psi_{t-1}-\widehat{\theta}\right)\right]^{\top}}_{C_3(\psi_{t-1})} \\
&\quad - \underbrace{\frac{1}{B}\left[\frac{1}{N}\sum_{n=1}^{N}\nabla\ell\left(x_n,y_n,\widehat{\theta}\right) + \mathcal{J}_n\left(\psi_{t-1}-\widehat{\theta}\right)\right]\left[\frac{1}{N}\sum_{n=1}^{N}\nabla\ell\left(x_n,y_n,\widehat{\theta}\right) + \mathcal{J}_n\left(\psi_{t-1}-\widehat{\theta}\right)\right]^{\top}}_{C_4(\psi_{t-1})}.
\end{aligned}
$$

Note that

$$
\begin{aligned}
\mathbb{E}\left[C_3(\psi_{t-1})\right] &= \frac{1}{BN}\sum_{n=1}^{N}\left[\nabla\ell\left(x_n,y_n,\widehat{\theta}\right)\right]\left[\nabla\ell\left(x_n,y_n,\widehat{\theta}\right)\right]^{\top} + \frac{1}{BN}\sum_{n=1}^{N}\mathcal{J}_n\mathbb{E}\left[\left(\psi_{t-1}-\widehat{\theta}\right)\left(\psi_{t-1}-\widehat{\theta}\right)^{\top}\right]\mathcal{J}_n^{\top} \\
&= \frac{1}{BN}\sum_{n=1}^{N}\left[\nabla\ell\left(x_n,y_n,\widehat{\theta}\right)\right]\left[\nabla\ell\left(x_n,y_n,\widehat{\theta}\right)\right]^{\top} + \frac{1}{BN}\sum_{n=1}^{N}\mathcal{J}_n\Sigma_\psi\mathcal{J}_n \\
&= \frac{1}{B}\mathcal{I} + \frac{1}{BN}\sum_{n=1}^{N}\mathcal{J}_n\Sigma_\psi\mathcal{J}_n.
\end{aligned}
$$

Also note that, using Equation (F.1),

$$
\begin{aligned}
\mathbb{E}\left[C_4(\psi_{t-1})\right] &= \frac{1}{B}\left\{\left(\frac{1}{N}\sum_{n=1}^{N}\nabla\ell\left(x_n,y_n,\widehat{\theta}\right)\right)\left(\frac{1}{N}\sum_{n=1}^{N}\nabla\ell\left(x_n,y_n,\widehat{\theta}\right)\right)^{\top} + \mathcal{J}\mathbb{E}\left[\left(\psi_{t-1}-\widehat{\theta}\right)\left(\psi_{t-1}-\widehat{\theta}\right)^{\top}\right]\mathcal{J}\right\} \\
&= \frac{1}{B}\left(\frac{1}{N^2}\Gamma\widetilde{\widehat{\theta\theta}}^{\top}\Gamma^{\top} + \mathcal{J}\Sigma_\psi\mathcal{J}\right).
\end{aligned}
$$

Therefore, we have

$$
\begin{aligned}
\overline{C}_\psi &= \mathbb{E}\left[C_3(\psi_{t-1})\right] - \mathbb{E}\left[C_4(\psi_{t-1})\right] \\
&= \frac{1}{B}\left(\mathcal{I} - \frac{1}{N^2}\Gamma\widetilde{\widehat{\theta\theta}}^{\top}\Gamma^{\top} + \frac{1}{N}\sum_{n=1}^{N}\mathcal{J}_n\Sigma_\psi\mathcal{J}_n - \mathcal{J}\Sigma_\psi\mathcal{J}\right).
\end{aligned}
$$

### F.3. Proof of Proposition 4.4

We analyze the mixing behavior under the proxy dynamics Equation (7). For each coordinate projection $f_i(\theta) := \theta_i$, the theoretical lag-$k$ autocorrelation is defined as

$$
\begin{aligned}
\rho_{k,i} &:= \mathrm{Corr}_{\pi_\theta}\left(\theta_{0,i},\theta_{k,i}\right) \\
&= \frac{\mathrm{Cov}_{\pi_\theta}(\theta_{0,i},\theta_{k,i})}{\mathrm{Var}_{\pi_\theta}(\theta_{0,i})} = \frac{\mathbb{E}_{\pi_\theta}\left[(\theta_{0,i}-\widehat{\theta}_i)(\theta_{k,i}-\widehat{\theta}_i)\right]}{(\Sigma_\psi)_{ii}} \\
&= \frac{\left(\mathbb{E}_{\pi_\theta}\left[(\theta_0-\widehat{\theta})(\theta_k-\widehat{\theta})^{\top}\right]\right)_{ii}}{(\Sigma_\psi)_{ii}}.
\end{aligned}
$$

Under the proxy update Equation (8), the iterates satisfy

$$\psi_t - \widehat{\theta} = (I - \Lambda\widehat{H})(\psi_{t-1} - \widehat{\theta}) - \Lambda\eta_{t-1} + \sqrt{2\beta^{-1}\Lambda}\,\xi_{t-1},$$

where $\xi_{t-1} \sim \mathcal{N}(0, I)$ and

$$\eta_{t-1} = \left[G_t(\widehat{\theta}) + \nabla G_t(\widehat{\theta})(\psi_{t-1} - \widehat{\theta})\right] - \mathbb{E}\left[G_t(\widehat{\theta}) + \nabla G_t(\widehat{\theta})(\psi_{t-1} - \widehat{\theta})\right].$$

Iterating forward,

$$\psi_{t+k} - \widehat{\theta} = (I - \Lambda\widehat{H})(\psi_{t+k-1} - \widehat{\theta}) - \Lambda\eta_{t+k-1} + \sqrt{2\beta^{-1}\Lambda}\,\xi_{t+k-1}.$$

Define the lag-$k$ cross-covariance matrix

$$\Xi_k := \mathbb{E}_{\pi_\psi}\left[(\psi_{t+k} - \widehat{\theta})(\psi_t - \widehat{\theta})^\top\right].$$

Then

$$\Xi_k = (I - \Lambda\widehat{H})\Xi_{k-1} - \Lambda\mathbb{E}_{\pi_\theta}\left[\eta_{t+k-1}(\psi_t - \widehat{\theta})^\top\right].$$

Since $\eta_{t+k-1}$ is conditionally mean-zero given the past,

$$\mathbb{E}_{\pi_\theta}\left[\eta_{t+k-1}(\psi_t - \widehat{\theta})^\top\right] = \mathbb{E}\left[\mathbb{E}\left[\eta_{t+k-1}(\psi_t - \widehat{\theta})^\top \mid \mathcal{F}_{t+k-1}\right]\right] = 0.$$

Therefore,

$$\Xi_k = (I - \Lambda\widehat{H})^k\Xi_0 = (I - \Lambda\widehat{H})^k\Sigma_\psi.$$

Consequently, the lag-$k$ autocorrelation for coordinate $i$ is

$$\rho_{k,i} = \frac{\left((I - \Lambda\widehat{H})^k\Sigma_\psi\right)_{ii}}{(\Sigma_\psi)_{ii}}.$$

When $\Lambda = \lambda I$ and $\widehat{H}$ is symmetric positive definite, the eigenvalues of $I - \lambda\widehat{H}$ are $1 - \lambda\mu_i(\widehat{H})$. Hence

$$\|I - \lambda\widehat{H}\|_2 = \max_i |1 - \lambda\mu_i(\widehat{H})|.$$

According to the condition $0 < \lambda < 2/\mu_{\max}(\widehat{H})$, we have $0 < \lambda\mu_i(\widehat{H}) < 2$ for all eigenvalues $\mu_i(\widehat{H})$, and therefore

$$|1 - \lambda\mu_i(\widehat{H})| < 1.$$

Since $\widehat{H}$ is symmetric positive definite, the spectral norm satisfies

$$\|A\|_2 := \sup_{\|x\|_2=1} \|Ax\|_2 = \max_i |\mu_i(A)|.$$

Hence,

$$\|I - \lambda\widehat{H}\|_2 = \max_i |1 - \lambda\mu_i(\widehat{H})| < 1.$$

Therefore, the Neumann series converges and

$$\sum_{k=0}^{\infty}(I - \lambda\widehat{H})^k = (\lambda\widehat{H})^{-1}.$$

Hence,

$$\sum_{k=1}^{\infty} \rho_{k,i} = \frac{\left(((\Lambda\widehat{H})^{-1} - I)\Sigma_\psi\right)_{ii}}{(\Sigma_\psi)_{ii}} = \frac{\left((\Lambda\widehat{H})^{-1}\Sigma_\psi\right)_{ii}}{(\Sigma_\psi)_{ii}} - 1.$$

The coordinate-wise integrated autocorrelation time

$$\tau_{\text{int}}(f_i) := 1 + 2\sum_{k=1}^{\infty} \rho_{k,i}$$

thus satisfies

$$\tau_{\text{int}}(f_i) = 2\frac{\left((\Lambda\widehat{H})^{-1}\Sigma_\psi\right)_{ii}}{(\Sigma_\psi)_{ii}} - 1.$$

Let $w := \Sigma_\psi^{1/2}v$. Then

$$\frac{v^\top(\Lambda\widehat{H})^{-1}\Sigma_\psi v}{v^\top\Sigma_\psi v} = \frac{w^\top\left(\Sigma_\psi^{1/2}(\Lambda\widehat{H})^{-1}\Sigma_\psi^{-1/2}\right)w}{w^\top w}.$$

The matrix

$$M := \Sigma_\psi^{1/2}(\Lambda\widehat{H})^{-1}\Sigma_\psi^{-1/2}$$

is similar to $(\Lambda\widehat{H})^{-1}$, hence $\mu_{\max}(M) = \mu_{\max}((\Lambda\widehat{H})^{-1}) = 1/\mu_{\min}(\Lambda\widehat{H})$.

By Rayleigh–Ritz,

$$\sup_{v\neq 0}\frac{v^\top(\Lambda\widehat{H})^{-1}\Sigma_\psi v}{v^\top\Sigma_\psi v} = \sup_{w\neq 0}\frac{w^\top M w}{w^\top w} = \lambda_{\max}(M) = \frac{1}{\lambda_{\min}(\Lambda\widehat{H})}.$$

Then we have

$$\tau := \sup_{v}\tau_{\text{int}}(f_v) = 2\cdot\frac{1}{\mu_{\min}(\Lambda\widehat{H})} - 1.$$

## F.4. Proof of Theorem 4.5

Since there exists a coupling of $\theta_0 \sim \nu$ and $\psi_0 \sim \nu'$ such that $W_2^2(\nu, \nu') = \mathbb{E}(\|\theta_0 - \psi_0\|^2)$, we assume $(\theta_0, \psi_0)$ follow this joint distribution. Using the recursions for $\theta_t$ and $\psi_t$, and using the assumption that $\Lambda = \lambda I$, we have

$$\|\theta_t - \psi_t\|^2 = \|\theta_{t-1} - \psi_{t-1}\|^2 + \underbrace{\left\|\lambda\left[G_t(\theta_{t-1}) - G_t(\widehat{\theta}) - \nabla G_t(\widehat{\theta})(\psi_{t-1} - \widehat{\theta})\right]\right\|^2}_{\star}$$

$$\underbrace{- 2\lambda\left\langle\theta_{t-1} - \psi_{t-1}, G_t(\theta_{t-1}) - G_t(\widehat{\theta}) - \nabla G_t(\widehat{\theta})(\psi_{t-1} - \widehat{\theta})\right\rangle}_{\star\star}$$

Let $(\mathcal{F}_t)_{t\geq 0}$ denote the filtration associated with $\{(\theta_t, \psi_t)\}_{t\geq 0}$ and $\mathbb{E}_t := \mathbb{E}(\cdot \mid \mathcal{F}_t)$. Let $I$ denote an independent random variable uniformly distributed on $\{1, \ldots, N\}$. We can bound the expected squared error as

$$\mathbb{E}_{t-1}(\star) = \lambda^2\mathbb{E}_{t-1}\left[\left\|G_t(\theta_{t-1}) - G_t(\widehat{\theta}) - \nabla G_t(\widehat{\theta})(\psi_{t-1} - \widehat{\theta})\right\|^2\right]$$

$$\leq \frac{\lambda^2}{B}\sum_{n\in S_t}\mathbb{E}_{t-1}\left[\left\|\left(\nabla\ell_n(\theta_{t-1}) - \nabla\ell_n(\widehat{\theta}) - \mathcal{J}_n(\psi_{t-1} - \widehat{\theta})\right)\right\|^2\right]$$

$$= \lambda^2\mathbb{E}_{t-1}\left[\left\|\left(\nabla\ell_I(\theta_{t-1}) - \nabla\ell_I(\widehat{\theta}) - \mathcal{J}_I(\psi_{t-1} - \widehat{\theta})\right)\right\|^2\right]$$

$$\leq 2\lambda^2\mathbb{E}_{t-1}\left[\|\nabla\ell_I(\theta_{t-1}) - \nabla\ell_I(\psi_{t-1})\|^2\right] + 2\lambda^2\mathbb{E}_{t-1}\left[\left\|\nabla\ell_I(\psi_{t-1}) - \nabla\ell_I(\widehat{\theta}) - \mathcal{J}_I(\psi_{t-1} - \widehat{\theta})\right\|^2\right].$$

It follows from Taylor's remainder theorem and Assumption (B) that

$$\left\|\nabla\ell_n(\psi_{t-1}) - \nabla\ell_n(\widehat{\theta}) - \mathcal{J}_n(\psi_{t-1} - \widehat{\theta})\right\|^2 \leq \frac{M_n^2}{4}\|\psi_{t-1} - \widehat{\theta}\|^4.$$

Using the fact that convexity and $L$-smoothness imply $L$-co-coercivity, we thus obtain

$$\mathbb{E}_{t-1}(\star)$$
$$\leq 2L\lambda^2\mathbb{E}_{t-1}\left[\langle\theta_{t-1} - \psi_{t-1}, \nabla\ell_I(\theta_{t-1}) - \nabla\ell_I(\psi_{t-1})\rangle\right] + \frac{\lambda^2}{2}\mathbb{E}_{t-1}\left[M_I^2\right]\|\psi_{t-1} - \widehat{\theta}\|^4$$
$$= 2L\lambda^2\langle\theta_{t-1} - \psi_{t-1}, \nabla\mathcal{L}(\theta_{t-1}) - \nabla\mathcal{L}(\psi_{t-1})\rangle + \frac{\lambda^2\overline{M^2}}{2}\|\psi_{t-1} - \widehat{\theta}\|^4.$$

Furthermore, for any $c > 0$, and again using Taylor's remainder theorem and Assumption (B), we have

$$\mathbb{E}_{t-1}(\star\star) = -2\lambda\mathbb{E}_{t-1}\left[\left\langle\theta_{t-1} - \psi_{t-1}, \left(\nabla\ell_I(\theta_{t-1}) - \nabla\ell_I(\widehat{\theta}) - \mathcal{J}_I(\psi_{t-1} - \widehat{\theta})\right)\right\rangle\right]$$
$$= -2\lambda\left\langle\theta_{t-1} - \psi_{t-1}, \nabla\mathcal{L}(\theta_{t-1}) - \nabla\mathcal{L}(\widehat{\theta}) - \mathcal{J}\left(\psi_{t-1} - \widehat{\theta}\right)\right\rangle$$
$$\leq -2\lambda\langle\theta_{t-1} - \psi_{t-1}, \nabla\mathcal{L}(\theta_{t-1}) - \nabla\mathcal{L}(\psi_{t-1})\rangle + \lambda\overline{M}\|\theta_{t-1} - \psi_{t-1}\|\|\psi_{t-1} - \widehat{\theta}\|^2$$
$$\leq -2\lambda\langle\theta_{t-1} - \psi_{t-1}, \nabla\mathcal{L}(\theta_{t-1}) - \nabla\mathcal{L}(\psi_{t-1})\rangle + 2\lambda c\|\theta_{t-1} - \psi_{t-1}\|^2 + \frac{\lambda\overline{M}^2}{8c}\|\psi_{t-1} - \widehat{\theta}\|^4.$$

Thus, using Assumption (C) and choosing $c = \mu/2$, we have

$$\mathbb{E}(\|\theta_t - \psi_t\|^2) \leq \left(1 - 2\lambda\mu + 2\lambda c + 2\lambda^2\mu L\right)\mathbb{E}(\|\theta_{t-1} - \psi_{t-1}\|^2) + \left\{\frac{\lambda^2\overline{M^2}}{2} + \frac{\lambda\overline{M}^2}{8c}\right\}\mathbb{E}\left(\|\psi_{t-1} - \widehat{\theta}\|^4\right)$$
$$= \left\{1 - \lambda\mu(1 - 2\lambda L)\right\}\mathbb{E}(\|\theta_{t-1} - \psi_{t-1}\|^2) + \lambda\left\{\frac{\lambda\overline{M^2}}{2} + \frac{\overline{M}^2}{4\mu}\right\}\mathbb{E}\left(\|\psi_{t-1} - \widehat{\theta}\|^4\right).$$

Hence, we obtain the overall bound given in Equation (14).

### F.5. Proof of Corollary 4.6

First, we give a lemma bounding the stationary fourth moment of $\psi_{t-1}$.

**Lemma F.2.** *Under the conditions of Corollary 4.6, if $\hat{\mu}$ and $\hat{L}$ denote, respectively, the smallest and largest eigenvalues of $\widehat{H} = \nabla^2\mathcal{L}(\widehat{\theta})$ and $\lambda \leq \min\{1/(4\hat{\mu}), B\hat{\mu}/(200L^2)\}$, then for $\psi_\infty \sim \pi_\psi$, satisfies*

$$\mathbb{E}(\|\psi_\infty - \widehat{\theta}\|^4) \leq 96\frac{\lambda^2\tau_4^4}{\hat{\mu}^2 B^2} + 24\frac{\lambda\tau_4^2}{\hat{\mu}^2 B\beta}D + 12\frac{D^2}{\hat{\mu}^2\beta^2} + 48\frac{\lambda D(D+2)}{\hat{\mu}\beta^2},$$

*where $\tau_4^4 := N^{-1}\sum_{n=1}^N\|\nabla\ell(x_I, y_I, \widehat{\theta})\|^4$.*

*Proof.* The recursion for $\psi_t$ can be rewritten as

$$\psi_t - \widehat{\theta} = (I - \lambda\widehat{H})(\psi_{t-1} - \widehat{\theta}) - \lambda\eta_{t-1} + \sqrt{2\beta^{-1}\lambda}\,\xi_{t-1}, \tag{F.2}$$

where $\eta_{t-1} := G_t(\widehat{\theta}) + \nabla G_t(\widehat{\theta})(\psi_{t-1} - \widehat{\theta}) - \widehat{H}(\psi_{t-1} - \widehat{\theta})$ and $\xi_t \sim \mathcal{N}(0, I_D)$.

Since the minibatch at time $t$ is independent of $\psi_{t-1}$ and $\mathbb{E}[G_t(\widehat{\theta})] = \nabla\mathcal{L}(\widehat{\theta}) = 0$, $\mathbb{E}[\nabla G_t(\widehat{\theta})] = \nabla^2\mathcal{L}(\widehat{\theta}) = \widehat{H}$, we have $\mathbb{E}_{t-1}[\eta_{t-1}] = 0$. Using the multinomial formula and the fact that the expected gradient is zero at $\widehat{\theta}$, we obtain

$$\mathbb{E}\{\|G_t(\widehat{\theta})\|^4\} \leq \frac{3}{B^2}\underbrace{\mathbb{E}\left\{\|\nabla\ell(x_I, y_I, \widehat{\theta})\|^4\right\}}_{\tau_4^4 :=} \quad \text{and} \quad \mathbb{E}\|G_t(\widehat{\theta})\|^2 \leq \frac{\tau_4^2}{B}; \tag{F.3}$$

Fix $u := \psi_{t-1} - \widehat{\theta}$, which is $\mathcal{F}_{t-1}$-measurable. With minibatch sampling *with replacement*, we can write

$$\nabla G_t(\widehat{\theta}) = \frac{1}{B} \sum_{b=1}^{B} \nabla^2 \ell(x_{I_b}, y_{I_b}, \widehat{\theta}), \qquad I_1, \ldots, I_B \text{ i.i.d. and independent of } \mathcal{F}_{t-1}.$$

Let $H_I := \nabla^2 \ell(x_I, y_I, \widehat{\theta})$ and $H := \mathbb{E}[H_I] = \nabla^2 \mathcal{L}(\widehat{\theta})$. Define the i.i.d. random vectors

$$Z_b := (H_{I_b} - H)u, \qquad b = 1, \ldots, B.$$

Then $\mathbb{E}_{t-1}[Z_b] = 0$ and

$$(\nabla G_t(\widehat{\theta}) - H)u = \frac{1}{B} \sum_{b=1}^{B} Z_b.$$

Moreover, by $L$-smoothness at $\widehat{\theta}$, then $\|H\|_2 \le \mathbb{E}\|H_I\|_2 \le L$, we have $\|H_I - H\|_2 \le \|H_I\|_2 + \|H\|_2 \le 2L$, hence

$$\|Z_b\| \le 2L\|u\|, \qquad \mathbb{E}_{t-1}\|Z_b\|^2 \le 4L^2\|u\|^2, \qquad \mathbb{E}_{t-1}\|Z_b\|^4 \le 16L^4\|u\|^4.$$

Using independence and $\mathbb{E}_{t-1}[Z_b] = 0$, the cross terms vanish:

$$\mathbb{E}_{t-1}\Big\| \frac{1}{B} \sum_{b=1}^{B} Z_b \Big\|^2 = \frac{1}{B^2} \sum_{b=1}^{B} \mathbb{E}_{t-1}\|Z_b\|^2 \le \frac{4L^2}{B}\|u\|^2.$$

Let $S := \sum_{b=1}^{B} Z_b$. Since $\mathbb{E}_{t-1}[Z_b] = 0$ and the $Z_b$'s are independent,

$$\|S\|^2 = \sum_{b=1}^{B} \|Z_b\|^2 + 2 \sum_{1 \le i < j \le B} \langle Z_i, Z_j \rangle.$$

By $\langle Z_i, Z_j \rangle^2 \le \|Z_i\|^2 \|Z_j\|^2$, we have

$$\mathbb{E}_{t-1}\|S\|^4 \le \mathbb{E}_{t-1}\Big( \sum_{b=1}^{B} \|Z_b\|^2 \Big)^2 + 4\,\mathbb{E}_{t-1}\Big( \sum_{i<j} \langle Z_i, Z_j \rangle \Big)^2 \le B\,\mathbb{E}_{t-1}\|Z_1\|^4 + 3B(B-1)\big(\mathbb{E}_{t-1}\|Z_1\|^2\big)^2.$$

Using $(\mathbb{E}_{t-1}\|Z_1\|^2)^2 \le \mathbb{E}_{t-1}\|Z_1\|^4$, we get $\mathbb{E}_{t-1}\|S\|^4 \le 3B^2\,\mathbb{E}_{t-1}\|Z_1\|^4$, hence

$$\mathbb{E}_{t-1}\Big\| \frac{1}{B} \sum_{b=1}^{B} Z_b \Big\|^4 = \frac{1}{B^4} \mathbb{E}_{t-1}\|S\|^4 \le \frac{3}{B^2}\,\mathbb{E}_{t-1}\|Z_1\|^4 \le \frac{3}{B^2} \cdot 16L^4\|u\|^4 = \frac{48L^4}{B^2}\|u\|^4.$$

Then we have

$$\mathbb{E}_{t-1}\big\|(\nabla G_t(\widehat{\theta}) - \widehat{H})(\psi_{t-1} - \widehat{\theta})\big\|^2 \le \frac{4L^2}{B}\|\psi_{t-1} - \widehat{\theta}\|^2, \tag{F.4}$$

$$\mathbb{E}_{t-1}\big\|(\nabla G_t(\widehat{\theta}) - \widehat{H})(\psi_{t-1} - \widehat{\theta})\big\|^4 \le \frac{48L^4}{B^2}\|\psi_{t-1} - \widehat{\theta}\|^4. \tag{F.5}$$

Since $\eta_{t-1} = G_t(\widehat{\theta}) + (\nabla G_t(\widehat{\theta}) - \widehat{H})(\psi_{t-1} - \widehat{\theta})$, the inequalities $\|a+b\|^2 \le 2\|a\|^2 + 2\|b\|^2$ and $\|a+b\|^4 \le 8\|a\|^4 + 8\|b\|^4$ combined with Equation (F.3), Equation (F.4), Equation (F.5) yield

$$\mathbb{E}_{t-1}\|\eta_{t-1}\|^2 \le \frac{2\tau_4^2}{B} + \frac{8L^2}{B}\|\psi_{t-1} - \widehat{\theta}\|^2, \tag{F.6}$$

$$\mathbb{E}_{t-1}\|\eta_{t-1}\|^4 \le \frac{24\tau_4^4}{B^2} + \frac{384L^4}{B^2}\|\psi_{t-1} - \widehat{\theta}\|^4.$$

By Equation (F.2), $\mathbb{E}_{t-1}[\eta_{t-1}] = 0$, $\mathbb{E}[\xi_{t-1}] = 0$, Equation (F.6), and $\|I - \lambda\widehat{H}\|_2 \leq 1 - \lambda\hat{\mu}$, we can obtain

$$
\begin{aligned}
\mathbb{E}_{t-1}\|\psi_t - \widehat{\theta}\|^2 &= \|(I - \lambda\widehat{H})(\psi_{t-1} - \widehat{\theta})\|^2 + \lambda^2 \mathbb{E}_{t-1}\|\eta_{t-1}\|^2 + 2\beta^{-1}\lambda D \\
&\leq (1 - \lambda\hat{\mu})^2\|\psi_{t-1} - \widehat{\theta}\|^2 + \lambda^2 \mathbb{E}_{t-1}\|\eta_{t-1}\|^2 + 2\beta^{-1}\lambda D \\
&\leq \Big((1 - \lambda\hat{\mu})^2 + 8\lambda^2 L^2/B\Big)\|\psi_{t-1} - \widehat{\theta}\|^2 + 2\lambda^2\tau_4^2/B + 2\beta^{-1}\lambda D.
\end{aligned}
$$

Letting $t \to \infty$ yields

$$
\mathbb{E}\|\psi_\infty - \widehat{\theta}\|^2 \leq \frac{\frac{2\lambda^2\tau_4^2}{B} + 2\beta^{-1}\lambda D}{\lambda\hat{\mu}(2 - \lambda\hat{\mu}) - \frac{8\lambda^2 L^2}{B}}.
$$

Since $\lambda \leq 1/4\hat{\mu}$, we have $2 - \lambda\hat{\mu} \geq 7/4$ and $\frac{8\lambda^2 L^2}{B} \leq \frac{1}{25}\lambda\hat{\mu}$, hence

$$
\mathbb{E}\|\psi_\infty - \widehat{\theta}\|^2 \leq \frac{6}{5}\Big(\frac{\lambda\tau_4^2}{\hat{\mu}B} + \frac{D}{\hat{\mu}\beta}\Big). \tag{F.7}
$$

Then we expand the fourth moment of $\psi_t$ and take conditional expectation,

$$
\begin{aligned}
\mathbb{E}_{t-1}\|\psi_t - \widehat{\theta}\|^4 &= \mathbb{E}_{t-1}\|(I - \lambda\widehat{H})(\psi_{t-1} - \widehat{\theta})\|^4 + \mathbb{E}_{t-1}\| - \lambda\eta_{t-1} + \sqrt{2\beta^{-1}\lambda}\xi_{t-1}\|^4 \\
&\quad + 4\,\mathbb{E}_{t-1}\| - \lambda\eta_{t-1} + \sqrt{2\beta^{-1}\lambda}\xi_{t-1}\|^2\Big\langle (I - \lambda\widehat{H})(\psi_{t-1} - \widehat{\theta}), -\lambda\eta_{t-1} + \sqrt{2\beta^{-1}\lambda}\xi_{t-1}\Big\rangle \\
&\quad + 2\,\mathbb{E}_{t-1}\Big(\|(I - \lambda\widehat{H})(\psi_{t-1} - \widehat{\theta})\|^2\| - \lambda\eta_{t-1} + \sqrt{2\beta^{-1}\lambda}\xi_{t-1}\|^2\Big) \\
&\quad + 4\,\mathbb{E}_{t-1}\Big\langle (I - \lambda\widehat{H})(\psi_{t-1} - \widehat{\theta}), -\lambda\eta_{t-1} + \sqrt{2\beta^{-1}\lambda}\xi_{t-1}\Big\rangle^2. \tag{F.8}
\end{aligned}
$$

Using $\langle a, b\rangle^2 \leq \|a\|^2\|b\|^2$ and Cauchy-Schwarz inequality,

$$
\begin{aligned}
\mathbb{E}_{t-1}\|\psi_t - \widehat{\theta}\|^4 &\leq \mathbb{E}_{t-1}\|(I - \lambda\widehat{H})(\psi_{t-1} - \widehat{\theta})\|^4 + 3\mathbb{E}_{t-1}\| - \lambda\eta_{t-1} + \sqrt{2\beta^{-1}\lambda}\xi_{t-1}\|^4 \\
&\quad + 8\,\mathbb{E}_{t-1}\Big(\|(I - \lambda\widehat{H})(\psi_{t-1} - \widehat{\theta})\|^2\| - \lambda\eta_{t-1} + \sqrt{2\beta^{-1}\lambda}\xi_{t-1}\|^2\Big).
\end{aligned}
$$

Moreover,

$$
\begin{aligned}
\mathbb{E}_{t-1}\| - \lambda\eta_{t-1} + \sqrt{2\beta^{-1}\lambda}\xi_{t-1}\|^2 &= \lambda^2\mathbb{E}_{t-1}\|\eta_{t-1}\|^2 + 2\beta^{-1}\lambda D \\
&\leq \lambda^2\Big(\frac{2\tau_4^2}{B} + \frac{8L^2}{B}\|\psi_{t-1} - \widehat{\theta}\|^2\Big) + 2\beta^{-1}\lambda D,
\end{aligned}
$$

and using $\|u + v\|^4 \leq 8\|u\|^4 + 8\|v\|^4$ together with $\mathbb{E}\|\xi\|^4 = D(D + 2)$,

$$
\begin{aligned}
\mathbb{E}_{t-1}\| - \lambda\eta_{t-1} + \sqrt{2\beta^{-1}\lambda}\xi_{t-1}\|^4 &\leq 8\lambda^4\mathbb{E}_{t-1}\|\eta_{t-1}\|^4 + 32\beta^{-2}\lambda^2 D(D + 2) \\
&\leq \frac{192\lambda^4\tau_4^4}{B^2} + \frac{3072\lambda^4 L^4}{B^2}\|\psi_{t-1} - \widehat{\theta}\|^4 + 32\beta^{-2}\lambda^2 D(D + 2). \tag{F.9}
\end{aligned}
$$

Combining Equation (F.8), Equation (F.9) and $\|(I - \lambda\widehat{H})\|_2 \leq 1 - \lambda\hat{\mu}$, we obtain

$$
\begin{aligned}
\mathbb{E}_{t-1}\|\psi_t - \widehat{\theta}\|^4 &\leq \Big((1 - \lambda\hat{\mu})^4 + \frac{64(1 - \lambda\hat{\mu})^2\lambda^2 L^2}{B} + \frac{9216\lambda^4 L^4}{B^2}\Big)\|\psi_{t-1} - \widehat{\theta}\|^4 \\
&\quad + 16(1 - \lambda\hat{\mu})^2\Big(\lambda^2\tau_4^2/B + \lambda D/\beta\Big)\|\psi_{t-1} - \widehat{\theta}\|^2 + 576\lambda^4\tau_4^4/B^2 + 96\beta^{-2}\lambda^2 D(D + 2).
\end{aligned}
$$

Taking full expectation and letting $t \to \infty$ gives

$$
\begin{aligned}
\Big(1 - (1 - \lambda\hat{\mu})^4 - \frac{64(1 - \lambda\hat{\mu})^2\lambda^2 L^2}{B} - \frac{9216\lambda^4 L^4}{B^2}\Big)\mathbb{E}\|\psi_\infty - \widehat{\theta}\|^4 &\leq 16(1 - \lambda\hat{\mu})^2\Big(\frac{\lambda^2\tau_4^2}{B} + \frac{\lambda D}{\beta}\Big)\mathbb{E}\|\psi_\infty - \widehat{\theta}\|^2 \\
&\quad + \frac{576\lambda^4\tau_4^4}{B^2} + 96\beta^{-2}\lambda^2 D(D + 2). \tag{F.10}
\end{aligned}
$$

Under $\lambda\hat{\mu} \leq 1/4$ and $\lambda \leq B\hat{\mu}/(200L^2)$,

$$
\begin{aligned}
1 - (1-\lambda\hat{\mu})^4 - \frac{64(1-\lambda\hat{\mu})^2\lambda^2 L^2}{B} - \frac{9216\lambda^4 L^4}{B^2} &\geq (4\lambda\hat{\mu} - 6\lambda^2\hat{\mu}^2) - 64\lambda^2 L^2/B - 9216\lambda^4 L^4/B^2 \\
&\geq (5/2)\lambda\hat{\mu} - (8/25)\lambda\hat{\mu} - (36/625)\lambda\hat{\mu} \\
&\geq 2\lambda\hat{\mu}.
\end{aligned}
\tag{F.11}
$$

Using $(1-\lambda\hat{\mu})^2 \leq 1$, Equation (F.7), and Equation (F.11) in Equation (F.10) yields

$$
\begin{aligned}
\mathbb{E}\|\psi_\infty - \hat{\theta}\|^4 &\leq \frac{48}{5\hat{\mu}^2}\left(\frac{\lambda\tau_4^2}{B} + \frac{D}{\beta}\right)^2 + 72\frac{\lambda^2\tau_4^4}{\hat{\mu}^2 B^2} + 48\frac{\lambda D(D+2)}{\hat{\mu}\beta^2} \\
&\leq \left(72 + \frac{48}{5}\right)\frac{\lambda^2\tau_4^4}{\hat{\mu}^2 B^2} + \frac{96}{5}\frac{\lambda\tau_4^2}{\hat{\mu}^2 B\beta}D + \frac{48}{5}\frac{D^2}{\hat{\mu}^2\beta^2} + 48\frac{\lambda D(D+2)}{\hat{\mu}\beta^2} \\
&\leq 96\frac{\lambda^2\tau_4^4}{\hat{\mu}^2 B^2} + 24\frac{\lambda\tau_4^2}{\hat{\mu}^2 B\beta}D + 12\frac{D^2}{\hat{\mu}^2\beta^2} + 48\frac{\lambda D(D+2)}{\hat{\mu}\beta^2}.
\end{aligned}
$$

$\square$

*Proof of Corollary 4.6.* Taking the limit $t \to \infty$ and combining with Lemma F.2 yields

$$
\begin{aligned}
W_2^2(\pi_\theta, \pi_\psi) &\leq \frac{\lambda}{1-\bar{\beta}}\left\{\frac{\lambda\overline{M^2}}{2} + \frac{\overline{M}^2}{4\mu}\right\}\mathbb{E}\left\|\psi_\infty - \hat{\theta}\right\|^4 \\
&\leq \underbrace{\frac{2}{\mu}\left\{\frac{\overline{M^2}}{8L} + \frac{\overline{M}^2}{4\mu}\right\}}_{C_0} \times \left\{\frac{96\lambda^2\tau_4^4}{\hat{\mu}^2 B^2} + \frac{24\lambda\tau_4^2}{\hat{\mu}^2 B\beta}D + \frac{12D^2}{\hat{\mu}^2\beta^2} + \frac{48\lambda D(D+2)}{\hat{\mu}\beta^2}\right\} \\
&\leq C_0 \times \left\{\frac{96\lambda^2\tau_4^4}{\hat{\mu}^2 B^2} + \frac{24\lambda\tau_4^2}{\hat{\mu}^2 B\beta}D + \frac{12D^2}{\hat{\mu}^2\beta^2} + \frac{48\lambda D(D+2)}{\hat{\mu}\beta^2}\right\} \\
&\leq \underbrace{\frac{96C_0\tau_4^4}{\hat{\mu}^2}}_{A_0}\frac{\lambda^2}{B^2} + \underbrace{\frac{12C_0\tau_4^2}{\hat{\mu}^2}D}_{A_1}\frac{2\lambda}{B\beta} + \underbrace{\left(\frac{12C_0}{\hat{\mu}^2}D^2 + \frac{48C_0}{\hat{\mu}^2}D(D+2)\right)}_{A_2}\frac{1}{\beta^2} \\
&\leq A^2\left(\frac{\lambda}{B} + \frac{1}{\beta}\right)^2
\end{aligned}
\tag{F.12}
$$

where $A^2 = \max\{A_0, A_1, A_2\}$ $\qquad\square$

## F.6. Proof of Theorem 4.1

*Proof.* Using Corollary 4.6, the proof of Theorem 4.1 is almost immediate. Under Assumptions (A)–(C), there exists a constant $c > 0$ such that $\Sigma_\theta \prec c\lambda I$ and $\Sigma_\psi \prec c\lambda I$ (Dieuleveut et al., 2020, Theorem 4) and therefore $\sigma_{\theta,d}$ and $\sigma_{\psi,d}$ are of order $\lambda^{1/2}$. Hence, it follows from Corollary 4.6 and Equation (13) that the relative errors of the stationary standard deviations and covariance satisfy Equations (9) and (10). $\qquad\square$

# G. Proofs for Momentum Results

## G.1. Proof of Proposition B.1

*Proof.* Let

$$
\eta_{t-1} = \left[G_t(\hat{\theta}) + \nabla G_t(\hat{\theta})(\psi_{t-1} - \hat{\theta})\right] - \mathbb{E}\left[G_t(\hat{\theta}) + \nabla G_t(\hat{\theta})(\psi_{t-1} - \hat{\theta})\right].
$$

Then, we have

$$
\psi_t - \hat{\theta} = \psi_{t-1} - \hat{\theta} - \Lambda\kappa\, m_{t-1} - \Lambda\mathbb{E}\left[G_t(\hat{\theta}) + \nabla G_t(\hat{\theta})(\psi_{t-1} - \hat{\theta})\right] - \Lambda\,\eta_{t-1} + \sqrt{2\beta^{-1}\Lambda}\,\xi_{t-1}.
\tag{G.1}
$$

Since by assumption $\mathcal{R}(\psi) = \frac{1}{2}\psi^\top \Gamma \psi$, we have

$$\mathbb{E}(G_t(\hat{\theta})) = \frac{1}{N}\sum_{n=1}^{N} \nabla \ell\left(x_n, y_n, \hat{\theta}\right) + \frac{1}{N}\Gamma\hat{\theta} = 0.$$

Then Equation (G.1) can be rewritten as

$$\psi_t - \hat{\theta} = \psi_{t-1} - \hat{\theta} - \Lambda\kappa m_{t-1} - \frac{\Lambda}{N}\sum_{n=1}^{N}\left\{\mathcal{J}_n\left(\psi_{t-1} - \hat{\theta}\right)\right\} - \frac{1}{N}\Lambda\Gamma(\psi_{t-1} - \hat{\theta}) - \Lambda\eta_{t-1} + \sqrt{2\beta^{-1}\Lambda}\,\xi_{t-1}$$

$$= \left(I - \Lambda\mathcal{J} - \frac{\Lambda}{N}\Gamma\right)\left(\psi_{t-1} - \hat{\theta}\right) - \Lambda\kappa m_{t-1} - \Lambda\eta_{t-1} + \sqrt{2\beta^{-1}\Lambda}\,\xi_{t-1}$$

$$= \left(I - \Lambda\widehat{H}\right)\left(\psi_{t-1} - \hat{\theta}\right) - \Lambda\kappa m_{t-1} - \Lambda\eta_{t-1} + \sqrt{2\beta^{-1}\Lambda}\,\xi_{t-1}$$

Assuming $\theta_t$ and $m_t$ are jointly sampled from stationary distribution, we have

$$\Sigma_\psi = \mathbb{E}\left[\left(\psi_t - \hat{\theta}\right)\left(\psi_t - \hat{\theta}\right)^\top\right]$$

$$= \left(I - \Lambda\widehat{H}\right)\Sigma_\psi\left(I - \Lambda\widehat{H}\right)^\top + \kappa^2\Lambda M\Lambda + \Lambda\overline{C}_\psi\Lambda - (D + D^\top) + \frac{2\Lambda}{\beta},$$

where $D = \kappa\left(I - \Lambda\widehat{H}\right)\mathbb{E}\left[\left(\psi_{t-1} - \hat{\theta}\right)m_{t-1}^\top\right]\Lambda$, and $M = \mathbb{E}\left[m_{t-1}m_{t-1}^\top\right]$.

The rest of proof mainly follows the proof of Liu et al. (2021, Theorem 3). According to Equation (B.1), we have $\Lambda m_t = \psi_{t-1} - \psi_t$, so

$$\Lambda M\Lambda = \mathbb{E}\left[\left(\psi_{t-1} - \hat{\theta} - \psi_{t-2} + \hat{\theta} - \sqrt{2\beta^{-1}\Lambda}\,\xi_{t-1}\right)\left(\psi_{t-1} - \hat{\theta} - \psi_{t-2} + \hat{\theta}\right)^\top - \sqrt{2\beta^{-1}\Lambda}\,\xi_{t-1}\right]$$

$$= 2\Sigma_\psi - \mathbb{E}\left[\left(\psi_{t-1} - \hat{\theta}\right)\left(\psi_{t-2}^\top - \hat{\theta}\right)\right] - \mathbb{E}\left[\left(\psi_{t-2} - \hat{\theta}\right)\left(\psi_{t-1}^\top - \hat{\theta}\right)\right] + \frac{2\Lambda}{\beta}.$$

and

$$D = \kappa\left(I - \Lambda\widehat{H}\right)\mathbb{E}\left[\left(\psi_{t-1} - \hat{\theta}\right)m_{t-1}^\top\right]\Lambda$$

$$= \kappa\left(I - \Lambda\widehat{H}\right)\mathbb{E}\left[\left(\psi_{t-1} - \hat{\theta}\right)\left(\psi_{t-2} - \psi_{t-1} + \sqrt{2\beta^{-1}\Lambda}\,\xi_{t-1}\right)^\top\right]$$

$$= \kappa\left(I - \Lambda\widehat{H}\right)\left(\mathbb{E}\left[\left(\psi_{t-1} - \hat{\theta}\right)\left(\psi_{t-2} - \hat{\theta}\right)^\top\right] - \Sigma_\psi\right).$$

Note that

$$\mathbb{E}\left[\left(\psi_{t-1} - \hat{\theta}\right)\left(\psi_{t-2} - \hat{\theta}\right)^\top\right] = \mathbb{E}\left[\left(\psi_t - \hat{\theta}\right)\left(\psi_{t-1} - \hat{\theta}\right)^\top\right]$$

$$= \mathbb{E}\left[\left(\left(I - \Lambda\widehat{H}\right)\left(\psi_{t-1} - \hat{\theta}\right) - \Lambda\kappa m_{t-1} - \Lambda\eta_{t-1} + \sqrt{2\beta^{-1}\Lambda}\,\xi_{t-1}\right)\left(\psi_{t-1} - \hat{\theta}\right)^\top\right]$$

$$= \left(I - \Lambda\widehat{H}\right)\Sigma_\psi - \Lambda\kappa\mathbb{E}\left[m_{t-1}\left(\psi_{t-1} - \hat{\theta}\right)^\top\right]$$

$$= \left(I - \Lambda\widehat{H}\right)\Sigma_\psi - \kappa\mathbb{E}\left[\left(\psi_{t-2} - \psi_{t-1}\right)\left(\psi_{t-1} - \hat{\theta}\right)^\top\right]$$

$$= \left(I - \Lambda\widehat{H}\right)\Sigma_\psi + \kappa\Sigma_\psi - \kappa\mathbb{E}\left[\left(\psi_{t-2} - \hat{\theta}\right)\left(\psi_{t-1} - \hat{\theta}\right)^\top\right].$$

Solving, we obtain

$$\mathbb{E}\left[\left(\psi_{t-1} - \hat{\theta}\right)\left(\psi_{t-2} - \hat{\theta}\right)^\top\right] = \frac{1}{1+\kappa}\left[\left(I - \Lambda\widehat{H}\right)\Sigma_\psi + \kappa\Sigma_\psi\right].$$

so we conclude that

$$(1 - \kappa)(\Lambda\widehat{H}\Sigma + \Sigma\widehat{H}\Lambda) + \frac{\kappa}{1 - \kappa^2}(\Lambda\widehat{H}\Lambda\widehat{H}\Sigma + \Sigma\widehat{H}\Lambda\widehat{H}\Lambda) = \Lambda\overline{C}_\psi\Lambda + \frac{1 + \kappa^2}{1 - \kappa^2}\Lambda\widehat{H}\Sigma\widehat{H}\Lambda + (1 + \kappa^2)\frac{2\Lambda}{\beta}.$$

$\square$

### G.2. Proof of Theorem B.3

Since there exists a coupling of $\theta_0 \sim \nu$ and $\psi_0 \sim \nu'$ with $W_2^2(\nu, \nu') = \mathbb{E}\|\theta_0 - \psi_0\|^2$, we take $(\theta_0, \psi_0)$ from this joint distribution, and initialize the momenta at $m_0 = \nu_0 = 0$. The proof proceeds in three steps: first, a coupled $2 \times 2$ linear recursion for $(\mathbb{E}\|\theta_t - \psi_t\|^2, \mathbb{E}\|m_t - \nu_t\|^2)$ with coefficient matrix $A$; second, a weighted Lyapunov function $V_t = \mathbb{E}\|\theta_t - \psi_t\|^2 + \gamma\mathbb{E}\|m_t - \nu_t\|^2$ whose optimal weight $\gamma^\star$ yields the sharpest contraction rate $\bar{\beta} = \rho(A)$; third, iteration of the resulting one-step recursion.

By $L$-smoothness of $\nabla\ell_n$ and $\|a + b\|^2 \leq (1 + c)\|a\|^2 + (1 + 1/c)\|b\|^2$ with $c = (1 - \kappa)/\kappa$,

$$\mathbb{E}_{t-1}\|m_t - \nu_t\|^2 = \mathbb{E}_{t-1}\big\|\kappa(m_{t-1} - \nu_{t-1}) + G_t(\theta_{t-1}) - G_t(\widehat{\theta}) - \nabla G_t(\widehat{\theta})(\psi_{t-1} - \widehat{\theta})\big\|^2$$

$$\leq \kappa\|m_{t-1} - \nu_{t-1}\|^2 + \frac{1}{1 - \kappa}\mathbb{E}_{t-1}\big\|G_t(\theta_{t-1}) - G_t(\widehat{\theta}) - \nabla G_t(\widehat{\theta})(\psi_{t-1} - \widehat{\theta})\big\|^2$$

$$\leq \kappa\|m_{t-1} - \nu_{t-1}\|^2 + \frac{1}{1 - \kappa}\Big(2L^2\|\theta_{t-1} - \psi_{t-1}\|^2 + \frac{\overline{M^2}}{2}\|\psi_{t-1} - \widehat{\theta}\|^4\Big).$$

Similarly, expanding $\mathbb{E}_{t-1}\|\theta_t - \psi_t\|^2$ and bounding the inner-product term via Taylor's theorem and $2ab \leq \mu a^2 + b^2/\mu$,

$$\mathbb{E}_{t-1}\|\theta_t - \psi_t\|^2 \leq \Big(1 - \lambda\mu + \lambda\kappa + \frac{2\lambda^2 L^2}{1 - \kappa}\Big)\|\theta_{t-1} - \psi_{t-1}\|^2 + \lambda\kappa(1 + \lambda)\|m_{t-1} - \nu_{t-1}\|^2$$

$$+ \Big(\frac{\lambda^2\overline{M^2}}{2(1 - \kappa)} + \frac{\lambda\overline{M}^2}{4\mu}\Big)\|\psi_{t-1} - \widehat{\theta}\|^4.$$

Taking full expectation, this yields the linear recursion

$$\begin{pmatrix} \mathbb{E}\|\theta_t - \psi_t\|^2 \\ \mathbb{E}\|m_t - \nu_t\|^2 \end{pmatrix} \leq \underbrace{\begin{pmatrix} 1 - \lambda\mu + \lambda\kappa + 2\lambda^2 L^2/(1 - \kappa) & \lambda\kappa(1 + \lambda) \\ 2L^2/(1 - \kappa) & \kappa \end{pmatrix}}_{A := (a_{ij})_{2\times 2}} \begin{pmatrix} \mathbb{E}\|\theta_{t-1} - \psi_{t-1}\|^2 \\ \mathbb{E}\|m_{t-1} - \nu_{t-1}\|^2 \end{pmatrix} + \begin{pmatrix} \mathcal{A} \\ \mathcal{B} \end{pmatrix} C_{t-1}, \qquad \text{(G.2)}$$

where $\mathcal{A} = \frac{\lambda^2}{2(1 - \kappa)}\overline{M^2} + \frac{\lambda}{4\mu}\overline{M}^2$ and $\mathcal{B} = \frac{\overline{M^2}}{2(1 - \kappa)}$.

For $\gamma > 0$, let $V_t := \mathbb{E}\|\theta_t - \psi_t\|^2 + \gamma\mathbb{E}\|m_t - \nu_t\|^2$. From Equation (G.2),

$$V_t = (a_{11} + \gamma a_{21})\mathbb{E}\|\theta_{t-1} - \psi_{t-1}\|^2 + (a_{12} + \gamma a_{22})\mathbb{E}\|m_{t-1} - \nu_{t-1}\|^2 + (\mathcal{A} + \gamma\mathcal{B})C_{t-1}$$

$$\leq \beta(\gamma)V_{t-1} + (\mathcal{A} + \gamma\mathcal{B})C_{t-1},$$

where $\beta(\gamma) := \max\{a_{11} + \gamma a_{21}, a_{22} + a_{12}/\gamma\}$.

Minimizing $\beta(\gamma)$ over $\gamma > 0$ at $\gamma^\star = \big[(a_{22} - a_{11}) + \sqrt{(a_{11} - a_{22})^2 + 4a_{12}a_{21}}\big]/(2a_{21}) > 0$ gives

$$\bar{\beta} := \frac{1 - \lambda\mu + \frac{2\lambda^2 L^2}{1 - \kappa} + (1 + \lambda)\kappa + \sqrt{X^2 + Y}}{2},$$

where $X := 1 - \lambda\mu + \frac{2\lambda^2 L^2}{1 - \kappa} - (1 - \lambda)\kappa$ and $Y := \frac{8L^2\lambda\kappa(1 + \lambda)}{1 - \kappa}$, and the source coefficient

$$\mathcal{P} := \mathcal{A} + \gamma^\star\mathcal{B} = \frac{\lambda^2}{2(1 - \kappa)}\overline{M^2} + \frac{\lambda}{4\mu}\overline{M}^2 + \frac{\overline{M^2}}{8L^2}\big(\sqrt{X^2 + Y} - X\big). \qquad \text{(G.3)}$$

Since $\mathbb{E}\|\theta_t - \psi_t\|^2 \leq V_t$ and $V_t \leq \bar{\beta}V_{t-1} + \mathcal{P}C_{t-1}$, iterating and using $V_0 = \mathbb{E}\|\theta_0 - \psi_0\|^2$ (as $m_0 = \nu_0 = 0$) gives

$$\mathbb{E}\|\theta_t - \psi_t\|^2 \leq \bar{\beta}^t\,\mathbb{E}\|\theta_0 - \psi_0\|^2 + \mathcal{P}\sum_{s=1}^{t}\bar{\beta}^{t-s}C_{s-1}.$$

### G.3. Proof of Proposition B.2

According to Section F.3, assume that the SG-MCMC iterates have reached stationarity.

Define the lag-$k$ autocovariance

$$\Xi_k := \mathbb{E}_{\pi_\psi}\left[(\psi_{t+k} - \widehat{\theta})(\psi_t - \widehat{\theta})^\top\right].$$

Under stationarity and the linearized dynamics, this autocovariance satisfies

$$\Xi_k = (I - \Lambda\widehat{H})^k\Xi_0 = (I - \Lambda\widehat{H})^k\Sigma_\psi,$$

where $\Sigma_\psi := \Xi_0$ denotes the stationary covariance.

Next, we approximate the stationary covariance of the averaged iterate

$$\bar{\psi}_k := \frac{1}{k}\sum_{k'=1}^{k}\psi_{k'}.$$

Assuming stationarity, its covariance can be computed as

$$\begin{aligned}
\Sigma_\psi^{(k)} &:= \mathbb{E}\left[(\bar{\psi}_k - \widehat{\theta})(\bar{\psi}_k - \widehat{\theta})^\top\right] \\
&= \frac{1}{k^2}\mathbb{E}\left[\left(\sum_{k'=1}^{k}(\psi_{k'} - \widehat{\theta})\right)\left(\sum_{k''=1}^{k}(\psi_{k''} - \widehat{\theta})\right)^\top\right] \\
&= \frac{1}{k^2}\left(k\Sigma_\psi + 2\sum_{k'=1}^{k-1}\Xi_{k'}\right) \\
&= \frac{1}{k^2}\left(k\Sigma_\psi + 2\sum_{k'=1}^{k-1}(I - \Lambda\widehat{H})^{k'}\Sigma_\psi\right).
\end{aligned}$$

### G.4. Proof of Corollary B.4

**Lemma G.1.** *Under the conditions of Corollary B.4, let $\hat{\mu}, \hat{L}$ denote the smallest and largest eigenvalues of $\widehat{H} = \nabla^2\mathcal{L}(\widehat{\theta})$, and let $\underline{U}, \underline{V}$ be constants such that $\mathbb{E}\|\psi_t - \widehat{\theta}\|^2 \leq \underline{U}, \mathbb{E}\|\nu_t\|^2 \leq \underline{V}$. For any step size $\lambda \leq \min\{1/\hat{L}, 1/(4\hat{\mu})\}$ and momentum $\kappa$ as in Corollary B.4, there exists $C > 0$ such that, whenever $M = (m_{ij})$ below satisfies $\rho(M) < 1$, the stationary iterate $\psi_\infty \sim \pi_\psi$ of the linearized proxy with momentum Equation (B.2) satisfies*

$$\mathbb{E}\|\psi_\infty - \widehat{\theta}\|^4 \leq \frac{(1 - m_{22})\mathcal{A} + m_{12}\mathcal{B}}{(1 - m_{11})(1 - m_{22}) - m_{12}m_{21}},$$

*where*

$$\begin{aligned}
m_{11} &:= (1 + \lambda\hat{\mu})(1 - \lambda\hat{\mu})^4 + C\left\{\frac{\lambda^2 L^2}{B} + \hat{\mu}^{-1}\frac{\lambda^3\kappa^2 L^2}{B} + \frac{\lambda^4 L^4}{B^2}\right\}, \\
m_{12} &:= C\left\{\hat{\mu}^{-3}\lambda\kappa^4 + \hat{\mu}^{-1}\frac{\lambda^3\kappa^2 L^2}{B}\right\}, \\
m_{21} &:= C\left\{\hat{L}^4 + \kappa^2\hat{L}^2 + \frac{\hat{L}^2 L^2}{B} + \frac{L^4}{B^2}\right\}, \\
m_{22} &:= C\left\{\kappa^4 + \kappa^2\hat{L}^2\right\}, \\
\mathcal{A} &:= C\left(\frac{\lambda^4\tau_4^4}{B^2} + \frac{\lambda^2(D^2 + 2D)}{\beta^2}\right) + C\left(\frac{\lambda^2\tau_4^2}{B} + \frac{\lambda D}{\beta}\right)\left[(1 + \lambda\hat{\mu})(1 - \lambda\hat{\mu})^2\,\underline{U} + \left(1 + (\lambda\hat{\mu})^{-1}\right)\lambda^2\kappa^2\,\underline{V}\right], \\
\mathcal{B} &:= C\frac{\tau_4^4}{B^2} + C\frac{\tau_4^2}{B}\left(\hat{L}^2\,\underline{U} + \kappa^2\,\underline{V}\right) + C\frac{\kappa^2 L^2}{B}\,\underline{V}.
\end{aligned}$$

*Proof.* The true linearized proxy with momentum at $\theta = \widehat{\theta}$ can be written as

$$\begin{cases} \nu_t = \kappa \nu_{t-1} + \widehat{H}(\psi_{t-1} - \widehat{\theta}) + \eta_{t-1}, \\ \psi_t - \widehat{\theta} = (I - \lambda \widehat{H})(\psi_{t-1} - \widehat{\theta}) - \lambda \kappa \nu_{t-1} - \lambda \eta_{t-1} + \sqrt{2\beta^{-1}\lambda}\, \xi_{t-1}, \end{cases} \tag{G.4}$$

where $\eta_{t-1} := G_t(\widehat{\theta}) + \nabla G_t(\widehat{\theta})(\psi_{t-1} - \widehat{\theta}) - \widehat{H}(\psi_{t-1} - \widehat{\theta})$ and $\xi_{t-1} \sim \mathcal{N}(0, I_D)$.

Following the proof of Theorem 4.5 and $\mathbb{E}_{t-1}[\eta_{t-1}] = 0$, we have

$$\mathbb{E}_{t-1}\|\eta_{t-1}\|^2 \leq \frac{2\tau_4^2}{B} + \frac{8L^2}{B}\|\psi_{t-1} - \widehat{\theta}\|^2, \tag{G.5}$$

$$\mathbb{E}_{t-1}\|\eta_{t-1}\|^4 \leq \frac{24\tau_4^4}{B^2} + \frac{384L^4}{B^2}\|\psi_{t-1} - \widehat{\theta}\|^4. \tag{G.6}$$

Let $e_t := \psi_t - \widehat{\theta}$, Equation (G.4) can be written as

$$e_t = (I - \lambda \widehat{H})e_{t-1} - \lambda \kappa \nu_{t-1} - \lambda \eta_{t-1} + \sqrt{2\beta^{-1}\lambda}\, \xi_{t-1}.$$

Then

$$\mathbb{E}_{t-1}\|e_t\|^2 = \|(I - \lambda \widehat{H})e_{t-1} - \lambda \kappa \nu_{t-1}\|^2 + \lambda^2 \mathbb{E}_{t-1}\|\eta_{t-1}\|^2 + 2\beta^{-1}\lambda D.$$

Under the stepsize condition $\|I - \lambda \widehat{H}\|_2 \leq 1 - \lambda\hat{\mu}$ and by $\|x - y\|^2 \leq (1+a)\|x\|^2 + (1 + a^{-1})\|y\|^2$ $(a > 0)$, we can obtain

$$\mathbb{E}_{t-1}\|e_t\|^2 \leq (1 + \lambda\hat{\mu})(1 - \lambda\hat{\mu})^2\|e_{t-1}\|^2 + (1 + (\lambda\hat{\mu})^{-1})\lambda^2\kappa^2\|\nu_{t-1}\|^2 \\ + \lambda^2 \mathbb{E}_{t-1}\|\eta_{t-1}\|^2 + 2\beta^{-1}\lambda D.$$

By Equation (G.5),

$$\mathbb{E}_{t-1}\|e_t\|^2 \leq \left\{ (1 + \lambda\hat{\mu})(1 - \lambda\hat{\mu})^2 + \frac{8\lambda^2 L^2}{B} \right\} \|e_{t-1}\|^2 \\ + (1 + (\lambda\hat{\mu})^{-1})\lambda^2\kappa^2\|\nu_{t-1}\|^2 + \frac{2\lambda^2\tau_4^2}{B} + 2\beta^{-1}\lambda D. \tag{G.7}$$

Similarly, using $\|x + y\|^2 \leq 2\|x\|^2 + 2\|y\|^2$ yields

$$\mathbb{E}_{t-1}\|\nu_t\|^2 = \|\kappa \nu_{t-1} + \widehat{H}\, e_{t-1}\|^2 + \mathbb{E}_{t-1}\|\eta_{t-1}\|^2 \\ \leq \left(2\hat{L}^2 + 8L^2/B\right)\|e_{t-1}\|^2 + 2\kappa^2\|\nu_{t-1}\|^2 + 2\tau_4^2/B. \tag{G.8}$$

Taking expectations in Equations (G.7) and (G.8) gives the linear recursion

$$\begin{pmatrix} \mathbb{E}\|e_t\|^2 \\ \mathbb{E}\|\nu_t\|^2 \end{pmatrix} \leq \underbrace{\begin{pmatrix} (1 + \lambda\hat{\mu})(1 - \lambda\hat{\mu})^2 + \frac{8\lambda^2 L^2}{B} & (1 + (\lambda\hat{\mu})^{-1})\lambda^2\kappa^2 \\ 2\hat{L}^2 + \frac{8L^2}{B} & 2\kappa^2 \end{pmatrix}}_{A := (a_{ij})_{2\times 2}} \begin{pmatrix} \mathbb{E}\|e_{t-1}\|^2 \\ \mathbb{E}\|\nu_{t-1}\|^2 \end{pmatrix} + \begin{pmatrix} c_e \\ c_\nu \end{pmatrix}, \tag{G.9}$$

where

$$c_e := \frac{2\lambda^2\tau_4^2}{B} + 2\beta^{-1}\lambda D, \qquad c_\nu := \frac{2\tau_4^2}{B}.$$

Under the step-size and momentum restrictions, we bound the entries of $A$.

Observe that

$$a_{11} = (1 + \lambda\hat{\mu})(1 - \lambda\hat{\mu})^2 + \frac{8\lambda^2 L^2}{B},$$

and let $s := \lambda\hat{\mu} \leq \frac{1}{4}$, we have

$$(1+s)(1-s)^2 = 1 - s - s^2 + s^3 \leq 1 - s.$$

Then

$$1 - a_{11} \geq \lambda\hat{\mu} - \frac{8\lambda^2 L^2}{B}.$$

Thus, by choosing the universal constant in the stepsize condition $\lambda \leq \frac{B\hat{\mu}}{CL^2}$ large enough, we have $1 - a_{11} \geq \frac{1}{2}\lambda\hat{\mu}$.

Note that $a_{22} = 2\kappa^2$ and $\kappa \leq 1/2$,

$$1 - a_{22} = 1 - 2\kappa^2 \geq \frac{1}{2}.$$

It remains to control the off-diagonal product:

$$a_{12} = (1 + (\lambda\hat{\mu})^{-1})\lambda^2\kappa^2 \leq C\lambda^3,$$

and

$$a_{21} = 2\hat{L}^2 + 8L^2/B \leq C.$$

For $\lambda \leq (\hat{\mu}/C)^{1/2}$

$$a_{12}a_{21} \leq C\lambda^3 \leq \tfrac{1}{8}\lambda\hat{\mu}.$$

Thus,

$$\Delta_2 := (1 - a_{11})(1 - a_{22}) - a_{12}a_{21} \geq \frac{1}{2}\lambda\hat{\mu} \cdot \frac{1}{2} - \frac{1}{8}\lambda\hat{\mu} = \frac{1}{8}\lambda\hat{\mu} > 0.$$

Together with $a_{11}, a_{22} < 1$ and $A$ entrywise nonnegative, it implies $\rho(A) < 1$.

Starting from $e_0 = 0$ and $\nu_0 = 0$, the recursion gives, for every $t \geq 1$,

$$\begin{pmatrix} \mathbb{E}\|e_t\|^2 \\ \mathbb{E}\|\nu_t\|^2 \end{pmatrix} \leq \sum_{j=0}^{t-1} A^j \begin{pmatrix} c_e \\ c_\nu \end{pmatrix}.$$

Since $\rho(A) < 1$, the geometric series is bounded entrywise by

$$\sum_{j=0}^{t-1} A^j \leq \sum_{j=0}^{\infty} A^j = (I - A)^{-1}.$$

Hence

$$\begin{pmatrix} \mathbb{E}\|e_t\|^2 \\ \mathbb{E}\|\nu_t\|^2 \end{pmatrix} \leq (I - A)^{-1} \begin{pmatrix} c_e \\ c_\nu \end{pmatrix}.$$

Writing

$$I - A = \begin{pmatrix} 1 - a_{11} & -a_{12} \\ -a_{21} & 1 - a_{22} \end{pmatrix},$$

Cramer's rule yields

$$(I - A)^{-1} \begin{pmatrix} c_e \\ c_\nu \end{pmatrix} = \frac{1}{\Delta_2} \begin{pmatrix} (1 - a_{22})c_e + a_{12}c_\nu \\ a_{21}c_e + (1 - a_{11})c_\nu \end{pmatrix}.$$

Therefore, with

$$\underline{U} = \frac{(1 - a_{22})\,c_e + a_{12}\,c_\nu}{\Delta_2}, \qquad \underline{V} = \frac{a_{21}\,c_e + (1 - a_{11})\,c_\nu}{\Delta_2}, \tag{G.10}$$

we have, uniformly for all $t \geq 1$,

$$\mathbb{E}\|\psi_t - \widehat{\theta}\|^2 \leq \underline{U}, \qquad \mathbb{E}\|\nu_t\|^2 \leq \underline{V}.$$

Thus constants as in the statement exist.

If $u$ is deterministic and $v$ is a mean-zero random vector, then

$$\mathbb{E}\|u + v\|^4 \leq \|u\|^4 + 8\|u\|^2\,\mathbb{E}\|v\|^2 + 3\,\mathbb{E}\|v\|^4. \tag{G.11}$$

From Equation (G.4), write

$$e_t = u_{t-1} + w_{t-1},$$

where

$$u_{t-1} := (I - \lambda \widehat{H}) e_{t-1} - \lambda \kappa \nu_{t-1}, \qquad w_{t-1} := -\lambda \eta_{t-1} + \sqrt{2\beta^{-1}\lambda}\, \xi_{t-1}.$$

Conditioning on the $\mathcal{F}_{t-1}$, $u_{t-1}$ is deterministic. Moreover, since $\mathbb{E}_{t-1}[\eta_{t-1}] = 0$ and $\mathbb{E}[\xi_{t-1}] = 0$, we have

$$\mathbb{E}_{t-1}[w_{t-1}] = 0.$$

Applying Equation (G.11) therefore gives

$$\mathbb{E}_{t-1}\|e_t\|^4 \le \|u_{t-1}\|^4 + 8\|u_{t-1}\|^2 \,\mathbb{E}_{t-1}\|w_{t-1}\|^2 + 3\,\mathbb{E}_{t-1}\|w_{t-1}\|^4. \tag{G.12}$$

For the leading term, we apply the weighted Young inequality: for any $\epsilon \in (0,1)$,

$$\|x + y\|^4 \le (1 + \epsilon)\|x\|^4 + C\epsilon^{-3}\|y\|^4.$$

Taking $\epsilon = \lambda\hat{\mu}$, which is admissible under $\lambda\hat{\mu} \le 1/4$, and using

$$x = (I - \lambda \widehat{H}) e_{t-1}, \qquad y = -\lambda \kappa \nu_{t-1},$$

we obtain

$$\|u_{t-1}\|^4 \le (1 + \lambda\hat{\mu})\|(I - \lambda\widehat{H})e_{t-1}\|^4 + C(\lambda\hat{\mu})^{-3}\lambda^4 \kappa^4 \|\nu_{t-1}\|^4.$$

Note that $\|(I - \lambda\widehat{H})e_{t-1}\| \le (1 - \lambda\hat{\mu})\|e_{t-1}\|$, we have

$$\|u_{t-1}\|^4 \le (1 + \lambda\hat{\mu})(1 - \lambda\hat{\mu})^4 \|e_{t-1}\|^4 + C\hat{\mu}^{-3}\lambda\kappa^4 \|\nu_{t-1}\|^4. \tag{G.13}$$

Since $\xi_{t-1}$ is independent of $(\mathcal{F}_{t-1}, \eta_{t-1})$ and mean zero, the cross term vanishes, and using Equation (G.5),

$$\begin{aligned}
\mathbb{E}_{t-1}\|w_{t-1}\|^2 &= \lambda^2 \mathbb{E}_{t-1}\|\eta_{t-1}\|^2 + 2\beta^{-1}\lambda D \\
&\le \lambda^2 \Big(2\tau_4^2/B + 8L^2/B \,\|e_{t-1}\|^2\Big) + 2\beta^{-1}\lambda D.
\end{aligned}$$

Using $\|x + y\|^4 \le 8(\|x\|^4 + \|y\|^4)$, $\mathbb{E}\|\xi_{t-1}\|^4 = D^2 + 2D$ and Equation (G.6),

$$\begin{aligned}
\mathbb{E}_{t-1}\|w_{t-1}\|^4 &\le 8\lambda^4 \mathbb{E}_{t-1}\|\eta_{t-1}\|^4 + 8(2\beta^{-1}\lambda)^2 \mathbb{E}\|\xi_{t-1}\|^4 \\
&\le 8\lambda^4 \Big(24\tau_4^4/B^2 + 384L^4/B^2\|e_{t-1}\|^4\Big) + 32\beta^{-2}\lambda^2 (D^2 + 2D). \tag{G.14}
\end{aligned}$$

From the inequality $\|x - y\|^2 \le (1 + \lambda\hat{\mu})\|x\|^2 + (1 + (\lambda\hat{\mu})^{-1})\|y\|^2$, we obtain

$$\|u_{t-1}\|^2 \le (1 + \lambda\hat{\mu})(1 - \lambda\hat{\mu})^2 \|e_{t-1}\|^2 + \big(1 + (\lambda\hat{\mu})^{-1}\big)\lambda^2 \kappa^2 \|\nu_{t-1}\|^2. \tag{G.15}$$

By Equation (G.15) and $\|e\|^2\|\nu\|^2 \le \frac{1}{2}(\|e\|^4 + \|\nu\|^4)$, it follows that

$$\begin{aligned}
8\|u_{t-1}\|^2 \mathbb{E}_{t-1}\|w_{t-1}\|^2 \le\; & C\left[(1 + \lambda\hat{\mu})(1 - \lambda\hat{\mu})^2 \frac{\lambda^2 L^2}{B} + (1 + (\lambda\hat{\mu})^{-1})\frac{\lambda^4 \kappa^2 L^2}{B}\right]\|e_{t-1}\|^4 \\
& + C\left(\lambda^2\tau_4^2/B + \lambda D/\beta\right)(1 + \lambda\hat{\mu})(1 - \lambda\hat{\mu})^2\|e_{t-1}\|^2 \\
& + C(1 + (\lambda\hat{\mu})^{-1})\frac{\lambda^4 \kappa^2 L^2}{B}\|\nu_{t-1}\|^4 \\
& + C\left(\lambda^2\tau_4^2/B + \lambda D/\beta\right)(1 + (\lambda\hat{\mu})^{-1})\lambda^2\kappa^2\|\nu_{t-1}\|^2. \tag{G.16}
\end{aligned}$$

Combining Equations (G.10), (G.12) to (G.14) and (G.16), then taking full expectation,

$$\mathbb{E}\|e_t\|^4 \le m_{11}\mathbb{E}\|e_{t-1}\|^4 + m_{12}\mathbb{E}\|\nu_{t-1}\|^4 + \mathcal{A}.$$

From Equation (G.4), $\nu_t = b_{t-1} + \eta_{t-1}$ with $b_{t-1} := \kappa\nu_{t-1} + \widehat{H}e_{t-1}$ and $\mathbb{E}_{t-1}[\eta_{t-1}] = 0$. Applying Equation (G.11) gives

$$\mathbb{E}_{t-1}\|\nu_t\|^4 \leq \|b_{t-1}\|^4 + 8\|b_{t-1}\|^2\mathbb{E}_{t-1}\|\eta_{t-1}\|^2 + 3\mathbb{E}_{t-1}\|\eta_{t-1}\|^4.$$

Using $\|a+b\|^2 \leq 2\|a\|^2 + 2\|b\|^2$, $\|\widehat{H}\|_2 = \hat{L}$, and $2ab \leq a^2 + b^2$,

$$\|b_{t-1}\|^4 \leq C(\hat{L}^4 + \kappa^2\hat{L}^2)\|e_{t-1}\|^4 + C(\kappa^4 + \kappa^2\hat{L}^2)\|\nu_{t-1}\|^4.$$

For the cross term, $\|b_{t-1}\|^2 \leq C(\hat{L}^2\|e_{t-1}\|^2 + \kappa^2\|\nu_{t-1}\|^2)$ together with Equation (G.5) gives

$$8\|b_{t-1}\|^2\mathbb{E}_{t-1}\|\eta_{t-1}\|^2 \leq C\left(\frac{\hat{L}^2L^2}{B} + \frac{L^4}{B^2}\right)\|e_{t-1}\|^4 + C\frac{\tau_4^2}{B}\left(\hat{L}^2\|e_{t-1}\|^2 + \kappa^2\|\nu_{t-1}\|^2\right) + C\frac{\kappa^2L^2}{B}\|\nu_{t-1}\|^2.$$

Taking full expectation, bounding the second moments $\|e_{t-1}\|^2$, $\|\nu_{t-1}\|^2$ by $\underline{U}, \underline{V}$ (so that they contribute to $\mathcal{B}$), and using Equation (G.6) for the remaining $\mathbb{E}_{t-1}\|\eta_{t-1}\|^4$ term, yields

$$\mathbb{E}\|\nu_t\|^4 \leq m_{21}\mathbb{E}\|e_{t-1}\|^4 + m_{22}\mathbb{E}\|\nu_{t-1}\|^4 + \mathcal{B}.$$

Thus

$$\begin{pmatrix}\mathbb{E}\|e_t\|^4 \\ \mathbb{E}\|\nu_t\|^4\end{pmatrix} \leq M \begin{pmatrix}\mathbb{E}\|e_{t-1}\|^4 \\ \mathbb{E}\|\nu_{t-1}\|^4\end{pmatrix} + \begin{pmatrix}\mathcal{A} \\ \mathcal{B}\end{pmatrix}.$$

Since $M$ is entrywise nonnegative and $\rho(M) < 1$, $I - M$ is a nonsingular $M$-matrix; hence

$$\Delta_4 := \det(I - M) = (1 - m_{11})(1 - m_{22}) - m_{12}m_{21} > 0$$

and $(I - M)^{-1} = \sum_{j=0}^{\infty} M^j$ is entrywise nonnegative.

As the recursion is a componentwise inequality with nonnegative $M$, then we have

$$\limsup_{t\to\infty} \begin{pmatrix}\mathbb{E}\|e_t\|^4 \\ \mathbb{E}\|\nu_t\|^4\end{pmatrix} \leq (I - M)^{-1} \begin{pmatrix}\mathcal{A} \\ \mathcal{B}\end{pmatrix}.$$

By Cramer's rule, the first component yields

$$\mathbb{E}\|\psi_\infty - \widehat{\theta}\|^4 \leq \frac{(1 - m_{22})\mathcal{A} + m_{12}\mathcal{B}}{(1 - m_{11})(1 - m_{22}) - m_{12}m_{21}}.$$

$\square$

We now prove Corollary B.4. Under the notation of Theorem B.3 and Lemma G.1, combining the two results gives the master inequality

$$W_2^2(\pi_\theta, \pi_\psi) \leq \frac{\mathcal{P}}{1 - \bar{\beta}} \cdot \frac{(1 - m_{22})\mathcal{A} + m_{12}\mathcal{B}}{(1 - m_{11})(1 - m_{22}) - m_{12}m_{21}}. \tag{G.17}$$

Throughout, $C > 0$ denotes a generic constant (depending only on $\hat{\mu}, \hat{L}, L, \mu, c_\kappa$ and possibly changing between displays), while $c_2, c_3, c_4$ and $C_1$ denote the specific constants fixed in the statement and below. We bound the two factors in turn.

For the quantities $X, Y$ of Theorem B.3, the conditions $\kappa = c_\kappa\lambda \leq \frac{1}{2}$ and $\lambda \leq \mu/(c_2L^2)$ (whence $\lambda\mu \leq \mu^2/(c_2L^2) \leq 1/c_2$, using $\mu \leq L$) give $X \geq \frac{1}{4}$ and $Y \leq 32L^2c_\kappa\lambda^2$, hence by $\sqrt{X^2 + Y} - X \leq Y/(2X)$,

$$\sqrt{X^2 + Y} - X \leq \frac{Y}{2X} \leq 64L^2c_\kappa\lambda^2. \tag{G.18}$$

Writing $1 - \bar{\beta} = (1 - \kappa) - \frac{1}{2}(X + \sqrt{X^2 + Y})$ and substituting $X = 1 - \lambda\mu + \frac{2\lambda^2L^2}{1-\kappa} - (1 - \lambda)\kappa$ with $\kappa = c_\kappa\lambda$,

$$1 - \bar{\beta} \geq (1 - \kappa) - X - Y = \lambda\mu - \frac{2\lambda^2L^2}{1 - \kappa} - \lambda\kappa - Y \geq \tfrac{1}{4}\lambda\mu, \tag{G.19}$$

the last step using $\lambda \leq \mu/(c_2 L^2)$. Substituting Equations (G.18) and (G.19) into the definition of $\mathcal{P}$,

$$\frac{\mathcal{P}}{1-\bar{\beta}} \leq \frac{4}{\lambda\mu}\Big[\frac{\lambda^2}{2(1-\kappa)}\overline{M^2} + \frac{\lambda}{4\mu}\overline{M}^2 + \frac{\overline{M^2}}{8L^2}\big(\sqrt{X^2+Y}-X\big)\Big] \leq \frac{\overline{M}^2}{\mu^2} + \frac{4(1+8c_\kappa)}{\mu}\overline{M^2} := C_1. \tag{G.20}$$

Write $S := \lambda\tau_4^2/B + D/\beta$. Since $c_e = 2\lambda S$, $c_\nu = 2\tau_4^2/B$, $a_{21} \leq C$, $1 - a_{11} \leq C\lambda$, and $\Delta_2 \geq \frac{1}{8}\lambda\hat{\mu}$, Cramer's rule gives

$$\underline{\mathsf{U}} \leq CS, \qquad \underline{\mathsf{V}} \leq CS + \frac{C\tau_4^2}{B}.$$

We claim the step-size restrictions imply

$$1 - m_{11} \geq \lambda\hat{\mu}, \qquad 1 - m_{22} \geq \tfrac{1}{2}, \qquad m_{12}m_{21} \leq \tfrac{1}{4}\lambda\hat{\mu}. \tag{G.21}$$

We verify the three inequalities in Equation (G.21) in turn.

For the first, with $s := \lambda\hat{\mu} \leq \frac{1}{4}$,

$$(1+s)(1-s)^4 \leq 1 - \tfrac{19}{8}s, \qquad C\Big\{\tfrac{\lambda^2 L^2}{B} + \hat{\mu}^{-1}\tfrac{\lambda^3\kappa^2 L^2}{B} + \tfrac{\lambda^4 L^4}{B^2}\Big\} \leq \tfrac{11}{8}\lambda\hat{\mu}$$

under $\lambda \leq B\hat{\mu}/(c_3 L^2)$, so subtracting gives $1 - m_{11} \geq \lambda\hat{\mu}$.

For the second, since $\kappa = c_\kappa\lambda \leq \frac{1}{2}$,

$$m_{22} = C\{\kappa^4 + \kappa^2\hat{L}^2\} \leq Cc_\kappa^2\lambda^2(1+\hat{L}^2) \leq \tfrac{1}{2}$$

for $\lambda_0$ and the admissible bound on $c_\kappa$ small enough, so $1 - m_{22} \geq \frac{1}{2}$.

For the third, $m_{12} \leq C\lambda^5$ and $m_{21} \leq C$ give

$$m_{12}m_{21} \leq C\lambda^5 \leq \tfrac{1}{4}\lambda\hat{\mu} \qquad \text{whenever } \lambda \leq (\hat{\mu}/c_4)^{1/4}.$$

Combining Equation (G.21),

$$\Delta_4 = (1-m_{11})(1-m_{22}) - m_{12}m_{21} \geq \tfrac{1}{2}\lambda\hat{\mu} - \tfrac{1}{4}\lambda\hat{\mu} = \tfrac{1}{4}\lambda\hat{\mu} > 0.$$

Since $M$ is entrywise nonnegative with $m_{11}, m_{22} < 1$ and $\det(I-M) = \Delta_4 > 0$, it follows that $\rho(M) < 1$.

Using $\Delta_4 \geq \frac{1}{4}\lambda\hat{\mu}$ and $1 - m_{22} \leq 1$,

$$\frac{(1-m_{22})\mathcal{A} + m_{12}\mathcal{B}}{\Delta_4} \leq C\lambda^{-1}\mathcal{A} + C\lambda^{-1}m_{12}\mathcal{B}.$$

For the first term, since $(1 + (\lambda\hat{\mu})^{-1})\lambda^2\kappa^2 \leq C\lambda^3$, Lemma G.1 and $\lambda \leq 1$ give

$$\lambda^{-1}\mathcal{A} \leq C\Big(\tfrac{\lambda^3\tau_4^4}{B^2} + \tfrac{\lambda(D^2+2D)}{\beta^2} + S^2 + \lambda^3 S^2 + \lambda^3\tfrac{\tau_4^2}{B}S\Big) \leq C\Big(\tfrac{\lambda^2\tau_4^4}{B^2} + \tfrac{\lambda D\tau_4^2}{B\beta} + \tfrac{D^2+2D}{\beta^2}\Big). \tag{G.22}$$

For the second term, $m_{12} \leq C\lambda^5$ and $\mathcal{B} \leq C(\tau_4^4/B^2 + \tau_4^2 S/B)$ give

$$\lambda^{-1}m_{12}\mathcal{B} \leq C\lambda^4\Big(\tfrac{\tau_4^4}{B^2} + \tfrac{\tau_4^2}{B}S\Big) \leq C\Big(\tfrac{\lambda^2\tau_4^4}{B^2} + \tfrac{\lambda D\tau_4^2}{B\beta}\Big). \tag{G.23}$$

Substituting Equations (G.22) and (G.23) and Equation (G.20) into Equation (G.17),

$$W_2^2(\pi_\theta, \pi_\psi) \leq \underbrace{C\,\tau_4^4\,\frac{\lambda^2}{B^2}}_{A_0^\star} + \underbrace{C\,\tau_4^2 D\,\frac{\lambda}{B\beta}}_{A_1^\star} + \underbrace{C\,(D^2+2D)\,\frac{1}{\beta^2}}_{A_2^\star}$$

$$\leq A_\star^2\Big(\tfrac{\lambda}{B} + \tfrac{1}{\beta}\Big)^2, \tag{G.24}$$

where $A_\star^2 := \max\{A_0^\star, A_1^\star, A_2^\star\}$ depends only on $(\mu, L, \hat{\mu}, \hat{L}, c_\kappa, \tau_4, D, \overline{M}^2, \overline{M^2})$ and is independent of $(\lambda, B, \beta)$.

