# OpenReview forum: "Accurate Large-sample Uncertainty Quantification using Stochastic Gradient Markov Chain Monte Carlo"
_ICML.cc/2026/Conference — ICML 2026 regular_

### Official Review · Reviewer_zCmX · 2026-03-02

**Soundness:** 2
**Presentation:** 2
**Significance:** 2
**Originality:** 3
**Overall Recommendation:** 4
**Confidence:** 3

**Summary:**

The manuscript addresses the challenge of tuning hyperparameters, such as learning rate and batch size, for SG(L)D to achieve a target stationary covariance for UQ. Traditional theoretical tuning rules rely on continuous time SDE limits, which often fail at practically relevant large batch sizes. To solve this, the authors introduce a discrete time proxy framework. They apply a second order Taylor expansion to the individual observation level losses around the MAP estimate. Because these proxy steps are linear in the parameters, the authors are able to derive a closed form solution for the gradient covariance. Building on this derivation, the paper provides non asymptotic error bounds for the proxy approach and presents a tuning algorithm (Algorithm 1) to match target covariances.

**Compliance With Llm Reviewing Policy:**

Affirmed.

**Final Justification:**

The authors provided substantive follow-up addressing my core concerns: a concrete plan to fix the very misleading framing (W1), a neural network experiment for indications beyond the convex setting (W2), and a NUTS comparison (W3, particularly the complexity trade off is well put). While assumptions remain strong and high-dimensional scaling is limited (especially in the "modern large-scale" setting as claimed in line 671), the revised scope (large-sample, low-to-moderate dimension) is honest and well-supported. I raise my score to 4 **conditional on the authors following through on the substantial revisions promised throughout the discussion**.

**Key Questions For Authors:**

Please address my following questions. Satisfactory responses (in particular to [Q1]) would likely strengthen the paper and clarify key issues that currently limit the impact of the work.

* [Q1] Can you explicitly state the computational complexity (time and memory) of Algorithm 1 in terms of the parameter dimension? Given this complexity, how do you justify the claims regarding its applicability to large scale machine learning?
* [Q2] The paper is partly motivated by the need to circumvent the problem of discretization bias in practical SGLD (which is often not directly considered in continuous time based SDE analyses). However, in my experience, discretization bias is often not a severe issue in reality when practitioners utilize a reasonable learning rate (schedulers) compared to other larger sources of approximation error like initialization or Monte Carlo error. Could you provide evidence on the claim that “Existing theory [...] can become quantitatively inaccurate in these regimes” for  real world ML tasks beyond settings like in Fig 1?
* [Q3] Why was the proposed method not compared against state of the art out of the box MCMC methods like NUTS for the low dimensional experiments? (for the used datasets this is easily possible and for larger dataset size one should compare against competing methods like coreset MCMC on the considered tasks)
* [Q4] Given the strong convexity assumptions, could you relate your method to the (linearized) Laplace approximation?

**Limitations:**

No. The authors have not adequately discussed the computational complexity bottlenecks of their algorithm in the large scale setting (see title). I strongly suggest adding a dedicated Limitations section in the main text that clarifies the maximum parameter dimension this algorithm can realistically handle and acknowledges its potential failure modes on realistic non convex loss landscapes often encountered in modern ML.

**Strengths And Weaknesses:**

## Strengths

* The technical execution and originality of the provided derivations are of high quality.
* The derivation of Theorem 4.3 is neat, as it captures the actual structural properties of subsampling rather than relying on generic Gaussian noise assumptions.
* The focus on the non-asymptotic setting addresses a very relevant gap in the traditional SDE based literature (often focusing on the continuous-time case).

## Weaknesses

* **[W1]** The paper repeatedly frames its contribution as solving problems in "large scale optimization" and "modern practice." (very prominently also in the title) This **framing is highly misleading**. To the best of my understanding, Algorithm 1 requires at least $O(D^2)$ memory and $O(D^3)$ runtime. It requires explicitly computing and storing dense empirical Hessian and covariance matrices, and subsequently solving a continuous Lyapunov equation. For actual large scale UQ where $D$ is large, this is computationally intractable. The framework seems to be strictly limited to low dimensional regimes.
* **[W2]** The paper achieves its non asymptotic bounds by swapping the standard asymptotic limits of SDEs with very strong assumptions on the convexity of the loss. Such (strong) convexity does not hold for realistic modern machine learning use cases like neural networks.
* **[W3]** Given the bold claims regarding large scale UQ, the empirical validation is underwhelming. The experiments are limited to toy linear and Poisson regression problems with dimensions less than 50. For problems of this dimensionality, state of the art out of the box methods like NUTS empirically work exceptionally well even in settings where assumptions like strong convexity are violated. The authors seem to omit test against these standard, robust baselines.
* **[W4]** (Minor) The structural organization of the paper is unusual and in parts disrupts the narrative flow. Specifically, placing the Conclusion in the Appendix is a rather unintuitive formatting choice. Further, I think the main part of the paper would benefit from a dedicated limitations section.

---

> ### Author Rebuttal · Authors · 2026-03-31
>
> We thank the reviewer for recognizing the quality and originality of our work, and for noting that Theorem 4.3 addresses an important gap in SDE-based analyses. We hope the clarifications below address the remaining concerns.
>
> **Q1 (Complexity and Large-scale ML).**
>
> **A1.** For a dense implementation, the offline tuning stage of Algorithm 1 requires $O(D^3)$ time (solving Lyapunov-type equations) and $O(D^2)$ memory (storing covariance and preconditioner matrices). We agree that this limits direct applicability to very high-dimensional settings.
>
> We discuss extensions to high-dimensional and sparse regimes (p.6 and Appendix D) at a theoretical level. In practice, scaling to such settings will likely require structured or approximate methods (e.g., low-rank or diagonal approximations, or trajectory-based estimators), as noted in Appendix A. We will move this discussion to the main text and make these limitations explicit.
>
> By “large-scale,” we primarily refer to regimes with large sample size $N$, where stochastic methods (SGD/SGLD) are necessary. Our contribution is not a fully scalable algorithm for high-dimensional models, but a discrete-time theoretical framework that accurately characterizes stationary covariance and subsampling noise beyond SDE-based approximations.
>
> Algorithm 1 is included to instantiate the theory and support empirical validation. Developing scalable implementations remains an important direction for future work, and we will clarify this positioning in the main text in the camera-ready version.
>
> **Q2 (Discretization bias / evidence on real world ML tasks ).**
>
> **A2.** Thank you for raising this important point. Our work focuses on the regime where SG(L)D is used with moderate-to-large batch sizes, which requires a large learning rate for correct uncertainty quantification. Large learning rates tend to lead to large errors, as our empirical results show. But knowing where this transition occurs a priori is difficult. Moreover, choosing the step size too small to avoid discretization error can lead to slow mixing and poor sampling efficiency. But large step sizes become unstable without guidance such as ours. So, in practice, users often select learning rates to balance the mixing / instability effects, rather than to satisfy conditions required by continuous-time (CT) approximations for the correct stationary distribution.
>
> Our analysis applies after convergence to a high-probability region, so initialization error is not dominant.
>
> Beyond Fig. 1, Section 5.2 (including real-world datasets) shows that CT-based tuning leads to noticeable covariance errors in large-batch settings, while our method consistently recovers the sandwich covariance.
>
> Also for ML community, Theorem 4.3 together with Proposition 4.2 characterizes the stationary covariance under a user-specified fixed learning rate, without relying on vanishing step size assumptions which is validated via both synthetic and real-world dataset in Appendix F.
>
> **Q3 (Why not NUTS).**
>
> **A3.** Methods such as NUTS operate under a fundamentally different objective and computational regime than the one considered in our work. In particular, NUTS is not a stochastic-gradient-based method: it relies on full-batch gradients and incorporates a Metropolis–Hastings correction, ensuring that its stationary distribution is the exact Bayesian posterior. (Hoffman et al., 2014)
>
> In contrast, constant-step SG(L)D without Metropolis–Hastings correction induce a different stationary distribution, which generally does not coincide with the posterior ((Mandt et al., 2017). Our work focuses on this class of stochastic-gradient methods and provides principled tuning rules to match a target stationary covariance—specifically, the sandwich covariance structure that is appropriate under model misspecification.
> Therefore, the goals of the two approaches are different: NUTS is designed to sample from the exact posterior, while our method aims to control and characterize the stationary distribution induced by stochastic gradient dynamics and account for model misspecification. For this reason, our experimental comparisons focus on methods within the same framework (i.e., SGD/SGLD and related covariance approximations), which share both the same objective and computational constraints.
>
> **Q4 (Laplace connection).**
>
> **A4.** The Laplace approximation uses a Gaussian $\mathcal{N}(\hat{\theta}, \hat{J}^{-1})$ to approximate the posterior around the MLE. Under our tuning (Algorithm 1), the stationary distribution of SGD/SGLD is also centered at $\hat{\theta}$, but has covariance $\hat{\mathcal{S}} = \hat{J}^{-1}\hat{\mathcal{I}}\hat{J}^{-1}$ (the sandwich covariance). This coincides with $\hat{J}^{-1}$ only in the well-specified case, where $\hat{\mathcal{I}}=\hat{J}$. We will clarify this connection alongside the Bernstein–von Mises discussion in Section 2.2.

---

> > ### Author Rebuttal · Reviewer_zCmX · 2026-04-02
> >
> > Thank you for engaging with my questions. Q4 is resolved and I appreciate the commitment to fix the organization (answer to Reviewer qDDf).
> >
> > My core concerns remain:
> >
> > 1. W1: You confirm the cubic complexity, then dismiss my comment by defining "large-scale" to mean large n, not large p. This does not match how the title and abstract will be read. Promising to "clarify in camera-ready" is too vague. Could you provide a concrete plan? For example, changing the title, adding explicit dimensionality limits upfront, or restructuring the abstract to accurately scope the contribution?
> >
> > 2. W2 was not addressed at all.
> >
> > 3. W3/Q3: I understand NUTS targets a different stationary distribution (arguably a more desirable one). But for the presented d < 50, NUTS is trivially applicable, and practitioners could combine it with post-hoc sandwich corrections to achieve the same goal. Even if the targets differ, comparing approximation quality against such a baseline would ground the contribution practically, especially since your method appears comparably expensive (potentially even more expensive) for these problem sizes. Alternatively, different experimental setups need to be presented.
> >
> > Given the gap between framing and scope (esp. intransparency about limitations), weak evaluation and presentation issues also noted by other reviewers, I maintain my score for now.

---

> > > ### Author Response · Authors · 2026-04-03
> > >
> > > Thanks for the reviewer's response! We would like to further address the remaining concerns as follows:
> > >
> > > **W1.** Discussion on "Large-scale".
> > >
> > > **A1:** We agree that “large-scale” is misleading. Our results target the large-sample ($N$) regime rather than high-dimensional ($D$) settings. In the camera-ready version, we will:
> > >
> > > 1. Replace “large-scale” with **“large-sample”** to clearly reflect the scope of our contribution.
> > > 2. Explicitly state that our method is designed for regimes with large $N$ and low-to-moderate $D$, and does not target high-dimensional scaling.
> > > 3. Add a discussion of the $O(D^3)$ dependence and its limitations in high-dimensional settings.
> > > 4. Clarify that our method complements, rather than replaces, high-dimensional approaches.
> > >
> > > **W2.** Convex assumption and real-world ML task with neural network
> > >
> > > **A2.1 Convex assumption:** Regarding the convexity assumption, most finite-sample analyses of SG(L)D require at least convexity (e.g., Durmus & Moulines, 2017; Dieuleveut et al., 2020; Brosse et al., 2018). Extending such guarantees to non-convex settings—such as neural networks—typically relies on additional approximations, e.g., NTK or mean-field regimes. Developing comparable finite-sample characterizations beyond convexity remains an important direction for future work.
> > >
> > > **A2.2 Neural Network Experiment:** We evaluate on a real-world neural network task using YearPredictionMSD ($N=463{,}715$), with $10$ standardized features and a two-hidden-layer $\tanh$ network (widths $5,5$, $D=91$). We compare predicted stationary covariances (by different theories in Table 1) against empirical SGD covariances across learning rates.
> > >
> > > As we can see from https://anonymous.4open.science/api/repo/ICML_table_nn-86E8/file/ICML_table_nn.pdf?v=fe88e969. At small learning rates, all methods are comparable. However, as the learning rate increases, continuous-time approximations become quantitatively inaccurate, while our discrete-time method remains accurate.
> > >
> > > In practice, it is unclear whether a chosen learning rate is sufficiently small for SDE-based assumptions to hold, motivating discrete-time corrections in realistic regimes.
> > >
> > > Finally, while our non-asymptotic error analysis assumes convexity, the characterization of the stationary covariance $\Sigma$ and minibatch noise $\bar{C}$ does not rely on this assumption. Empirically, we observe that our discrete-time proxy remains effective for neural networks, suggesting that the approach can potentially extend beyond the convex setting and motivating future work on non-convex analysis.
> > >
> > > We thank the reviewer for highlighting the importance of non-convex settings. We will include these experiments and clarify plans for extending the analysis to non-convex settings with rigorous guarantees as future work in the camera-ready!
> > >
> > > **Q3.** W3/Q3: NUTS.
> > >
> > > **A3:** We agree it is useful to include NUTS as a reference, particularly in settings where the posterior is not well calibrated (e.g., under $\beta$-divergence loss).
> > >
> > > We revisit our experiment in Sec 5.1.
> > >
> > > As we can see from https://anonymous.4open.science/api/repo/ICML_tables-6163/file/ICML_table%201.pdf?v=ce7163b7:
> > > (1) Under log-loss, NUTS matches the exact posterior, validating it as a reference. (2) Under $\beta$-loss, NUTS differs from the Sandwich Gaussian, highlighting posterior–sandwich discrepancy. (3) our tuning consistently yields a stationary covariance closer to sandwich covariance especially when $B$ is large.
> > >
> > >
> > > The proposed preconditioned SG(L)D method has complexity $O(MD^2 + D^3) + O\left(T(BD + D^2)\right),$ there the first term corresponds to a one-time offline cost using a subsample of size $M$, and the second term is the cost of $T$ stochastic gradient iterations with minibatch size $B \ll N$. Since mixing is O(1) epochs ($O(N/B)$ iterations), relative MC error of $\delta$ is achievable with $T = O(N/[B\delta])$ iterations. The $O(D^3)$ term in our method is incurred only once. In the large-sample, low-dimensional regime ($N \gg D^2$), this one-time cost is negligible compared to the sampling cost. So, the overall computational complexity is $O(N[D+D^2/B]/\delta)$. This result suggests a benefit of using large $B$ to make SGLD more efficient (fewer preconditioning operations).
> > >
> > > In contrast, for $K$ iterations, NUTS has complexity $O(KLND),$ since each iteration requires multiple (average $L$) full-data gradient evaluations. Hence, we should expect similar computational complexity for both algorithms, in the *best-case scenario* where NUTS mixes in a constant number of iterations (so take $K = O(1/\delta)$) -- in particular, in a manner independent of $N$ and $D$. If not in the best-case scenario, SGLD could be faster and provide UQ for robust likelihood.
> > >
> > > We will add all of the above into the camera-ready version!

---

### Official Review · Reviewer_KHRV · 2026-03-11

**Soundness:** 3
**Presentation:** 2
**Significance:** 3
**Originality:** 3
**Overall Recommendation:** 4
**Confidence:** 4

**Summary:**

This paper focuses on the uncertainty quantification of the stochastic gradient methods with constant step sizes, especially SGD and SGLD with and without momentum. The motivation is that the existing guidance based on the continuous-time SDE limits or strong statistical asymptotics is unreliable in practically important regimes such as large batch sizes and model misspecification. The authors propose a discrete-time proxy framework and derive predictions for stationary covariance, iteate-average covariance, and integrated autocorrelation time, along with non-asymptotic error bounds forr practical tuning and uncertainty quantification. The numerical experiments help demonstrate the effectiveness of the proposed method and theory, including the misspecified settings and robust generalized Bayes settings using the $\beta$-divergence. Overall, the paper is comparable with previous literature and provides new approaches for UQ in the constant step-size settings.

**Compliance With Llm Reviewing Policy:**

Affirmed.

**Final Justification:**

The authors have addressed my concerns and clarified their contributions compared to the previous literature. Although it still has room for improvement, I think the topic is relevant and the results are interesting. Therefore, my final evaluation is that this paper is at the boarder line of acceptance.

**Key Questions For Authors:**

Please see the weaknesses part above.

**Limitations:**

Yes.

**Strengths And Weaknesses:**

Strengths:

(1) The motivation is clear. The practical tuning of SG-MCMC is indeed complicated and difficult in practice, especially when considering misspecified regimes, which remains a gap in the literature;

(2) The focus on the discrete-time analysis is reasonable. The sequential works by Dieuleveut et al. (2020) provides a new toolkit to analyze SGD-type algorithms from a Markov chain perspective, which is explicitly stated and key idea is borrowed for this paper's analysis;

(3) The paper also considers practical guidance beyond the theoretical contributions, including analyzing the exact relationship between the learning rate matrix $\Lambda$, the stationary covariance $\Sigma_{\psi}$, and the average noise $\\bar{C}_{\\psi}$;

(4) I personally like the misspecification part. The experiments with $\beta$-divergence seems to have a broad impact beyond the well-specified log-loss setting.

Weaknesses:

(1) My major concern is the novelty relative to the previous constant-step-size SGD works. For example, the recent NeurIPS paper by Wei et al. (2025) "Gaussian Approximation and Concentration of Constant Learning-Rate Stochastic Gradient Descent", which characterizes the exact stationary distribution $\pi_{\psi}$. According to the drawbacks of Eq. (5) in the current paper, will the method by Wei et al. (2025) help avoid this issue?

(2) Compared to other literature in the SGD-type of methods, what is the novelty of the current paper for the algorithms with momentum? Any particular techniques proposed are just customized for this type of algorithms? What are the key techniques for the misspecification or $\beta$-divergence rather than merely on discrete-time linearization? A brief summary of these techniques shall be helpful;

(3) Another major concern is the interpretability of the assumptions behind Proposition 4.4. The condition $\\|I-\Lambda \hat{H}\\|<1$ is a contraction or stability condition, but in its current form, it is not directly actionable for practitioners. Could this be translated into explicit step-size (sufficient) conditions?

(4) Some minor suggestions:

(a) For the reference "Authors (2026)", I can tell that this is another submission to ICML 2026. However, if the authors could cite a few other relevant and published results, it would be better for clearness;

(b) The format of the multiple citations is somewhat confusing. They should either be chronically or alphabetically sorted. Another thing is that some references have the url links while the others don't. It should be unified as well.

---

> ### Author Rebuttal · Authors · 2026-03-31
>
> We are glad to see that the reviewer likes our motivation and appreciates the importance of addressing misspecification. Thank you for raising a number of important points that we are happy to clarify.
>
> **Q1.** Whether the drawbacks of Eq. (5) in the current paper, will the method by Wei et al. (2025) help avoid this issue?
>
> **A1.** The method of Wei et al. (2025) does not address the issue with Eq.(5).
> As shown in Lemma 3.3 and Theorem 3.4 of Wei et al. (2025), the stationary covariance $\Sigma$ satisfies $\hat{H} \Sigma + \Sigma H = \lambda \bar{C}$ which is the same covariance structure obtained under diffusion-based (continuous-time) approximations (e.g., Mandt et al., 2017; Negrea et al., 2022; Kushner & Yang, 1993).
> As a result, their characterization does not include the higher-order $\Lambda \hat{H} \Sigma_{\psi} \hat{H} \Lambda$ or ($\lambda^{2}\hat{H} \Sigma_{\psi} \hat{H} $ ) term, which capture finite learning-rate effects intrinsic to discrete-time dynamics. Consequently, similar to SDE-based approaches, it becomes inaccurate when the step size $\lambda$ is non-vanishing.
>
> Regarding the limitation of Eq. (5), the key issue with Eq. (5) is that the stationary noise covariance $\bar{C}$ depends on the stationary distribution and is not explicitly characterized. Prior work either leaves $\bar{C}$ implicit (Dieuleveut et al., 2020), uses heuristic approximations (e.g., $\bar{C}=\frac{1}{B}\hat{H}$ in Liu et al., 2021), or focuses on restricted settings (e.g., Ziyin et al., 2022). In contrast, Theorem 4.3 provides an explicit, computable form of $\bar{C}$ induced by minibatch subsampling.
>
> **Q2.** Compared to other literature in the SGD-type of methods, what is the novelty of the current paper for the algorithms with momentum? Any particular techniques proposed are just customized for this type of algorithms? What are the key techniques for the misspecification rather than merely on discrete-time linearization?
>
> **A2.** Thank you for this question. We summarize our contributions relative to prior work (see also Table 1) along three key aspects:
>
> (1) Beyond diffusion-based approximations.
>  Compared with continuous-time (diffusion-based) analyses (e.g., Mandt et al., 2017; Negrea et al., 2022; Wang et al., 2025), our proxy framework operates directly in discrete time and captures higher-order effects. In particular, our characterization (Proposition 4.2) includes the $\lambda^{2}\hat{H} \Sigma_{\psi} \hat{H} $ term, which is neglected by diffusion approximations but becomes significant for non-vanishing step sizes or large batch sizes.
>
> (2) Explicit characterization of subsampling noise. As we discussed in **A1** on limitations of Eq. (5).
>
> (3) Handling misspecification and general losses.
>  Unlike prior works that rely on well-specified linear model assumptions, our derivations (Proposition 4.2 and Theorem 4.3) do not require well-specified models. As a result, our framework applies to misspecified settings and general loss functions.
>
> (4) Novelty on momentum.
> Our framework extends naturally to momentum methods by analyzing their dynamics in discrete time. Proposition C.1 characterizes the stationary covariance for momentum updates, and Theorem 4.3 provides an explicit form of the minibatch noise covariance $\bar{C}$. Compared to prior work based on diffusion approximations or simplified noise models, this yields a more accurate prediction and tuning of stationary covariance in large step-size or large batch regimes.
>
> We will revise the paper to make these distinctions clearer around Table 1.
>
> **Q3.** Question on assumptions of Proposition 4.4. $||I-\Lambda \widehat{H}||<1$.
>
> **A3.** The condition is not restrictive—it is a standard discrete-time stability requirement ensuring the existence of a stationary distribution (see, e.g., Theorem 4 in dieuleveut2020bridging). In the scalar step-size setting $\Lambda=\lambda I$, it can be written explicitly as
> $
> 0 < \lambda < \frac{2}{\mu_{\max}(\hat{H})},
> $
> where $\mu_{\max}(\hat{H})$ denotes the largest eigenvalue of $\hat{H}$. Under $L$-smoothness, this further simplifies to $0 < \lambda < 2/L$. We will add a clarification in the paper to highlight this more interpretable form, which is consistent with step-size conditions used in Dieuleveut et al., 2020.
>
>
> **Q4.** The format of the multiple citations is somewhat confusing. They should either be chronically or alphabetically sorted. Another thing is that some references have the url links while the others don't. It should be unified as well.
>
> **A4.** Thank you for pointing this out. We will revise the citation formatting to ensure consistency, including ordering multiple citations (e.g., alphabetically) and unifying the inclusion of URLs across all references in the final version.

---

> > ### Author Rebuttal · Reviewer_KHRV · 2026-04-04
> >
> > Thank you very much for the detailed clarification. I will consider raising my score.

---

> > > ### Author Response · Authors · 2026-04-07
> > >
> > > We are very grateful for your time and for your thoughtful and positive evaluation of our work.
> > >
> > > We are glad to hear that we successfully addressed your concerns and clarified our contributions relative to prior work. We are highly encouraged by your recognition of the topic's relevance and the interesting nature of our results.
> > >
> > > To address the remaining room for improvement in the camera-ready, we will implement your valuable feedback by translating the stability condition in Proposition 4.4 into explicit step-size conditions and connect to the literature. Additionally, we will revise the citation formatting for consistency and further clarify our novelty and contributions in the text surrounding Table 1.
> > >
> > > Thank you again for your constructive engagement!

---

### Official Review · Reviewer_2jZr · 2026-03-13

**Soundness:** 4
**Presentation:** 4
**Significance:** 4
**Originality:** 4
**Overall Recommendation:** 5
**Confidence:** 4

**Summary:**

This paper addresses the known limitations of stochastic gradient descent (SGD) and stochastic gradient Langevin dynamics (SGLD) for approximate sampling and uncertainty quantification, despite their practical use in machine learning model optimization. The authors were particularly motivated by the inability of existing tuning guidelines for SG-MCMC to simultaneously resolve issues associated with large batch or step sizes (prominent in practical settings) and those encountered during model misspecification. They tackle this by first presenting a self-contained narrative of existing continuous-time and discrete-time proxies for tuning, revealing their limitations, including inaccurate approximation of the former in large batch-size regimes, failures under model misspecification for both settings, and unavailable estimates for the mixing time or iterate-average distribution.

The authors thus provide a theoretically rigorous discrete-time + exact tuning guideline for SGD and SGLD (with and without momentum) that remains accurate for large batch sizes and misspecified models. To achieve this, they apply a second-order Taylor approximation to each loss term to reformulate the single-update equation and thus enable direct theoretical analysis to obtain a solvable relationship between the stationary covariance $\Sigma_{\psi}$ and step size $\Lambda$, which serves as the differentiating step for their proposed method in the presented tuning algorithm procedures. Through the formulation and assumptions, the authors also show quantitative, non-asymptotic error analyses for practically accurate estimation. Across several simulated and practical experiments, they demonstrate improved tuning guidance with lower calibration error, lower covariance error, and greater generality beyond diffusion-based or quadratic approximations.

**Compliance With Llm Reviewing Policy:**

Affirmed.

**Final Justification:**

The paper tackles an important problem for scalable uncertainty quantification of machine learning models via SG(L)D variants. Although  the dimensionality limitation with exact Hessian or Fisher information computations remains -- which may contradicts the scalability claim--, the paper presents other incentives such as robustness under model-mispecification and applications in  large batch sizes that lend credence to the relevance of the proposed methodology.  From my assessment, these incentives coupled with the authors' effort in theoretical and empirical validations outweigh the fine-details concerns surrounding their assumptions which the authors also connected to the literature.

**Key Questions For Authors:**

1. How is $\Sigma_{\psi}$ computed via the numerical optimization procedure noted by the authors?

2. Can the authors comment on the limitations of their proposed method? If possible, moving the Conclusion and Future Work in Appendix A to the main text would be helpful.

3. Kindly correct the typographical error (repetitive use of solving) in page 7, line 333 as copied here: While our approach requires "solving jointly solving" Equations (11 and (12),.....

**Limitations:**

Kindly state the limitations, as noted in the questions and weaknesses segments. Also, including an impact statement would be standard.

**Strengths And Weaknesses:**

Strengths
=======================

1. This paper pushes SG(L)D toward improved tuning guidance in practical settings. With the continued widespread use of SGD-based approaches in machine learning, this work presents a viable framework for achieving better performance in reliable machine learning amid the ongoing quest for scalable and accurate uncertainty quantification procedures.

2. The paper presents a complete narrative, from motivation to theoretical analysis and empirical validation. The authors’ solid command of the literature, use of mathematical rigor, and presented algorithm for a tuning-free procedure lend credence to its potential application in the field of probabilistic machine learning.

3. The proposed method is mathematically principled, with open avenues for more analytic expressions. In fact, the derived relationship between stationary covariance and step size enables a one-time joint solution, leading to computational efficiencies.

Weaknesses
===================================

1. The paper did not clearly state its limitations or how they might be addressed, which is important for extending this body of work.

---

> ### Author Rebuttal · Authors · 2026-03-31
>
> We appreciate the reviewer’s thoughtful and positive evaluation, as well as the excellent questions.
>
> **Q1.** How is $\Sigma_{\psi}$ computed via the numerical optimization procedure noted by the authors?
>
> **A1.** For ML tasks such as the experiments in Appendix F, $\Sigma_{\psi}$ does not admit a closed-form expression and must be computed numerically. The goal is: for a fixed learning rate $\lambda$, determine the stationary covariance $\Sigma_{\psi}$ of SGD.
>
> To do this, we substitute the stationary noise model Eq.(12) into the stationary covariance equation Eq.(11), which yields a matrix equation of the form
> $$
> F(\Sigma_{\psi}) = 0,
> $$
> where $\Sigma_{\psi}$ is the only unknown.
>
> We solve this system numerically by vectorizing $\Sigma_{\psi}$ and applying $\texttt{scipy.optimize.root}$ with the Powell hybrid method, which is a standard quasi-Newton root-finding algorithm.
>
> Similarly, when targeting a desired stationary covariance (e.g., the sandwich covariance $\hat{\mathcal{S}}$), we instead treat $\lambda$ or $\Lambda$ as the unknown and solve the corresponding equation numerically.
>
> **Q2** Can the authors comment on the limitations of their proposed method? If possible, moving the Conclusion and Future Work in Appendix A to the main text would be helpful.
>
> **A2.** Thank you for pointing this out. Some limitations relate to high-dimensional models,   Methodologically, exact Hessian or Fisher information computations are infeasible, so we would require structured, approximate, or trajectory-based covariance estimators. The accuracy of the current theory can break down when the dimension is large – but not always. We discuss this point at the bottom of p. 6, and will highlight it again in our conclusion.
> Due to space constraints, we moved  the discussion into the appendix. In retrospect ,this section should have been kept in the main text. We will correct this omission in the camera-ready version.
>
> **Q3.** Kindly correct the typographical error
>
> **A3** Thanks for catching that! We will fix it!

---

> > ### Author Rebuttal · Reviewer_2jZr · 2026-04-03
> >
> > I appreciate the clarifications provided by the authors for the numerical optimization procedures. The optimization approach seems pretty standard to me and I have no issues with it. As such, I have increased my confidence to reflect this.
> >
> > I, however, maintain my score as it already reflect my assessment of this paper as a potentially contributive paper to the field - increasing the score further will necessitates a rigorous tightening of the assumptions used, the dimensionality tradeoff with Hessian computations, and other theoretical concerns noted by other Reviewers. Nonetheless, my final score reflects the relevance of the proposed method for robust uncertainty quantification via SG(L)D variants.

---

> > > ### Author Response · Authors · 2026-04-07
> > >
> > > We deeply appreciate the time and rigorous effort you dedicated to evaluating our work.
> > >
> > > In the camera-ready version, we will expand our discussion of the framework's limitations and move the Conclusion and Future Work section to the main text. Within that section, thanks to your insightful feedback alongside our valuable discussions with Reviewers zCmX and qDDf, we will also detail our plans for scaling the framework to high-dimensional problems and extending our error analysis to the non-convex case as future work.
> > >
> > > Thank you once again for your time and constructive guidance!

---

### Official Review · Reviewer_qDDf · 2026-03-13

**Soundness:** 3
**Presentation:** 2
**Significance:** 2
**Originality:** 2
**Overall Recommendation:** 4
**Confidence:** 2

**Summary:**

The paper proposes what it calls a proxy SGD algorithm, namely, a simplified version of the SGD (or SGLD) algorithm in which Analysis of the stationary properties is feasible. The main focus of the approach proposed on the paper is to have a proxy algorithm that remains a valid approximation of SGD even for large batches and under misspecification. It proceeds then to quantify the limiting behavior of the proposed proxy and also provides a Wasserstein bound between SGD distributions and proxy distributions under convexity assumptions.

The paper then shows that tuning SGD according to the limiting behavior of the proxy improves uncertainty quantification and calibration even in misspecified settings.

**Compliance With Llm Reviewing Policy:**

Affirmed.

**Final Justification:**

The reviewers addressed my questions and provided further numerical experiments that I believe increase the value of the paper.

**Key Questions For Authors:**

1. What are precisely the fundamental differences with Dieuleveult et al 2020? As far as I understand, the difference is mainly due to constant step size vs "matrix" step size, which while important, would be unfeasible in real large scale settings for SGD. I do not see, other than the fact that $\Lambda$ is not $\lambda \operatorname{I}$, how the noise is not considered constant.

2. Would it be possible to validate empirically the Wasserstein bound on the problems the authors proposed?

**Limitations:**

Yes, but in the appendix.

**Strengths And Weaknesses:**

* Soudness: The paper is sound and the claims are well supported. Furthermore, the experiments are pertinent and relate directly to the theoretical claims of the paper, although I would appreciate experiments that validate directly the Wasserstein bound.
* Presentation: While the paper is generally well presented, it has made some choices that go again the format of the conference paper. Namely, the conclusion, which is a fundamental part of a conference paper, is left to the appendices, as several details. Therefore, I think the presentation should either be reworked so that it fits better in the conference format or given that it is already a really long paper, that it is submitted to a Journal.
* Significance: The paper furnishes extended results of the proposed proxy which is in itself an interesting proxy for SGD which shows to account for relevant behavior, where the evidence is the overall performance when tuned with the paper's choices. I am struggling to see the difference between the current paper and Dieuleveult et al 2020, which also builds on a proxy algorithm and gives explicit bounds specifically in the quadratic case (which is somehow the proxy algorithm proposed, at least around $\hat{\theta}$).  (See question below)
* Originality: See significancy.

---

> ### Author Rebuttal · Authors · 2026-03-31
>
> We appreciate that reviewer noting the soundness of our claims and the clarity of the writing. We apologize for the conclusion appearing in the appendix. For the camera-ready version we will move it to the main text.
>
> **Q1.** What are precisely the fundamental differences with Dieuleveult et al 2020?
>
> **A1.** Thank you for the question. The fundamental difference can be summarized as follows.
>
> Dieuleveut et al. (2020) analyze constant-step SGD as a Markov chain and characterize its stationary covariance in discrete time. In their formulation, the stochastic gradient noise enters through a stationary covariance term
> $$
> \bar{C} = \int \mathcal{C}(\theta)\,\pi_\lambda(\mathrm{d}\theta),
> $$
> sometimes approximated by $\mathcal{C}(\hat{\theta})$. However, $\bar{C}$ is left implicit and not expressed in terms of the data distribution or minibatch subsampling (e.g. batch size $B$, Hessian of loss $\hat{H}$ or Fisher information $\hat{\mathcal{I}}$), limiting its use for practical tuning.
>
> We provide a discrete-time characterization that is both theoretically consistent and $\textbf{explicit for SG(L)D tuning under minibatch subsampling}$. While Proposition 4.2 recovers the same stationary covariance result as Dieuleveut et al. (2020) in the SGD setting, our key advance lies in modeling the stochastic gradient noise.
>
> $\textbf{Theorem 4.3}$ gives an explicit, computable form of the stationary noise covariance induced by minibatch sampling, directly tied to the data distribution, curvature, and gradient noise. In contrast to prior work where $\bar{C}$ is implicit, this enables practical tuning of preconditioners and step sizes.
>
> As also noted by Reviewer zCmX, "Theorem 4.3 captures the actual structural properties of subsampling rather than relying on generic Gaussian noise assumptions.''
>
> **Q2.** Would it be possible to validate empirically the Wasserstein bound on the problems the authors proposed?
>
> **A2.**
>
> Thank you for the insightful suggestion. We empirically validate the Wasserstein bound in both well-specified and misspecified settings using Poisson regression. Specifically, we consider (i) synthetic Poisson data (well-specified) and (ii) synthetic negative binomial data fitted with a Poisson model (misspecified).
>
> Let $\pi_{\theta}$ denote the stationary distribution of SGD under the original log-loss, and $\pi_{\psi}$ the stationary distribution under the quadratic proxy loss (Eq.(7)). We approximate the Wasserstein distance using a Gaussian plug-in estimator based on empirical means and covariances of SGD iterates after burn-in.
>
> To instantiate the theoretical upper bound, we use plug-in estimates at $\hat{\theta}$: the largest and smallest eigenvalues of the empirical Hessian provide estimates of the smoothness constant $L$ and strong convexity constant $\mu$, while higher-order quantities (e.g., $\overline{M}$, $\overline{M^2}$, and $\tau_4$) are estimated empirically from gradients.
>
> We evaluate $W_2(\pi_\theta,\pi_\psi)$ across different batch sizes $B$ and ratios $\lambda/B$. As shown in https://anonymous.4open.science/r/ICML_figure-7625/image.png, the empirical Wasserstein distance exhibits a clear linear scaling in $\lambda/B$ across all batch sizes. Moreover, in both the well-specified (Poisson) and misspecified (negative binomial) settings, the empirical curves remain consistently below the theoretical upper bound, confirming the validity of the bound. The bound is conservative but captures the correct scaling behavior and dependence on $\lambda/B$.
>
> We will include these empirical validations in the revised version.

---

> > ### Author Rebuttal · Reviewer_qDDf · 2026-04-03
> >
> > Thank you for the clarifications.

---

> > > ### Author Response · Authors · 2026-04-07
> > >
> > > We deeply appreciate the time and rigorous effort you dedicated to evaluating our work. We are very glad to hear that our responses and the additional numerical experiments effectively addressed your questions and added value to the paper.
> > >
> > > Following your constructive suggestions, we will add the empirical validation of the Wasserstein bound to the camera-ready. Furthermore, based on your feedback alongside our valuable discussions with Reviewers 2jZr and zCmX, we will expand our discussion on the limitations of this work and move the Conclusion and Future Work section into the main text.
> > >
> > > Thank you again for your time and valuable insights throughout this process!

---

### Decision · Program_Chairs · 2026-04-30

**Decision:**

Accept (regular)

**Comment:**

The paper proposes a discrete-time approach for tuning SGD/SGLD, provides practical quantitative non-asymptotic error bounds yielding stationary covariance estimates that are sufficiently accurate for practical tuning for the purposes of sampling and uncertainty quantification. Experiments on linear and Poisson regressions (both simulations and real data scenarios) overall show improved covariance calibration and covariance error compared to existing approximations. Statistical presentation of the empirical evidence includes confidence intervals and thus is sufficiently good.

Yet, several reviewers emphasise the framework to be practically limited to low-to-moderate dimensions due to Hessian/covariance handling and the need to solve Lyapunov-type equations. Reviewers also note the strong convexity (quite standard, I would say, though) assumptions that may limit practical applications to non-convex problems. In that sense, I would agree with some of the reviewers' claims that "large-scale" framing may be misleading, as the approach might not work well in high-dimensional neural network regimes. Presentation is also somewhat poor; several reviewers scored 2 for the presentation.  Three reviewers recommend weak accept, while one recommends accept.

I thus follow the majority of the reviewers and recommend weak accept for the paper. Yet, post-rebuttal, I received comments from the reviewers explicitly stating that upon acceptance, major improvements are incorporated. Reviwer's zCmX updated score, specifically, is explicitly conditional on presentation and framing improvements changes being implemented.